# Time-series Generation by Contrastive Imitation

**Daniel Jarrett**
University of Cambridge, UK
daniel.jarrett@maths.cam.ac.uk

**Ioana Bica**
University of Oxford, UK
Alan Turing Institute, UK
ioana.bica@eng.ox.ac.uk

**Mihaela van der Schaar**
University of California, Los Angeles
University of Cambridge, UK
Alan Turing Institute, UK
mv472@cam.ac.uk

## Abstract

Consider learning a generative model for time-series data. The sequential setting poses a unique challenge: Not only should the generator capture the *conditional* dynamics of (stepwise) transitions, but its open-loop rollouts should also preserve the *joint* distribution of (multi-step) trajectories. On one hand, autoregressive models trained by MLE allow learning and computing explicit transition distributions, but suffer from compounding error during rollouts. On the other hand, adversarial models based on GAN training alleviate such exposure bias, but transitions are implicit and hard to assess. In this work, we study a generative framework that seeks to combine the strengths of both: Motivated by a moment-matching objective to mitigate compounding error, we optimize a local (but forward-looking) *transition policy*, where the reinforcement signal is provided by a global (but stepwise-decomposable) *energy model* trained by contrastive estimation. At training, the two components are learned cooperatively, avoiding the instabilities typical of adversarial objectives. At inference, the learned policy serves as the generator for iterative sampling, and the learned energy serves as a trajectory-level measure for evaluating sample quality. By expressly training a policy to imitate sequential behavior of time-series features in a dataset, this approach embodies "*generation by imitation*". Theoretically, we illustrate the correctness of this formulation and the consistency of the algorithm. Empirically, we evaluate its ability to generate predictively useful samples from real-world datasets, verifying that it performs at the standard of existing benchmarks.

## 1 Introduction

Time-series data are ubiquitous in diverse machine learning applications, such as financial, industrial, and healthcare settings. At the same time, lack of public access to data is a recurring obstacle to the development and reproducibility of research in domains where datasets are proprietary [1]. Generating synthetic—but realistic—time-series data is a promising solution [2], and has received increasing attention in recent years, driven by advances in deep learning and generative adversarial networks [3,4].

Owing to the fact that time-series features are generated sequentially, generative modeling in the temporal setting faces a two-pronged challenge: First, a good generator should accurately capture the conditional dynamics of *stepwise* transitions $p(x_t|x_1, ..., x_{t-1})$; this is important, as the faithfulness of any conceivable downstream time-series analysis depends on the learned correlations across both temporal and feature dimensions. Second, however, the recursive rollouts of the generator should also respect the joint distribution of *multi-step* trajectories $p(x_1, ..., x_T)$; this is equally important, as synthetic trajectories that inadvertently wander beyond the support of original data are useless at best.

35th Conference on Neural Information Processing Systems (NeurIPS 2021).

Recent work falls into two main categories. On one hand, *autoregressive models* trained via MLE [5] explicitly factor the distribution of trajectories into a product of conditionals $\prod_t p(x_t|x_1,...,x_{t-1})$. While this allows directly learning and computing such transitions, with finite data this is prone to *compounding errors* during multi-step generation, due to the discrepancy between closed-loop training (i.e. conditioned on ground-truths as inputs) and open-loop sampling (i.e. conditioned on its own previous outputs) [6]. A variety of methods have sought to counteract this problem of exposure bias, employing auxiliary techniques from curriculum learning [7, 8] and adversarial domain adaptation [9, 10]; however, such remedies are not without biases [11], and empirical improvements have been mixed [12–14].

On the other hand, *adversarial models* based on GAN training and its relatives [15–17] directly model the distribution of trajectories $p(x_1,...,x_T)$ [18–20]. To provide a more granular learning signal for the generator, a popular variant matches the induced distribution of sub-trajectories instead, providing stepwise feedback from the discriminator [21, 22]. TimeGAN [12] is the most recent incarnation of this, and operates within a jointly optimized latent space. GAN-based approaches alleviate the risk of compounding errors, and have been applied to banking [23], sensors [24], biosignals [25], and smartgrids [26]. However, the conditional dynamics are only *implicitly learned*, yielding no way of inspecting or assessing the quality of sampled transitions nor trajectories. Moreover, the adversarial objective leads to characteristically challenging optimization—exacerbated by the temporal dimension.

**Three Operations** Consider a probabilistic generative model $p$ for some dataset $\mathcal{D}$. We are generally interested in performing one or more of the following operations: (1) *sampling* a time series $\tau \sim p$, (2) *evaluating* the likelihood $p(\tau)$, and (3) *learning* the model $p$ from a set of i.i.d. samples $\tau$. In light of the preceding, we investigate a generative framework that attempts to fulfill the following criteria:

- Samples should respect both the stepwise *conditional* distributions of features, as well as the *joint* distribution of full trajectories; unlike pure MLE, we wish to avoid multi-step compounding error.
- Evaluating likelihoods should be possible as generic measures of *sample quality* for both transitions and trajectories—often desired for sample comparison, model auditing, or bias correction [27, 28].
- Unlike black-box GAN discriminators, we wish that the evaluator be *decoupled* from any specific sampler, such that the two components can be trained *non-adversarially*, thus may be more stable.

**Contributions** In the sequel, we explore an approach that seeks to satisfy these criteria. We first give precise treatment of the "compounding error" problem, thus motivating a specific trajectory-centric optimization objective from first principles (Section 2). To carry it out, we develop a general training framework and practical algorithm, along with its theoretical justification: We train a forward-looking *transition policy* to imitate the sequential behavior of time series using a stepwise-decomposable *energy model* as reinforcement, giving a method that embodies "*generation by imitation*" (Section 3). Importantly, to understand its strengths and limitations, we compare the method to existing generative models for time-series data, and relate it to imitation learning of sequential behavior (Section 4). Lastly, through experiments with application to real-world time-series datasets, we verify that it generates predictively useful samples that perform at the standard of comparable benchmarks (Section 5).

## 2 Synthetic Time Series

### 2.1 Problem Setup

We operate in the standard discrete-time setting for time series. Let feature vectors $x_t \in \mathcal{X}$ be indexed by time steps $t$, and let a full trajectory of length $T$ be denoted $\tau := (x_1,...,x_T) \in \mathcal{T} := \mathcal{X}^T$. Also, denote with $h_t := (x_1,...,x_{t-1}) \in \mathcal{H} := \cup_{t=1}^{T} \mathcal{X}^t$ the history prior to time $t$. For ease of exposition we shall work with trajectories of fixed lengths $T$, but our results trivially generalize to the case where $T$ itself is a random variable (for instance, by employing padding tokens up to some maximum length).

Consider a dataset $\mathcal{D} := \{\tau_n\}_{n=1}^{N}$ of $N$ trajectories sampled from some true source $s$. We assume the trajectories are generated sequentially by some unknown transition process $\pi_s \in \Delta(\mathcal{X})^{\mathcal{H}}$, such that features at each step $t$ are sampled as $x_t \sim \pi_s(\cdot|h_t)$. In addition to this stepwise conditional, denote with $\mu_s(h) := \frac{1}{T}\sum_t p(h_t = h|\pi_s)$ the normalized occupancy measure—i.e. the distribution of histories induced by $\pi_s$. Intuitively, this is the visitation distribution of "history states" encountered by a generator when navigating about the feature space $\mathcal{X}$ by rolling out policy $\pi_s$. With slight abuse of notation, we may also write $\mu_s(h,x) := \mu_s(h)\pi_s(x|h)$ to indicate the marginal distribution of transitions. Finally, let the joint distribution of full trajectories be denoted by $p_s(\tau) := \prod_t \pi_s(x_t|h_t)$.

The goal is to learn a sequential generator $\pi_\theta$ parameterized as $\theta$ using samples $\tau \sim p_s$ from $\mathcal{D}$, such that $p_\theta \approx p_s$. Note here that we do not assume stationarity of the time-series data, nor stationarity of the transition conditionals; any influence of $t$ is implicit through the dependence of $\pi_s$ (and $\pi_\theta$) on variable-length histories. In line with recent work [14, 20], for simplicity we do not consider static metadata as supplemental inputs or outputs, as these are commonly and easily incorporated via an additional conditioning layer or auxiliary generator [12, 19]. Lastly, note that much recent work on sequential modeling is devoted to domain-specific, *architecture*-level designs for generating audio [29, 30], text [31, 32], and video [33, 34]. In contrast, our work is closer in spirit to [12, 14] in being an agnostic, *framework*-level study applicable to generic tabular data in any time-series setting.

**Measuring Sample Quality** How do we determine the "quality" of a sample? In specialized domains, of course, we often have prior access to *task-specific* metrics such as BLEU or ROUGE scores in text generation [6, 35]—then, the generator can simply be optimized for such scores via standard methods in reinforcement learning [36]. In generic time-series settings, however, the challenge is that any such metric must necessarily be *task-agnostic*, and access to it must necessarily come from learning.

So, for any data source $s$, let us speak of some hypothetical function $f_s : \mathcal{H} \times \mathcal{X} \to [-c, c]$ with $c < \infty$, such that $f_s(h, x)$ gives the quality of any sampled *transition*—that is, any tuple $(h, x)$. Intuitively, we may interpret this as quantifying how "typical" it is for the random process to be in state $h$ and step towards $x$. Likewise, let as also speak of some function $F_s : \mathcal{T} \to [-cT, cT]$ such that $F_s(\tau)$ gives the quality of any sampled *trajectory*. Naturally, in time-series settings where the underlying process is causally-conditioned, it is reasonable to define this as the decomposition $F_s(\tau) := \sum_t f_s(h_t, x_t)$. Now of course, we have no access to the true $F_s$. But clearly, in learning a generative model $p_\theta$ of $p_s$, we wish that the quality of samples $\tau$ drawn from $p_\theta$ and $p_s$ be similar in expectation. More precisely:

**Definition 1 (Expected Quality Difference)** Let $\Delta \bar{F}_s : \Theta \to [-2cT, 2cT]$ denote the *expected quality difference* between $p_s$ and $p_\theta$, where $\Theta$ indicates the space of parameterizations for generator $\pi_\theta$:

$$\Delta \bar{F}_s(\theta) := \mathbb{E}_{\tau \sim p_s} F_s(\tau) - \mathbb{E}_{\tau \sim p_\theta} F_s(\tau) \tag{1}$$

Our objective, then, is to learn a generator $\pi_\theta$ that minimizes the expected quality difference $\Delta \bar{F}_s(\theta)$. Two points bear emphasis. First, we know nothing about $F_s$—beyond it being the sequential aggregate of $f_s$. This challenge uniquely differentiates this agnostic setting from more popular media-specific applications—for which various predefined measures are readily available for supervision. Second, in addition to matching this *expectation* over samples, we also wish to match the *variety* of samples in the original data. After all, we want $p_\theta$ to mimic samples from $p_s$ of different degrees of "typicality". So we should expect to incorporate some measure of entropy, e.g. the commonly used Shannon entropy.

## 2.2 Matching Local Moments

Recall the apparent tradeoff between autoregressive models and adversarial models. In the spirit of the former, suppose we seek to directly learn *transition conditionals* via supervised learning. That is,

$$\arg\min_\theta \mathbb{E}_{h \sim \mu_s} \mathcal{L}(\pi_s(\cdot|h), \pi_\theta(\cdot|h)) \tag{2}$$

Consider the log likelihood loss $\mathcal{L}(\pi_s(\cdot|h), \pi_\theta(\cdot|h)) := \mathbb{E}_{x \sim \pi_s(\cdot|h)} \log \pi_\theta(x|h)$. In the case of exponential family models for $\pi_\theta(\cdot|h)$, a basic result is that this is dual to maximizing its conditional entropy subject to the constraint on feature expectations $\mathbb{E}_{h \sim \mu_s; x \sim \pi_\theta(\cdot|h)} T(x) = \mathbb{E}_{h \sim \mu_s; x \sim \pi_s(\cdot|h)} T(x)$, where $T : \mathcal{X} \to \mathbb{R}$ is some sufficient statistic [37–39]. More generally for deep energy-based models, we have (however, recall that strong duality does not generalize to the nonlinear case; see Appendix A):

$$\arg\min_\theta \left( \mathbb{E}_{\substack{h \sim \mu_s \\ x \sim \pi_\theta(\cdot|h)}} \log \pi_\theta(x|h) + \max_{f \in \mathbb{R}^{\mathcal{H} \times \mathcal{X}}} \left( \mathbb{E}_{\substack{h \sim \mu_s \\ x \sim \pi_s(\cdot|h)}} f(h, x) - \mathbb{E}_{\substack{h \sim \mu_s \\ x \sim \pi_\theta(\cdot|h)}} f(h, x) \right) \right) \tag{3}$$

Note that the moment-matching constraint is *local*—that is, at the level of individual transitions, and all conditioning is based on $h$ from $\mu_s$ alone. This is precisely the "exposure bias": The objective is only ever exposed to inputs $h$ drawn from the (perfect) source distribution $\mu_s$, and is thus unaware of the endogeneity of the (imperfect) synthetic distribution $\mu_\theta$ induced by $\pi_\theta$. This is not desirable since $\pi_\theta$ is rolled out by open-loop sampling at test time. Now, although at the global optimum the moment-matching discrepancy must be zero (i.e. the equality constraint is enforced), in practice there may be a variety of reasons why this is not perfectly achieved (e.g. error in estimating expectations, error in

function approximation, error in optimization, etc). Suppose we could bound how well we are able to enforce the moment-matching constraint; as it turns out, we cannot eliminate error compounding:[1]

**Lemma 1** Let $\max_{f \in \mathbb{R}^{\mathcal{H} \times \mathcal{X}}} \left( \mathbb{E}_{\substack{h \sim \mu_s \\ x \sim \pi_s(\cdot|h)}} f(h, x) - \mathbb{E}_{\substack{h \sim \mu_s \\ x \sim \pi_\theta(\cdot|h)}} f(h, x) \right) \leq \epsilon$. Then $\Delta \bar{F}_s(\theta) \in O(T^2 \epsilon)$.

*Proof.* Appendix A. $\hfill\square$

This reveals the problem with modeling conditionals per se: *Not all mistakes are equal.* An objective like Equation 2 penalizes unrealistic transitions $(h, x)$ by treating all conditioning histories $h$ equally—regardless of how realistic $h$ is to begin with. Clearly, however, we care much less about how $x$ looks like, if the current subsequence $h$ is already highly unlikely (and vice versa). Intuitively, earlier mistakes in a trajectory should weigh more: Once $\pi_\theta$ wanders into areas of $\mathcal{H}$ with low support in $\mu_s$, no amount of "good" transitions will bring the trajectory back to high-likelihood areas of $\mathcal{T}$ under $p_s$.

## 2.3 Matching Global Moments

Now suppose instead that we seek to directly constrain the *trajectory distribution* $p_\theta$ to be similar to $p_s$:

$$\arg\min_\theta \mathcal{L}(p_s, p_\theta) \tag{4}$$

Consider the Kullback-Leibler divergence $\mathcal{L}(p_s, p_\theta) \coloneqq D_{\mathrm{KL}}(p_s \| p_\theta)$. Like before, we know that in the case of exponential family models for $p_\theta$, this is dual to maximizing its entropy subject to the constraint $\mathbb{E}_{\tau \sim p_s} T(\tau) = \mathbb{E}_{\tau \sim p_\theta} T(\tau)$, where $T : \mathcal{T} \to \mathbb{R}$ is some sufficient statistic [40]. More broadly for deep energy-based models, we have $\arg\min_\theta \left( \mathbb{E}_{\tau \sim p_\theta} \log p_\theta(\tau) + \max_{F \in \mathbb{R}^{\mathcal{T}}} (\mathbb{E}_{\tau \sim p_s} F(\tau) - \mathbb{E}_{\tau \sim p_\theta} F(\tau)) \right)$ (but again, recall here that strong duality does not generalize to the nonlinear case; see Appendix A). Now, observe that by definition of occupancy measure $\mu$, for any function $f : \mathcal{H} \times \mathcal{X} \to \mathbb{R}$ it must be the case that $\mathbb{E}_{\tau \sim p} \sum_t f(h_t, x_t) = T \mathbb{E}_{h \sim \mu, x \sim \pi(\cdot|h)} f(h, x)$. Therefore we may equivalently write

$$\arg\min_\theta \left( \mathbb{E}_{\substack{h \sim \mu_\theta \\ x \sim \pi_\theta(\cdot|h)}} \log \pi_\theta(x|h) + \max_{f \in \mathbb{R}^{\mathcal{H} \times \mathcal{X}}} \left( \mathbb{E}_{\substack{h \sim \mu_s \\ x \sim \pi_s(\cdot|h)}} f(h, x) - \mathbb{E}_{\substack{h \sim \mu_\theta \\ x \sim \pi_\theta(\cdot|h)}} f(h, x) \right) \right) \tag{5}$$

Importantly, note that the moment-matching constraint is now *global*—that is, at the level of trajectory rollouts, and $\pi_\theta$ is now conditioned on histories $h$ drawn from its own induced occupancy measure $\mu_\theta$. There is no longer any "exposure bias" here: In order to respect the constraint, not only does $\pi_\theta(\cdot|h)$ have to be close to $\pi_s(\cdot|h)$ for any given $h$, but the occupancy measure $\mu_\theta$ induced by $\pi_\theta$ also has to be close to the occupancy measure $\mu_s$ induced by $\pi_s$. As it turns out, this seemingly minor difference is sufficient to mitigate compounding errors. As before, although at the global optimum the moment-matching discrepancy must be zero, in practice this may not be perfectly achieved. Now, suppose we could bound how well we are able to enforce the moment-matching constraint; but we now have:

**Lemma 2** Let $\max_{f \in \mathbb{R}^{\mathcal{H} \times \mathcal{X}}} \left( \mathbb{E}_{\substack{h \sim \mu_s \\ x \sim \pi_s(\cdot|h)}} f(h, x) - \mathbb{E}_{\substack{h \sim \mu_\theta \\ x \sim \pi_\theta(\cdot|h)}} f(h, x) \right) \leq \epsilon$. Then $\Delta \bar{F}_s(\theta) \in O(T \epsilon)$.

*Proof.* Appendix A. $\hfill\square$

This illustrates why even *transition-centric* adversarial models such as [12,21] have shown promise in generating realistic trajectories [23–26]. First, unlike *trajectory-centric* GANs [18,19] which directly attempt to minimize some form of Equation 4, in transition-centric GANs the objective is to match the transition marginals $\mu_\theta(h, x)$ and $\mu_s(h, x)$—so the discriminator provides more granular feedback to the generator for training. At the same time, we see from Lemma 2 that matching transition marginals is already—indirectly—performing the sort of moment-matching that alleviates compounding error.

Can we be more direct? In Section 3, we shall start by tackling Equation 5 itself. As we shall see, this endeavor gives rise to a technique that trains a conditional policy (for sampling), an energy model (for evaluation), and a non-adversarial framework (for learning)—addressing our three initial criteria.

## 3 Generating by Imitating

First, consider the most straightforward implementation: Let us parameterize $f \in \mathbb{R}^{\mathcal{H} \times \mathcal{X}}$ as $\phi$, and begin with the primal form of Equation 5, which yields the following adversarial learning objective:

---

[1]Lemmas 1 and 2 are similar in spirit to results for error accumulation in imitation by behavioral cloning and distribution matching. See Appendix A; this analogy with imitation learning is formally identified in Section 4.

$$\mathcal{L}(\theta, \phi) := \max_\phi \min_\theta \left( \mathbb{E}_{\substack{h \sim \mu_\theta \\ x \sim \pi_\theta(\cdot|h)}} \log \pi_\theta(x|h) + \mathbb{E}_{\substack{h \sim \mu_s \\ x \sim \pi_s(\cdot|h)}} f_\phi(h, x) - \mathbb{E}_{\substack{h \sim \mu_\theta \\ x \sim \pi_\theta(\cdot|h)}} f_\phi(h, x) \right) \quad (6)$$

It is easy to see that this effectively describes variational training of the energy-based model $p_\phi(\tau) := \exp(F_\phi(\tau) - \log Z_\phi)$—where $F_\phi(\tau) := \sum_t f_\phi(h_t, x_t)$—to approximate the true $p_s(\tau)$, using samples from the variational $p_\theta$. The (outer) energy player is the maximizing agent, and the (inner) policy player is the minimizing agent. The form of this objective naturally prescribes a bilevel optimization procedure in which we perform (gradient-based) updates of $\phi$ with nested (best-response) updates of $\theta$.

## 3.1 Challenges of Learning

Abstractly, of course, training energy models using variational samplers is not new: Multiple works in static domains—such as image modeling—have investigated this approach as a means of bypassing the expense and variance of MCMC sampling [41, 42]. In our setting, however, there is the additional *temporal* dimension: The negative energy $F_\phi(\tau)$ of any trajectory is computed as the sequential composition of stepwise qualities $f_\phi(h_t, x_t)$, and each trajectory sampled from $p_\theta$ must be generated as the sequential rollout of stepwise policies $\pi_\theta(x_t|h_t)$. Consider the gradient update for the energy,

$$\nabla_\phi \mathcal{L} = \mathbb{E}_{\substack{h \sim \mu_s \\ x \sim \pi_s(\cdot|h)}} \nabla_\phi f_\phi(h, x) - \mathbb{E}_{\substack{h \sim \mu_\theta \\ x \sim \pi_\theta(\cdot|h)}} \nabla_\phi f_\phi(h, x) \quad (7)$$

and the inner-loop update for the policy,

$$\arg\min_\theta \mathbb{E}_{\substack{h \sim \mu_\theta \\ x \sim \pi_\theta(\cdot|h)}} \log \pi_\theta(x|h) - \mathbb{E}_{\substack{h \sim \mu_\theta \\ x \sim \pi_\theta(\cdot|h)}} f_\phi(h, x) \quad (8)$$

Note that the max-min optimization requires complete optimization within each inner update in order for the outer update to be correct. Otherwise the gradients will be *biased*, and there would be no guarantee the procedure converges to anything meaningful. Yet unlike in the static setting—for which there exists variety of standard approximations for the inner update [41–44]—here the policy update amounts to entropy-regularized *reinforcement learning* [45–47] using $f_\phi(h_t, x_t)$ as reward function. Thus our first difficulty is computational: Repeatedly performing inner-loop RL is simply infeasible.

Now, an obvious alternative is to dispense with complete policy optimization at each step, and instead to employ *importance sampling* to ensure that the gradients for the energy updates are still unbiased:

$$\nabla_\phi \mathcal{L} = \mathbb{E}_{\tau \sim p_s} \nabla_\phi F_\phi(\tau) - \frac{1}{Z_\phi} \mathbb{E}_{\tau \sim p_\theta} \left[ \frac{\exp(\sum_t f_\phi(h_t, x_t))}{\prod_t \pi_\theta(x_t|h_t)} \nabla_\phi F_\phi(\tau) \right] \quad (9)$$

where the partition function is computed as $Z_\phi = \mathbb{E}_{\tau \sim p_\theta}[\exp(\sum_t f_\phi(h_t, x_t))/\sum_t \pi_\theta(x_t|h_t)]$, and the sampling policy $\pi_\theta$ is no longer required to be perfectly optimized with respect to $f_\phi$. Unfortunately, this strategy simply replaces the original difficulty with a statistical one: As soon as we consider time-series data of non-trivial lengths $T$, the multiplicative effect of each time step on the importance weights means the gradient estimates—albeit unbiased—will have impractically high variance [48,49].

## 3.2 Contrastive Imitation

We now investigate a generative framework that seeks to avoid these difficulties. The key idea is that instead of Equation 7, we shall learn $p_\phi$ by contrasting (real) "positive" samples $\tau \sim p_s$ and (any) "negative" samples $\tau \sim p_\theta$, which—as we shall see—rids us of the requirement that $\pi_\theta$ be fully optimized at each step for learning to be guaranteed. First, let us establish the notion of a "structured classifier":[2]

**Definition 2 (Structured Classification)** Recall the $\pi_\theta$-induced distribution $p_\theta(\tau) := \prod_t \pi_\theta(x_t|h_t)$. Denote with $\tilde{p}_\phi$ the *un-normalized* energy-based model such that $\tilde{p}_\phi(\tau) := \exp(\sum_t f_\phi(h_t, x_t))$, and let $Z_\phi$ be folded into $\phi$ as a learnable parameter. Define the *structured classifier* $d_{\theta,\phi} : \mathcal{T} \rightarrow [0, 1]$:

$$d_{\theta,\phi}(\tau) := \frac{\frac{1}{Z_\phi} \tilde{p}_\phi(\tau)}{\frac{1}{Z_\phi} \tilde{p}_\phi(\tau) + p_\theta(\tau)} \quad (10)$$

---

[2]The idea that density estimation can be performed by logistic regression goes back at least to [50], and formalized as negative sampling [51] and noise-contrastive estimation [52]. Structured classifiers have been studied in the context of imitation learning [53, 54] by analogy with GANs. In the time-series setting, however, we shall see that this approach is equivalent to noise-contrastive estimation with an adaptive noise distribution.

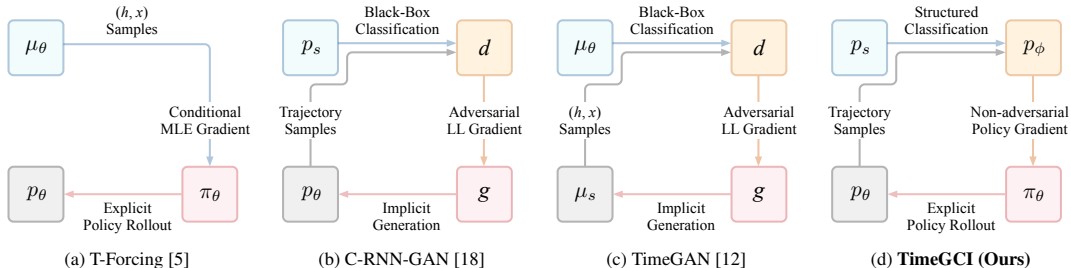

Figure 1: *Comparison of Time-series Generative Models.* Examples of (a) conditional MLE-based autoregressive model, (b) trajectory-centric GAN, and (c) transition-centric GAN. (d) Our proposed technique. See also Table 1.

That is, unlike a black-box classifier that may be arbitrarily parameterized—such as a generic discriminator $d$ in a GAN—here $d_{\theta,\phi}$ is "structured" in that it is modularly parameterized by the embedded energy and policy functions. Now, we shall train $\phi$ such that $d_{\theta,\phi}$ discriminates well between $\tau \sim p_s$ and $\tau \sim p_\theta$—that is, so that the output $d_{\theta,\phi}(\tau)$ represents the (posterior) probability that $\tau$ is real,

$$\mathcal{L}_{\text{energy}}(\phi;\theta) := -\mathbb{E}_{\tau \sim p_s} \log d_{\theta,\phi}(\tau) - \mathbb{E}_{\tau \sim p_\theta} \log \left(1 - d_{\theta,\phi}(\tau)\right) \tag{11}$$

and as before,

$$\mathcal{L}_{\text{policy}}(\theta;\phi) := \mathop{\mathbb{E}}_{\substack{h \sim \mu_\theta \\ x \sim \pi_\theta(\cdot|h)}} \log \pi_\theta(x|h) - \mathop{\mathbb{E}}_{\substack{h \sim \mu_\theta \\ x \sim \pi_\theta(\cdot|h)}} f_\phi(h, x) \tag{12}$$

Why is this better? As we now show formally, each gradient update no longer requires $\theta$ to be optimal for the current value of $\phi$—nor does it require importance sampling—unlike the procedure described by Equation 6. The only requirement is that $p_\theta$ can be sampled and evaluated efficiently, e.g. using learned Gaussian policies as usual, or—should more flexibility be required—with normalizing flow-based policies. As a practical result, this means policy updates can be *interleaved* with energy updates, instead of being *nested* within a repeated inner loop. Specifically, let us establish the following results:

**Proposition 3 (Global Optimality)** Let $f_\phi \in \mathbb{R}^{\mathcal{H} \times \mathcal{X}}$, and let $p_\theta \in \Delta(\mathcal{T})$ be any distribution satisfying positivity: $p_s(\tau) > 0 \Rightarrow p_\theta(\tau) > 0$ (this does not require $\pi_\theta$ be optimal for $f_\phi$). Then $\mathcal{L}_{\text{energy}}(\phi;\theta)$ is globally minimized at $F_\phi(\cdot) - \log Z_\phi = \log p_s(\cdot)$, whence $p_\phi$ is self-normalized with unit integral.

*Proof.* Appendix A. □

This result is intuitive by analogy with noise-contrastive estimation [52, 55]: $\phi$ is learnable as long as negative samples $\tau \sim p_\theta$ cover the support of the true $p_s$. The positivity condition is mild (e.g. take Gaussian policies $\pi_\theta$), and so is the realizability condition (e.g. take neural-networks for $f_\phi$). Importantly, note that at optimality classifier $d_{\theta,\phi}$ is *decoupled* from any specific value of $\theta$; contrast this with generic discriminators $d$ in GANs, which are only ever optimal for the current generator. Now, in practice we must approximate $p_s$ and $p_\theta$ using *finite* samples. In light of this, two questions are immediate: First, does the learned $\phi$ converge to the global optimum as the sample size increases? Second, what role does the "quality" of the policy's samples play in how $\phi$ is learned? For the former:

**Proposition 4 (Asymptotic Consistency)** Let $\phi^*$ denote the minimizer for $\mathcal{L}_{\text{energy}}(\phi;\theta)$, and let $\hat{\phi}^*_M$ denote the minimizer for its finite-data approximation—that is, where the expectations over $p_s$ and $p_\theta$ are approximated by $M$ samples. Then under some mild conditions, as $M$ increases $\hat{\phi}^*_M \xrightarrow{P} \phi^*$.

*Proof.* Appendix A. □

Now for the second question: Clearly if $p_\theta$ were too far from $p_s$, learning would be slow—the job would be too easy for the classifier $d_{\theta,\phi}$, and it may be able to distinguish samples via basic statistics alone. Indeed, in standard noise-contrastive estimation with a *fixed* noise distribution, learning is ineffective in the presence of many variables [56]. Precisely, however, that is why we continuously update the policy itself as an *adaptive* noise distribution: As $p_\phi$ moves closer to $p_s$, so does $p_\theta$—thus providing more "challenging" negative samples.[3] In fact, should we insist on greedily taking each policy update to optimality, we recover a "weighted" version of the original max-min gradient from before:

---

[3]It is easy to see that minimizing Equation 12 equivalently minimizes the reverse KL div. between $p_\phi$ and $p_\theta$.

**Proposition 5 (Gradient Equality)** Let $\phi_k$ be the value taken by $\phi$ after the $k$-th gradient update, and let $\theta_k^*$ denote the associated minimizer for $\mathcal{L}_{\text{policy}}(\theta; \phi_k)$. Suppose $p_\phi$ is already normalized; then

$$\nabla_\phi \mathcal{L}_{\text{energy}}(\phi; \theta_k^*) = -\tfrac{T}{2} \nabla_\phi \mathcal{L}(\theta_k^*, \phi)$$

That is, at $\theta_k^*$ the energy gradient (of Equation 11) recovers the original gradient (from Equation 7). In the general case, suppose $p_\phi$ is un-normalized, such that $p_{\theta_k^*} = p_\phi / K_\phi$ for some constant $K_\phi$; then

$$\nabla_\phi \mathcal{L}_{\text{energy}}(\phi; \theta_k^*) = \tfrac{TK_\phi}{K_\phi+1} \mathbb{E}_{\substack{h \sim \mu_{\theta_k^*} \\ x \sim \pi_{\theta_k^*}(\cdot|h)}} \nabla_\phi f_\phi(h, x) - \tfrac{T}{K_\phi+1} \mathbb{E}_{\substack{h \sim \mu_s \\ x \sim \pi_s(\cdot|h)}} \nabla_\phi f_\phi(h, x)$$

*Proof.* Appendix A.  □

This "weighting" is intuitive: If $p_\phi$ were un-normalized such that $K_\phi > 1$, the energy loss automatically places higher weights on negative samples $h \sim \mu_{\theta_k^*}, x \sim \pi_{\theta_k^*}(\cdot|h)$ to bring it down; conversely, if $p_\phi$ were un-normalized such that $K_\phi < 1$, the energy loss places higher weights on positive samples $h \sim \mu_s, x \sim \pi_s(\cdot|h)$ to bring it up. (If $p_\phi$ were normalized, then $K_\phi = 1$ and the weights are equal). In sum, we have arrived at a framework that learns an explicit sampling policy without exposure bias, a decoupled energy model without nested or saddle-point optimization, and is self-normalizing without importance sampling or estimating the partition function. Figure 1 gives a representative comparison.

### 3.3 Optimization Algorithm

---

**Algorithm 1** Time-series Generation by Contrastive Imitation  ▷ Details in Appendix B

---
1: **Input**: source dataset $\mathcal{D} \approx p_s$, mini-batch size $M$, regularization coefficient $\kappa$, learning rates $\lambda$
2: **Initialize**: replay buffer $\mathcal{B}$, energy parameter $\phi$, policy parameter $\theta$, critic parameter $\psi$
3: **for** each iteration **do**
4:   **for** each policy rollout **do**
5:     $\mathcal{B} \leftarrow \mathcal{B} \cup \{\tau \sim p_\theta\}$                                  ▷ Generate sample
6:   **for** each gradient step **do**
7:     $\theta \leftarrow \theta - \lambda_{\text{actor}} \nabla_\theta \mathcal{L}_{\text{actor}}(\theta; \phi, \psi) + \kappa \nabla_\theta \mathcal{L}_{\text{mle}}(\theta)$   ▷ Update policy
8:     $\phi \leftarrow \phi - \lambda_{\text{energy}} \nabla_\phi \mathcal{L}_{\text{energy}}(\phi; \theta)$               ▷ Update energy
9:     $\psi \leftarrow \psi - \lambda_{\text{critic}} \nabla_\psi \mathcal{L}_{\text{critic}}(\psi; \phi)$                 ▷ Update critic
10: **Output**: learned policy parameter $\theta^*$ and energy parameter $\phi^*$

---

The only remaining choice is the method of policy optimization. Here we employ *soft actor-critic* [57], although in principle any technique will do—the only requirement is that it performs reinforcement learning with *entropy-regularization* [45–47]. To optimize the policy per Equation 12, in addition to the policy "actor" itself, this trains a "critic" to estimate value functions. As usual, the actor takes soft policy improvement steps, minimizing $\mathcal{L}_{\text{actor}}(\theta; \phi, \psi) := \mathbb{E}_{h \sim \mathcal{B}} \mathbb{E}_{x \sim \pi_\theta(\cdot|h)}[\log \pi_\theta(x|h) - Q_\psi(h, x)]$, where $Q_\psi : \mathcal{H} \times \mathcal{X} \to \mathbb{R}$ is the transition-wise soft value function parameterized by $\psi$, and $\mathcal{B}$ is a replay buffer of samples generated by $\pi_\theta$. For stability, the actor is regularized with the conditional MLE loss $\mathcal{L}_{\text{mle}}(\theta) := \mathbb{E}_{x \sim \pi_s(\cdot|h)} \log \pi_\theta(x|h)$. The critic is trained to minimize the soft Bellman residual: $\mathcal{L}_{\text{critic}}(\psi; \phi) := \mathbb{E}_{h,x \sim \mathcal{B}}(Q_\psi(h, x) - f_\phi(h, x) - V_\psi(h'))^2$, where the state-values are bootstrapped as $V_\psi(h') := \mathbb{E}_{x' \sim \pi_\theta(\cdot|h')}[Q_\psi(h', x') - \log \pi_\theta(x'|h')]$. By expressly training an *imitation* policy to mimic time-series behavior using rewards from an energy model trained by *contrastive* learning, we call this framework Time-series Generation by Contrastive Imitation (TimeGCI): See Algorithm 1.

## 4  Discussion

Our theoretical motivations are apparent (Sections 2.2–3.1), and the practical mechanics of optimization are straightforward (Section 3.2–3.3). To understand the strengths and limitations of TimeGCI, two questions remain: First, how does this relate to bread-and-butter imitation learning of sequential decision-making? Second, how does this compare with recent deep generative models for time series?

**Imitation Perspective** In sequential decision-making, *imitation learning* deals with training a policy purely on the basis of demonstrated behavior—that is, with no knowledge of the reward signals that induced the behavior in the first place [58–60]. Consider the standard Markov decision process setting, with states $z \in \mathcal{Z}$, actions $u \in \mathcal{U}$, dynamics $\omega \in \Delta(\mathcal{Z})^{\mathcal{Z} \times \mathcal{U}}$, and rewards $\rho \in \mathbb{R}^{\mathcal{Z} \times \mathcal{U}}$. Classically, imitation learning seeks to minimize the regret $\mathcal{R}_s(\theta) := \mathbb{E}_{\pi_s}[\sum_t \rho(z_t, u_t)] - \mathbb{E}_{\pi_\theta}[\sum_t \rho(z_t, u_t)]$, with $\pi_s, \pi_\theta \in \Delta(\mathcal{U})^{\mathcal{Z}}$ here being the demonstrator and imitator policies, and expectations are taken over episodes generated per $u_t \sim \pi(\cdot|z_t)$ and $z_{t+1} \sim \omega(\cdot|z_t, u_t)$ [61,62]. First, observe that by interpreting $h$ as "states" and $x$ as "actions", our problem setup bears a precise resemblance to imitation learning:

Table 1: *Comparison of Time-series Generative Models*. Examples of conditional MLE-based autoregressive models, trajectory-centric GANs, transition-centric GANs, as well as our proposed technique. See also Figure 1.

| Type | Examples | Optimization Objective(s) | Generator Signal | Discrim. Signal | No Exposure Bias | Decoupled Discrim. | Non-Adversarial | Explicit Policy | Explicit Energy |
|---|---|---|---|---|---|---|---|---|---|
| Condit. MLE | T-Forcing [5] | Data LL | Stepwise | (N/A) | ✗ | (N/A) | ✓ | ✓ | ✗ |
| Condit. MLE | Z-Forcing [13] | Data LL (ELBO) | Stepwise | (N/A) | ✗ | (N/A) | ✓ | ✓ | ✗ |
| Condit. MLE | P-Forcing [10] | Data LL + Class. LL ($p_\theta$ v. $\tilde{p}_\theta$) | Stepwise | Global | ✗ | ✗ | ✗ | ✓ | ✗ |
| Traject. GAN | C-RNN-GAN [18] | Classification LL ($p_\theta$ v. $p_s$) | Global | Global | ✓ | ✗ | ✗ | ✗ | ✗ |
| Traject. GAN | DoppelGANger [19] | Classification LL ($p_\theta$ v. $p_s$) | Global | Global | ✓ | ✗ | ✗ | ✗ | ✗ |
| Traject. GAN | COT-GAN [20] | Sinkhorn Divergence ($p_\theta$ v. $p_s$) | Global | Global | ✓ | ✗ | ✗ | ✗ | ✗ |
| Transit. GAN | RC-GAN [21] | Classification LL ($\mu_\theta$ v. $\mu_s$) | Stepwise | Stepwise | ✓ | ✗ | ✗ | ✗ | ✗ |
| Transit. GAN | T-CGAN [22] | Classification LL ($\mu_\theta$ v. $\mu_s$) | Stepwise | Stepwise | ✓ | ✗ | ✗ | ✗ | ✗ |
| Transit. GAN | TimeGAN [12] | Class. LL ($\mu_\theta$ v. $\mu_s$) + Data LL | Stepwise | Stepwise | ✓ | ✗ | ✗ | ✗ | ✗ |
| **TimeGCI (Ours)** | | Discrim.: Class. LL ($p_\theta$ v. $p_s$) Generator: Policy Optimization | Stepwise | Global | ✓ | ✓ | ✓ | ✓ | ✓ |

**Corollary 6 (Generation as Imitation)** Let state space $\mathcal{Z} := \mathcal{H}$, action space $\mathcal{U} := \mathcal{X}$, and reward function $\rho := f_s$. In addition, let the dynamics be such that $\omega(\cdot|h_t, x_t)$ is the Dirac delta centered at $h_{t+1} := (x_1, ..., x_t)$. Then the regret exactly corresponds to the expected quality difference: $\mathcal{R}_s = \Delta \bar{F}_s$.

*Proof.* Immediate from Definition 1. □

Now, since we want low regret but have no knowledge of the true quality measure (i.e. "reward signal"), we may naturally learn it together. In this sense, TimeGCI is analogous to imitation by *inverse reinforcement learning* (IRL), which seeks to infer rewards that plausibly induced the demonstrated behavior, and to optimize imitating policies on that basis [63–66]. Further, in simultaneously optimizing for variety (cf. entropy) and typicality (cf. energy), TimeGCI is analogous to maximum-entropy IRL [67,68]. Our contrastive approach also bears mild resemblance to stepwise discriminators studied in this vein [54,69], although our framework focuses on trajectory-wise modeling, and is not adversarial (see Appendix D for more discussion on how TimeGCI relates to popular imitation learning methods).

There are also crucial differences: In imitation learning, dynamics are generally Markovian; states are readily defined as discrete elements or real vectors, and action spaces are small/discrete. The practical challenge is sample efficiency—to reduce the cost of environment interactions [70,71]. In time-series generation, however, rollouts are free—generating a synthetic trajectory does not require interacting with the real world. But dynamics are never Markovian: The practical challenge is that representations of variable-length histories must be jointly learned. Moreover, actions are the full-dimensional feature vectors themselves, which renders policy optimization more demanding than usual (see Appendix B); beyond the tractable tabular settings we experiment in, higher-dimensional data may prove challenging.

**Related Work** Table 1 summarizes the key differentiators of TimeGCI from prevailing techniques. As discussed in Section 1, MLE-based autoregressive models [5,10,13] are easy to optimize, and learn explicit conditional distributions that can be used for inspection, resampling, or uncertainty estimation, but they suffer from exposure bias [8,11,12]. GAN-based adversarial models fall into two camps: For trajectory-centric methods [18,19,72], with only sequence-level signals to guide the generator, they often struggles to converge to the adversarial objective without extensive tuning [12]—with the exception of [72], which utilizes Sinkhorn divergences instead. Transition-centric methods [12,21,22] provide more granular signals to guide the generator, but this simply alters the objective of learning $p_s$ to one of learning $\mu_s$, and still inherits the disadvantages of implicit, adversarial learning.

Our analysis is built on ideas from energy-based models (EBMs) [73–75] and reinforcement learning for sequence prediction [76–78]. In particular, our initial formulation (Section 3.1) can be viewed as a temporal extension of variational EBMs [41,42]. Moreover, by adaptively learning $\pi_\theta$ to give negative samples for $d_{\theta,\phi}$, the formulation we study (Section 3.2) is equivalent to a temporal analogue of noise-contrastive estimation (NCE) [55,79]. More tangentially, conditional EBMs have been trained with NCE for text generation [80–82], and the strength of global normalization has been studied [83]; that said, these are confined to the case where external input tokens are available for conditioning at each step—and not free-running as in our time-series setting. Finally, note that viewing sequence generation as a decision-making problem is present in language modeling [6,35] where task-specific metrics are available as signals. In the absence of predefined signals, GAN-based methods that jointly train discriminators to provide rewards for imitation have been studied [84–89], although they are adversarial, and all focus on the special case of generating discrete tokens for language modeling.

# 5 Experiments

**Benchmarks** We test Algorithm 1 (**TimeGCI**) against the following: The classic Teacher Forcing trains autoregressive networks using ground-truth conditioning (**T-Forcing**) [5]. Professor Forcing uses adversarial domain adaptation by training an auxiliary discriminator to encourage dynamics of the network's free-running and teacher-forced states to be similar (**P-Forcing**) [10]. Trajectory-centric recurrent GANs (**C-RNN-GAN**) directly plug RNNs into the GAN framework as generators and discriminators for full sequences [18]. Causal Optimal Transport GAN (**COT-GAN**) is the latest variant of this [20], proposing to approximate Sinkkorn divergences instead of the standard JS divergence. For transition-centric recurrent GANs (**RC-GAN**), the adversarial loss is computed as the sum of log likelihoods for the stepwise feature vectors conditioned on histories [21], instead of directly as the log likelihood for the entire sequence. Finally, Time-series GAN (**TimeGAN**) is its latest incarnation [12], proposing to generate and discriminate within a jointly optimized embedding space for efficiency.

**Datasets** We employ five tabular time-series datasets with a variety of different characteristics, such as periodicity, noise level, and correlations: First, we use a synthetic dataset of multivariate sinusoids with different frequencies and phases (**Sines**) [12]. Second, we use a UCI dataset from the monitored energy usage of household appliances in a low-energy house (**Energy**) [90].

Table 2: *Summary Statistics for Datasets Used.*

| *Dataset* | Dimension | Length | Autocor. | +3 Lag | +5 Lag |
|---|---|---|---|---|---|
| Sines | 5 | 24 | 0.875 | 0.623 | 0.377 |
| Metro | 9 | 24 | 0.429 | 0.200 | 0.029 |
| Gas | 20 | 24 | 0.656 | 0.382 | 0.170 |
| Energy | 29 | 24 | 0.702 | 0.411 | 0.176 |
| MIMIC-III | 52 | 24 | 0.532 | 0.212 | 0.059 |

Third, we use a UCI dataset from temperature-modulated semiconductor gas sensors for chemical detection (**Gas**) [91]. Fourth, we use a UCI dataset of hourly interstate vehicle volume at a state traffic recording station (**Metro**) [92]. Fifth, we use a medical dataset of intensive-care patients from the Medical Information Mart for Intensive Care (**MIMIC-III**) [93]. All datasets are accessible from their sources, and we use the original source code for preprocessing sines and the UCI datasets by [12], publicly available at [94]. Table 2 shows summary statistics for the datasets used in the experiments.

**Implementation** Experiments for each dataset are arranged as follows: The real trajectories that constitute the original dataset $\mathcal{D}$ are fed as input to train all algorithms. Each algorithm is subsequently used in test mode to generate 10,000 synthetic trajectories. Then, the performance of each algorithm is evaluated on the basis of these generated trajectories. This process is then performed for a total of 10 repetitions, from which we compile the means and standard errors for each reported result. For fair comparison, analogous network components across all benchmarks share the same recurrent architecture: Wherever a generator, policy, discriminator, energy, or critic network applies, we use LSTMs with one hidden layer of 32 units to compute hidden states for representing histories $h$, and two fully-connected hidden layers of 32 units each and ELU activations to compute task-specific output variables (i.e. the generator output, policy parameters, discriminator output, energy functions, or critic values). In other respects, we use the publicly available source code to construct the benchmark algorithms—accessible at [94–98]. See Appendix C for additional detail on hyperparameters and implementations.

**Evaluation and Results** In the tabular data setting, assessing synthetic data generation is inherently tricky [27, 99, 100]: Unlike in media-specific applications, we have no predefined measures such as music polyphony or BLEU scores, nor can we use human evaluation of realism as done for videos. For tabular time-series, the generally accepted standard for comparing synthetic data is to apply the *Train-on-Synthetic, Test-on-Real* (TSTR) framework, first proposed by [21] and employed by most recent work in synthetic time-series generation [12, 14, 21, 22, 26, 101], as well as more generally for tabular synthetic data of any kind [99, 100, 102]. Specifically, we apply the performance measure used by [12, 14, 101] to quantify how much the synthetic sequences inherit the predictive characteristics of the original dataset (**Predictive Score**): Using synthetic samples, a generic post-hoc sequence-prediction model is learned to forecast next-step feature vectors over training sequences. Then, the trained model is evaluated on the original data, and its predictive performance is quantified in terms of the mean absolute error. We use the original source code for computing this metric, publicly available at [94].

Further to prior works using this measure, we additionally believe that synthetic data evaluation should be more general than just next-step TSTR forecasting. After all, the distinguishing characteristic of sequential (vs. static) data generation is that we care about evolution of features *over time*. Hence we also compute TSTR metrics for horizons of other lengths (**+3 Steps Ahead** and **+5 Steps Ahead**). Importantly, note that a key strength of TSTR evaluation is in its sensitivity to *mode collapse*: If any generation scheme suffers from mode collapse (as GAN methods are prone to), TSTR scores would degrade due to the synthetic data failing to capture the diversity of the real data, which means any

Table 3: *Performance Comparison of TimeGCI and Benchmarks.* Bold numbers indicate best-performing results.

| Benchmark | Metric | Sines | Energy | Gas | Metro | MIMIC-III |
|---|---|---|---|---|---|---|
| T-Forcing | Predictive Score | $0.108 \pm 0.002$ | $0.310 \pm 0.001$ | $0.035 \pm 0.003$ | $0.242 \pm 0.001$ | $0.017 \pm 0.001$ |
| | +3 Steps Ahead | $0.115 \pm 0.001$ | $0.281 \pm 0.001$ | $0.080 \pm 0.001$ | $0.244 \pm 0.001$ | $0.024 \pm 0.007$ |
| | +5 Steps Ahead | $0.122 \pm 0.003$ | $0.270 \pm 0.002$ | $0.111 \pm 0.001$ | $0.248 \pm 0.001$ | $0.018 \pm 0.003$ |
| | $x$-Corr. Score | $8.369 \pm 0.015$ | $194.1 \pm 0.043$ | $150.8 \pm 0.067$ | $4.222 \pm 0.013$ | $400.9 \pm 3.203$ |
| P-Forcing | Predictive Score | $0.105 \pm 0.001$ | $0.303 \pm 0.002$ | $0.037 \pm 0.001$ | $0.241 \pm 0.001$ | $0.023 \pm 0.006$ |
| | +3 Steps Ahead | $0.110 \pm 0.001$ | $0.268 \pm 0.002$ | $0.086 \pm 0.002$ | $0.241 \pm 0.001$ | $0.018 \pm 0.001$ |
| | +5 Steps Ahead | $0.115 \pm 0.001$ | $0.259 \pm 0.002$ | $0.121 \pm 0.002$ | $0.242 \pm 0.001$ | $0.017 \pm 0.001$ |
| | $x$-Corr. Score | $8.156 \pm 0.010$ | $207.6 \pm 0.057$ | $150.5 \pm 0.023$ | $3.014 \pm 0.006$ | $346.6 \pm 2.901$ |
| C-RNN-GAN | Predictive Score | $0.751 \pm 0.001$ | $0.500 \pm 0.001$ | $0.242 \pm 0.001$ | $0.419 \pm 0.005$ | $0.019 \pm 0.001$ |
| | +3 Steps Ahead | $0.769 \pm 0.001$ | $0.500 \pm 0.001$ | $0.243 \pm 0.001$ | $0.416 \pm 0.002$ | $0.020 \pm 0.001$ |
| | +5 Steps Ahead | $0.786 \pm 0.001$ | $0.501 \pm 0.001$ | $0.241 \pm 0.001$ | $0.416 \pm 0.003$ | $0.019 \pm 0.001$ |
| | $x$-Corr. Score | $10.76 \pm 0.012$ | $644.2 \pm 0.112$ | $266.4 \pm 0.008$ | $18.39 \pm 0.003$ | $1720. \pm 0.339$ |
| COT-GAN | Predictive Score | $0.099 \pm 0.001$ | $0.259 \pm 0.001$ | $0.022 \pm 0.001$ | $0.245 \pm 0.001$ | $0.014 \pm 0.001$ |
| | +3 Steps Ahead | $0.109 \pm 0.001$ | $0.261 \pm 0.001$ | $0.050 \pm 0.001$ | $0.246 \pm 0.001$ | $0.013 \pm 0.001$ |
| | +5 Steps Ahead | $0.110 \pm 0.001$ | $0.262 \pm 0.001$ | $0.072 \pm 0.001$ | $0.245 \pm 0.001$ | $0.013 \pm 0.001$ |
| | $x$-Corr. Score | $3.114 \pm 0.038$ | $\mathbf{67.93 \pm 0.227}$ | $\mathbf{25.56 \pm 0.156}$ | $3.055 \pm 0.013$ | $497.7 \pm 2.581$ |
| RC-GAN | Predictive Score | $0.751 \pm 0.001$ | $0.498 \pm 0.001$ | $0.243 \pm 0.001$ | $0.412 \pm 0.003$ | $0.019 \pm 0.001$ |
| | +3 Steps Ahead | $0.770 \pm 0.001$ | $0.500 \pm 0.001$ | $0.244 \pm 0.001$ | $0.415 \pm 0.004$ | $0.019 \pm 0.001$ |
| | +5 Steps Ahead | $0.786 \pm 0.001$ | $0.499 \pm 0.001$ | $0.243 \pm 0.001$ | $0.418 \pm 0.004$ | $0.018 \pm 0.001$ |
| | $x$-Corr. Score | $5.649 \pm 0.012$ | $582.3 \pm 0.047$ | $231.2 \pm 0.003$ | $19.77 \pm 0.001$ | $1592. \pm 0.192$ |
| TimeGAN | Predictive Score | $0.196 \pm 0.006$ | $0.261 \pm 0.001$ | $0.264 \pm 0.011$ | $0.245 \pm 0.002$ | $0.502 \pm 0.023$ |
| | +3 Steps Ahead | $0.223 \pm 0.006$ | $0.263 \pm 0.001$ | $0.251 \pm 0.014$ | $0.243 \pm 0.001$ | $0.484 \pm 0.021$ |
| | +5 Steps Ahead | $0.246 \pm 0.005$ | $0.262 \pm 0.005$ | $0.252 \pm 0.012$ | $0.242 \pm 0.001$ | $0.453 \pm 0.020$ |
| | $x$-Corr. Score | $17.86 \pm 0.001$ | $667.5 \pm 0.001$ | $282.5 \pm 0.001$ | $17.11 \pm 0.001$ | $2140. \pm 0.010$ |
| **TimeGCI (Ours)** | Predictive Score | $\mathbf{0.097 \pm 0.001}$ | $\mathbf{0.251 \pm 0.001}$ | $\mathbf{0.018 \pm 0.000}$ | $\mathbf{0.239 \pm 0.001}$ | $\mathbf{0.002 \pm 0.000}$ |
| | +3 Steps Ahead | $\mathbf{0.104 \pm 0.001}$ | $\mathbf{0.251 \pm 0.001}$ | $\mathbf{0.042 \pm 0.001}$ | $\mathbf{0.239 \pm 0.001}$ | $\mathbf{0.001 \pm 0.000}$ |
| | +5 Steps Ahead | $\mathbf{0.109 \pm 0.001}$ | $\mathbf{0.251 \pm 0.001}$ | $\mathbf{0.067 \pm 0.001}$ | $\mathbf{0.239 \pm 0.001}$ | $\mathbf{0.001 \pm 0.000}$ |
| | $x$-Corr. Score | $\mathbf{1.195 \pm 0.011}$ | $105.2 \pm 0.433$ | $47.91 \pm 0.811$ | $\mathbf{0.738 \pm 0.019}$ | $\mathbf{194.3 \pm 0.180}$ |

prediction model trained on that basis would also fail to capture this variation). Finally, similar to some recent works [19, 20], we also compute the cross-correlations of real and synthetic feature vectors, and report the sum of the absolute differences between them, averaged over time ($x$-**Corr. Score**); this serves to verify if feature relationships are preserved well, in addition to temporal relationships. Table 3 shows the results: With respect to these metrics, we find that TimeGCI somewhat consistently produces synthetic samples that perform similarly or better than benchmark algorithms in all datasets. (Note that we do empirically observe several instances of mode collapse in GAN-based benchmarks).

## 6    Conclusion

In this work, we invite an explicit analogy between time-series generation and imitation learning, and explore a framework that fleshes out this connection. Two caveats are in order: First, while we began from the notion of moment-matching to address the error compounding problem, in practice there is no guarantee that this is accomplished well during optimization. In particular, *scalability* is a major limitation beyond the range of feature dimensions and sequence lengths considered in our experiments. Sample-based estimates could rapidly degrade with the horizon, especially if transitions are highly stochastic. A relevant question is whether or not training on fixed subsequence lengths could potentially alleviate this concern for longer sequences. In addition, while our approach seeks to dispense with the instabilities typical of adversarial training, we are instead left with the difficulties of policy optimization, which may prove a prohibitive challenge in higher-dimensional feature spaces. For the datasets we consider, we find that pre-training and regularizing the policy with maximum likelihood, combined with a small enough learning rate, had the most impact in promoting stability and learning. Second, we reiterate that a perennial challenge in modeling tabular data is in choosing the metric for *evaluation*. While we opted for the most commonly accepted method of TSTR, this may not be general enough to capture the range of downstream tasks that may be performed on the synthetic data. Future work will benefit from a deeper investigation into more sophisticated measures for time series, such as contrastive methods and how to evaluate different aspects of the "quality" of the generated trajectories.

## Acknowledgments

We would like to thank the reviewers for all their invaluable feedback. This work was supported by Alzheimer's Research UK, The Alan Turing Institute under the EPSRC grant EP/N510129/1, the US Office of Naval Research, as well as the National Science Foundation under grant number 1722516.

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
