# A  Proofs of Propositions

For Lemmas 1 and 2, we first introduce some additional quantities to enable more compact notation; in particular, we adopt the value-function terminology from imitation learning. The following definitions are standard and immediate from the mapping given by Corollary 6, but are explicitly stated here for completeness. Recall that $f_s : \mathcal{H} \times \mathcal{X} \to [-c, c]$ for some finite $c$. At any state $h_t$, define the "value function" to be the (forward-looking) expected sum of future quantities $f_s(h_u, x_u)$ for $u = t, ..., T$. Specifically, let $V_{s,t}^{\pi_\theta}(h) : \mathcal{H} \to [-cT, cT]$ and $Q_{s,t}^{\pi_\theta}(h, x) : \mathcal{H} \times \mathcal{X} \to [-cT, cT]$ be given as follows:

$$V_{s,t}^{\pi_\theta}(h) := \mathbb{E}_{\tau \sim p_\theta}[\Sigma_{u=t}^T f_s(h_u, x_u)|h_t = h] \tag{13}$$

$$Q_{s,t}^{\pi_\theta}(h, x) := \mathbb{E}_{\tau \sim p_\theta}[\Sigma_{u=t}^T f_s(h_u, x_u)|h_t = h, x_t = x] \tag{14}$$

where the notation for both $V_{s,t}^{\pi_\theta}$ and $Q_{s,t}^{\pi_\theta}$ is explicit as to their dependence on the policy $\pi_\theta$ being followed, the source $s$ under consideration, and the time $t$—unlike in typical imitation learning, we operate in a non-stationary (and non-Markovian) setting. For Lemma 1, we require an additional result:

**Lemma 7 (Expected Quality Difference)** $\Delta \bar{F}_s(\theta) = T\mathbb{E}_{\substack{h \sim \mu_s \\ x \sim \pi_s(\cdot|h)}} Q_{s,t}^{\pi_\theta}(h, x) - T\mathbb{E}_{\substack{h \sim \mu_s \\ x \sim \pi_\theta(\cdot|h)}} Q_{s,t}^{\pi_\theta}(h, x)$.

*Proof.* From Definition 2,

$$\Delta \bar{F}_s(\theta) = \mathbb{E}_{\tau \sim p_s} \sum_t f_s(h_t, x_t) - \mathbb{E}_{\tau \sim p_\theta} \sum_t f_s(h_t, x_t) \tag{15}$$

$$= \mathbb{E}_{\tau \sim p_s} \sum_t (f_s(h_t, x_t) + V_{s,t}^{\pi_\theta}(h_t) - V_{s,t}^{\pi_\theta}(h_t)) - \mathbb{E}_{\tau \sim p_\theta} \sum_t f_s(h_t, x_t) \tag{16}$$

$$= \mathbb{E}_{\tau \sim p_s} \sum_t (f_s(h_t, x_t) + V_{s,t+1}^{\pi_\theta}(h_{t+1}) - V_{s,t}^{\pi_\theta}(h_t)) \tag{17}$$

$$= \mathbb{E}_{\tau \sim p_s} \sum_t (Q_{s,t}^{\pi_\theta}(h_t, x_t) - V_{s,t}^{\pi_\theta}(h_t)) \tag{18}$$

$$= T\mathbb{E}_{h \sim \mu_s, x \sim \pi_s(\cdot|h)}(Q_{s,t}^{\pi_\theta}(h, x) - V_{s,t}^{\pi_\theta}(h)) \tag{19}$$

$$= T\mathbb{E}_{h \sim \mu_s, x \sim \pi_s(\cdot|h)} Q_{s,t}^{\pi_\theta}(h, x) - T\mathbb{E}_{h \sim \mu_s, x \sim \pi_\theta(\cdot|h)} Q_{s,t}^{\pi_\theta}(h, x) \tag{20}$$

where (16) to (17) telescopes terms, and we use the fact $V_{s,T+1}^\pi(h) = 0$. This derivation can be viewed as a non-stationary, non-Markovian analogue of the "performance difference" result in [103]. $\square$

**Lemma 1** Let $\max_{f \in \mathbb{R}^{\mathcal{H} \times \mathcal{X}}} \left( \mathbb{E}_{\substack{h \sim \mu_s \\ x \sim \pi_s(\cdot|h)}} f(h, x) - \mathbb{E}_{\substack{h \sim \mu_s \\ x \sim \pi_\theta(\cdot|h)}} f(h, x) \right) \le \epsilon$. Then $\Delta \bar{F}_s(\theta) \in O(T^2 \epsilon)$.

*Proof.* From Lemma 7,

$$\Delta \bar{F}_s(\theta) = T\mathbb{E}_{h \sim \mu_s, x \sim \pi_s(\cdot|h)} Q_{s,t}^{\pi_\theta}(h, x) - T\mathbb{E}_{h \sim \mu_s, x \sim \pi_\theta(\cdot|h)} Q_{s,t}^{\pi_\theta}(h) \tag{21}$$

$$= T\mathbb{E}_{h \sim \mu_s}[\mathbb{E}_{x \sim \pi_s(\cdot|h)} Q_{s,t}^{\pi_\theta}(h, x) - \mathbb{E}_{x \sim \pi_\theta(\cdot|h)} Q_{s,t}^{\pi_\theta}(h, x)] \tag{22}$$

$$\le \max_{Q \in [-cT, cT]^{\mathcal{H} \times \mathcal{X}}} T\mathbb{E}_{h \sim \mu_s}[\mathbb{E}_{x \sim \pi_s(\cdot|h)} Q(h, x) - \mathbb{E}_{x \sim \pi_\theta(\cdot|h)} Q(h, x)] \tag{23}$$

$$\le \max_{f \in \mathbb{R}^{\mathcal{H} \times \mathcal{X}}} T\mathbb{E}_{h \sim \mu_s}[\mathbb{E}_{x \sim \pi_s(\cdot|h)} Tf(h, x) - \mathbb{E}_{x \sim \pi_\theta(\cdot|h)} Tf(h, x)] \tag{24}$$

$$\le T^2 \epsilon \tag{25}$$

where the final inequality applies the assumption from the lemma. Note that this is similar in spirit to various results for error accumulation in imitation learning through behavioral cloning. The most well-known one is [61], where a quadratic bound is given with respect to the *probability* that the learned policy makes a small mistake. Another well-known one is in [104], where the bound is given with respect to *sample complexity*. Here, in order to motivate our perspective from the notion of expected quality difference, our bound is given with respect to the *moment-matching* discrepancy, and can be interpreted as a non-Markovian variant of the "off-policy upper bound" result in [105]. $\square$

**Lemma 2** Let $\max_{f \in \mathbb{R}^{\mathcal{H} \times \mathcal{X}}} \left( \mathbb{E}_{\substack{h \sim \mu_s \\ x \sim \pi_s(\cdot|h)}} f(h, x) - \mathbb{E}_{\substack{h \sim \mu_\theta \\ x \sim \pi_\theta(\cdot|h)}} f(h, x) \right) \le \epsilon$. Then $\Delta \bar{F}_s(\theta) \in O(T \epsilon)$.

*Proof.* From Definition 1,

$$\Delta \bar{F}_s(\theta) = \mathbb{E}_{\tau \sim p_s} \sum_t f_s(h_t, x_t) - \mathbb{E}_{\tau \sim p_\theta} \sum_t f_s(h_t, x_t) \tag{26}$$

$$= T\mathbb{E}_{h \sim \mu_s, \pi_s(\cdot|h)} f_s(h, x) - T\mathbb{E}_{h \sim \mu_\theta, \pi_\theta(\cdot|h)} f_s(h, x) \tag{27}$$

$$\le \max_{f \in [-c, c]^{\mathcal{H} \times \mathcal{X}}} (T\mathbb{E}_{h \sim \mu_s, x \sim \pi_s(\cdot|h)} f(h, x) - T\mathbb{E}_{h \sim \mu_\theta, x \sim \pi_\theta(\cdot|h)} f(h, x)) \tag{28}$$

$$\le \max_{f \in \mathbb{R}^{\mathcal{H} \times \mathcal{X}}} (T\mathbb{E}_{h \sim \mu_s, x \sim \pi_s(\cdot|h)} f(h, x) - T\mathbb{E}_{h \sim \mu_\theta, x \sim \pi_\theta(\cdot|h)} f(h, x)) \tag{29}$$

$$\le T\epsilon \tag{30}$$

where the final inequality applies the assumption from the lemma. Note that this is similar in spirit to various results for error accumulation in imitation learning through distribution matching. For instance, [106] shows a bound in terms of *divergences* in occupancy measures, while [104] shows a bound in terms of *sample complexity*. Here, in order to motivate our perspective from the notion of expected quality difference, our bound is given with respect to the *moment-matching* discrepancy, and can similarly be interpreted as a non-Markovian variant of the "reward upper bound" in [105]. $\square$

For Propositions 3 and 4, we use the fact that training the structured classifier (Definition 2) using the energy loss (Equation 11) amounts to a specific form of (sequence-wise) noise-contrastive estimation, and where the "noise" $p_\theta$ employed happens to be adaptively trained via the policy loss (Equation 12):

**Proposition 3 (Global Optimality)** Let $f_\phi \in \mathbb{R}^{\mathcal{H} \times \mathcal{X}}$, and let $p_\theta \in \Delta(\mathcal{T})$ be any distribution satisfying positivity: $p_s(\tau) > 0 \Rightarrow p_\theta(\tau) > 0$ (this does not require $\pi_\theta$ be optimal for $f_\phi$). Then $\mathcal{L}_{\text{energy}}(\phi; \theta)$ is globally minimized at $F_\phi(\cdot) - \log Z_\phi = \log p_s(\cdot)$, whence $p_\phi$ is self-normalized with unit integral.

*Proof.* Briefly, a noise-contrastive estimator [79] operates as follows: Suppose we have some data $y \in \mathcal{Y}$ distributed as $p_{\text{data}}(y)$. Consider that we wish to learn a model distribution $p_{\text{model}}$, as follows:

$$p_{\text{model}}(y; a, b) := \tilde{p}_{\text{model}}(y; a) \exp(b) \tag{31}$$

parameterized by $a$ and $b$, where we emphasize that the model is not necessarily normalized as $b$ is simply a learnable parameter. Also denote any noise distribution that can be sampled and evaluated:

$$p_{\text{noise}}(y; c) \tag{32}$$

parameterized by $c$. Now, define a classifier $d(\cdot; a, b, c)$ as follows, which we shall train to discriminate between $p_{\text{data}}$ and $p_{\text{noise}}$—that is, given some $y$, to represent the (posterior) probability that it is real:

$$d(y; a, b, c) := \sigma(\log p_{\text{model}}(y; a, b) - \log p_{\text{noise}}(y; c)) \tag{33}$$

where $\sigma$ indicates the usual sigmoid function, i.e. $\sigma(u) := 1/(1 + \exp(-u))$ for any $u \in \mathbb{R}$. The noise contrastive estimator maximizes the likelihood of the parameters $a, b$ in $d$ given $p_{\text{data}}$ and $p_{\text{noise}}$:

$$\mathcal{L}_{\text{class}}(a, b; c) := -\mathbb{E}_{y \sim p_{\text{data}}} \log d(y; a, b, c) - \mathbb{E}_{y \sim p_{\text{noise}}} \log(1 - d(y; a, b, c)) \tag{34}$$

In this optimization problem, a basic result is that $\mathcal{L}_{\text{class}}$ attains a minimum at $\log p_{\text{model}} = \log p_{\text{data}}$ and that there are no other minima if $p_{\text{noise}}$ is chosen such that $p_{\text{data}}(y) > 0 \Rightarrow p_{\text{noise}}(y) > 0$ holds: see the "nonparametric estimation" result in [52]. Now, let us consider the following correspondence:

$$\left(\mathcal{Y}, p_{\text{data}}, \tilde{p}_{\text{model}}(\cdot; a), b, p_{\text{noise}}(\cdot; c)\right) := \left(\mathcal{T}, p_s, \tilde{p}_\phi, -\log Z_\phi, p_\theta\right) \tag{35}$$

In other words, let the underlying space be that of trajectories $\mathcal{Y} := \mathcal{T}$; let the data distribution be $p_{\text{data}} := p_s$; let the model distribution be given by the un-normalized energy model $\tilde{p}_{\text{model}}(\cdot; a) := \tilde{p}_\phi$ and partition function $b = -\log Z_\phi$; and let the noise distribution be given by rollouts of the policy, $p_{\text{noise}}(\cdot; c) := p_\theta$. Then it is easy to see that the classifier and its loss function correspond as follows:

$$\begin{aligned} d_{\theta, \phi}(\cdot) &= d(\cdot; a, b, c) \\ \mathcal{L}_{\text{energy}}(\phi; \theta) &= \mathcal{L}_{\text{class}}(a, b; c) \end{aligned} \tag{36}$$

But then the optimality result above directly maps to the statement that $\mathcal{L}_{\text{energy}}$ is globally minimized at $F_\phi(\cdot) - \log Z_\phi = \log p_s(\cdot)$ assuming that the positivity condition $p_s(\tau) > 0 \Rightarrow p_\theta(\tau) > 0$ holds. Technicality: Note that here $\tilde{p}_\phi$ is constrained as $F_\phi(\tau) := \sum_t f_\phi(h_t, x_t)$ instead of being arbitrarily parameterizable, but this does not affect realizability as we assumed that $F_s(\tau) := \sum_t f_s(h_t, x_t)$. $\square$

**Proposition 4 (Asymptotic Consistency)** Let $\phi^*$ denote the minimizer for $\mathcal{L}_{\text{energy}}(\phi; \theta)$, and let $\hat{\phi}^*_M$ denote the minimizer for its finite-data approximation—that is, where the expectations over $p_s$ and $p_\theta$ are approximated by $M$ samples. Then under some mild conditions, as $M$ increases $\hat{\phi}^*_M \xrightarrow{p} \phi^*$.

*Proof.* Continuing the exposition above, let $\mathcal{L}^M_{\text{class}}(a, b; c)$ indicate the finite-data approximation of $\mathcal{L}_{\text{class}}(a, b; c)$—that is, by using $M$ samples to approximate the true expectations over $p_{\text{data}}$ and $p_{\text{noise}}$:

$$\mathcal{L}^M_{\text{class}}(a, b; c) := -\frac{1}{M} \sum_{m=1}^{M} \log d(y^{(m)}_{\text{data}}; a, b, c) - \frac{1}{M} \sum_{m=1}^{M} \log(1 - d(y^{(m)}_{\text{noise}}; a, b, c)) \tag{37}$$

where the samples are drawn as $y^{(m)}_{\text{data}} \sim p_{\text{data}}$ and $y^{(m)}_{\text{noise}} \sim p_{\text{noise}}$. Consider the following conditions:

1. Positivity: $p_{\text{data}}(y) > 0 \Rightarrow p_{\text{noise}}(y) > 0$;
2. Uniform convergence: $\sup_{a,b} |\mathcal{L}_{\text{class}}^M(a, b; c) - \mathcal{L}_{\text{class}}(a, b; c)| \xrightarrow{p} 0$; and
3. The following matrix is full-rank: $\mathcal{I} := \int g(y)g(y)^\top p_{\text{data}}(y)p_{\text{noise}}(y)/(p_{\text{data}}(y)+p_{\text{noise}}(y))dy$, where $g(y) := \nabla_{(a,b)} \log p_{\text{model}}(y; a, b)|_{(a^*,b^*)}$ and $a^*, b^*$ denote the optimal values of the model.

Note that (1) is same as before, and (2) and (3) are analogous to standard assumptions in maximum likelihood estimation. Let $\hat{a}_M^*, \hat{b}_M^*$ denote the minimizers for $\mathcal{L}_{\text{class}}^M(a, b; c)$. Under the preceding conditions, another basic result is that $(\hat{a}_M^*, \hat{b}_M^*)$ converges in probability to $(a^*, b^*)$ as $M$ grows: see the "consistency" result in [52]. But continuing the correspondence from before, it is easy to see that

$$\mathcal{L}_{\text{energy}}^M(\phi; \theta) = \mathcal{L}_{\text{class}}^M(a, b; c) \tag{38}$$

where we similarly define $\mathcal{L}_{\text{energy}}^M(\phi; \theta)$ to be the finite-data approximation of $\mathcal{L}_{\text{energy}}(\phi; \theta)$—that is, by using $M$ samples to approximate the true expectations over $p_s$ and $p_\theta$, and $y_s^{(m)} \sim p_s$ and $y_\theta^{(m)} \sim p_\theta$:

$$\mathcal{L}_{\text{energy}}^M(\phi; \theta) := -\frac{1}{M}\sum_{m=1}^M \log d_{\theta,\phi}(\tau_s^{(m)}) - \frac{1}{M}\sum_{m=1}^M \log\left(1 - d_{\theta,\phi}(\tau_\theta^{(m)})\right) \tag{39}$$

which directly maps the above convergence result to the statement that as $M$ increases $\hat{\phi}_M^* \xrightarrow{p} \phi^*$. $\square$

**Proposition 5 (Gradient Equality)** Let $\phi_k$ be the value taken by $\phi$ after the $k$-th gradient update, and let $\theta_k^*$ denote the associated minimizer for $\mathcal{L}_{\text{policy}}(\theta; \phi_k)$. Suppose $p_\phi$ is already normalized; then

$$\nabla_\phi \mathcal{L}_{\text{energy}}(\phi; \theta_k^*) = -\frac{T}{2}\nabla_\phi \mathcal{L}(\theta_k^*, \phi)$$

That is, at $\theta_k^*$ the energy gradient (of Equation 11) recovers the original gradient (from Equation 7). In the general case, suppose $p_\phi$ is un-normalized, such that $p_{\theta_k^*} = p_\phi/K_\phi$ for some constant $K_\phi$; then

$$\nabla_\phi \mathcal{L}_{\text{energy}}(\phi; \theta_k^*) = \frac{TK_\phi}{K_\phi+1}\mathbb{E}_{\substack{h\sim\mu_{\theta_k^*} \\ x\sim\pi_{\theta_k^*}(\cdot|h)}} \nabla_\phi f_\phi(h, x) - \frac{T}{K_\phi+1}\mathbb{E}_{\substack{h\sim\mu_s \\ x\sim\pi_s(\cdot|h)}} \nabla_\phi f_\phi(h, x)$$

*Proof.* From Equation 11,

$$\nabla_\phi \mathcal{L}_{\text{energy}}(\phi; \theta_k^*) = \nabla_\phi\left(-\mathbb{E}_{\tau\sim p_s}\log d_{\theta_k^*,\phi}(\tau) - \mathbb{E}_{\tau\sim p_{\theta_k^*}}\log\left(1 - d_{\theta_k^*,\phi}(\tau)\right)\right) \tag{40}$$

$$= \nabla_\phi\left(-\mathbb{E}_{\tau\sim p_s}\log\frac{p_\phi(\tau)}{p_\phi(\tau)+p_{\theta_k^*}(\tau)} - \mathbb{E}_{\tau\sim p_{\theta_k^*}}\log\frac{p_{\theta_k^*}(\tau)}{p_\phi(\tau)+p_{\theta_k^*}(\tau)}\right) \tag{41}$$

$$= -\mathbb{E}_{\tau\sim p_s}\nabla_\phi(\log p_\phi(\tau) - \log(p_\phi(\tau)+p_{\theta_k^*}(\tau))) + \mathbb{E}_{\tau\sim p_{\theta_k^*}}\nabla_\phi\log(p_\phi(\tau)+p_{\theta_k^*}(\tau)) \tag{42}$$

$$= -\mathbb{E}_{\tau\sim p_s}\left[\nabla_\phi\log p_\phi(\tau) - \frac{\nabla_\phi p_\phi(\tau)}{p_\phi(\tau)+p_{\theta_k^*}(\tau)}\right] + \mathbb{E}_{\tau\sim p_{\theta_k^*}}\frac{\nabla_\phi p_\phi(\tau)}{p_\phi(\tau)+p_{\theta_k^*}(\tau)} \tag{43}$$

$$= -\mathbb{E}_{\tau\sim p_s}\left[\nabla_\phi\log p_\phi(\tau) - \frac{p_\phi(\tau)\nabla_\phi\log p_\phi(\tau)}{p_\phi(\tau)+p_{\theta_k^*}(\tau)}\right] + \mathbb{E}_{\tau\sim p_{\theta_k^*}}\frac{p_\phi(\tau)\nabla_\phi\log p_\phi(\tau)}{p_\phi(\tau)+p_{\theta_k^*}(\tau)} \tag{44}$$

$$= -\mathbb{E}_{\tau\sim p_s}\left[\nabla_\phi\log p_\phi(\tau) - \frac{1}{2}\nabla_\phi\log p_\phi(\tau)\right] + \frac{1}{2}\mathbb{E}_{\tau\sim p_{\theta_k^*}}\nabla_\phi\log p_\phi(\tau) \tag{45}$$

$$= \frac{1}{2}\mathbb{E}_{\tau\sim p_{\theta_k^*}}\nabla_\phi\log p_\phi(\tau) - \frac{1}{2}\mathbb{E}_{\tau\sim p_s}\nabla_\phi\log p_\phi(\tau) \tag{46}$$

$$= \frac{1}{2}\mathbb{E}_{\tau\sim p_{\theta_k^*}}\nabla_\phi(\log \tilde{p}_\phi(\tau) - \log Z_\phi) - \frac{1}{2}\mathbb{E}_{\tau\sim p_s}\nabla_\phi(\log \tilde{p}_\phi(\tau) - \log Z_\phi) \tag{47}$$

$$= \frac{1}{2}\left(\mathbb{E}_{\tau\sim p_{\theta_k^*}}\nabla_\phi\sum_t f_\phi(h_t, x_t) - \mathbb{E}_{\tau\sim p_s}\nabla_\phi\sum_t f_\phi(h_t, x_t)\right) \tag{48}$$

$$= \frac{1}{2}\left(T\mathbb{E}_{h\sim\mu_{\theta_k^*},x\sim\pi_{\theta_k^*}(\cdot|h)}\nabla_\phi f_\phi(h, x) - T\mathbb{E}_{h\sim\mu_s,x\sim\pi_s(\cdot|h)}\nabla_\phi f_\phi(h, x)\right) \tag{49}$$

$$= -\frac{T}{2}\nabla_\phi \mathcal{L}(\theta_k^*, \phi) \tag{50}$$

where the fourth and fifth lines repeatedly use the identity $\nabla_z p_z \equiv p_z \nabla_z \log p_z$ for any $p_z$ parameterized by $z$, and the sixth line uses the fact that the current value of $\theta$ (i.e. $\theta_k^*$) is the minimizer for $\mathcal{L}_{\text{policy}}(\theta; \phi)$ at the current value of $\phi$ (i.e. $\phi_k$), hence it must be the case that $p_\theta = p_\phi$ at those values. Note that this assumes that $p_\phi$ is normalized; in practice this will be approximately true, for instance if we pre-train $\phi$ beforehand, using a fixed $\pi_\theta$ pre-trained by maximum likelihood (see Appendix B).

In the more general case of any arbitrary un-normalized $p_\phi$, we only know $p_{\theta_k^*} = \frac{1}{K_\phi} p_\phi$ for some constant $K_\phi$; then we recover a generalized "weighted" version of Equation 7. From the fifth line above,

$$\nabla_\phi \mathcal{L}_{\text{energy}}(\phi; \theta_k^*) = -\mathbb{E}_{\tau \sim p_s}\left[\nabla_\phi \log p_\phi(\tau) - \frac{p_\phi(\tau)\nabla_\phi \log p_\phi(\tau)}{p_\phi(\tau) + p_{\theta_k^*}(\tau)}\right] + \mathbb{E}_{\tau \sim p_{\theta_k^*}}\frac{p_\phi(\tau)\nabla_\phi \log p_\phi(\tau)}{p_\phi(\tau) + p_{\theta_k^*}(\tau)} \quad (51)$$

$$= -\mathbb{E}_{\tau \sim p_s}\left[\nabla_\phi \log p_\phi(\tau) - \frac{K_\phi}{K_\phi+1}\nabla_\phi \log p_\phi(\tau)\right] + \frac{K_\phi}{K_\phi+1}\mathbb{E}_{\tau \sim p_{\theta_k^*}}\nabla_\phi \log p_\phi(\tau) \quad (52)$$

$$= \frac{K_\phi}{K_\phi+1}\mathbb{E}_{\tau \sim p_{\theta_k^*}}\nabla_\phi \log p_\phi(\tau) - \frac{1}{K_\phi+1}\mathbb{E}_{\tau \sim p_s}\nabla_\phi \log p_\phi(\tau) \quad (53)$$

$$= \frac{K_\phi}{K_\phi+1}\mathbb{E}_{\tau \sim p_{\theta_k^*}}\nabla_\phi(\log \tilde{p}_\phi(\tau) - \log Z_\phi) - \frac{1}{K_\phi+1}\mathbb{E}_{\tau \sim p_s}\nabla_\phi(\log \tilde{p}_\phi(\tau) - \log Z_\phi) \quad (54)$$

$$= \frac{K_\phi}{K_\phi+1}\mathbb{E}_{\tau \sim p_{\theta_k^*}}\nabla_\phi \sum_t f_\phi(h_t, x_t) - \frac{1}{K_\phi+1}\mathbb{E}_{\tau \sim p_s}\nabla_\phi \sum_t f_\phi(h_t, x_t) \quad (55)$$

$$= \frac{TK_\phi}{K_\phi+1}\mathbb{E}_{h \sim \mu_{\theta_k^*}, x \sim \pi_{\theta_k^*}(\cdot|h)}\nabla_\phi f_\phi(h, x) - \frac{T}{K_\phi+1}\mathbb{E}_{h \sim \mu_s, x \sim \pi_s(\cdot|h)}\nabla_\phi f_\phi(h, x) \quad (56)$$

This "weighting" is intuitive: If $p_\phi$ is un-normalized such that $K_\phi > 1$, the energy loss automatically places higher weights on negative samples $h \sim \mu_{\theta_k^*}, x \sim \pi_{\theta_k^*}(\cdot|h)$ to bring it down; conversely, if $p_\phi$ is un-normalized such that $K_\phi < 1$, the energy loss places higher weights on positive samples $h \sim \mu_s, x \sim \pi_s(\cdot|h)$ to bring it up. (If $p_\phi$ is normalized, then $K_\phi = 1$ and the weights are equal). $\square$

**Note on Duality** Lemmas 1 and 2 are for building intuition, and we are *not* implying that Equations 3 and 5 are direct generalizations of the duality between maximum likelihood and maximum entropy. To be clear, we shall explain Equation 5; Equation 3 is similar. Consider first the case of finite $\mathcal{X}$ (therefore finite $\mathcal{T}$). In the usual linear case, given basis functions $T(\tau)$, the optimization problem is:

$$\arg\min_\theta E_{\tau \sim p_\theta} \log p_\theta(\tau) \quad \text{s.t.} \quad E_{\tau \sim p_s}T(\tau) = E_{\tau \sim p_\theta}T(\tau) \quad (57)$$

Internalizing the constraint, we may write $\arg\min_\theta(E_{\tau \sim p_\theta} \log p_\theta(\tau) + \max_F(\langle F, E_{\tau \sim p_s}T(\tau)\rangle - \langle F, E_{\tau \sim p_\theta}T(\tau)\rangle))$. To generalize to the nonlinear case, let us now specifically define the feature vector $T(\tau)$ to be the indicator function (i.e. a finite-length vector, each zero-one entry of which corresponds to each element in $\mathcal{T}$). Then note that $\langle F, E_{\tau \sim p}T(\tau)\rangle = \langle F, p\rangle = E_{\tau \sim p}F(\tau)$, therefore:

$$\arg\min_\theta(E_{\tau \sim p_\theta} \log p_\theta(\tau) + \max_F(E_{\tau \sim p_s}F(\tau) - E_{\tau \sim p_\theta}F(\tau))) \quad (58)$$

Finally, to arrive at Equation 5 we use the fact that $E_{\tau \sim p} \sum_t f(h_t, x_t) = TE_{h \sim \mu, x \sim \pi(\cdot|h)}f(h, x)$, as already noted. All in all, this is a "generalization" from the case of linearity in *known* basis functions, to the case of *unknown* basis functions—where we "linearize" the expression using indicator functions. But we cannot use the same logic to claim the same for *infinite* $\mathcal{X}$ (which is the setting we operate in).

## B Details on Algorithm

**Policy Optimization** Recall the policy update (Equation 12); this corresponds to entropy-regularized reinforcement learning using $f_\phi(h, x)$ as transition-wise reward function. Here we give a brief review of entropy-regularized reinforcement learning [45–47] in our context, as well as the practical method we employ (i.e. soft actor-critic). First, we introduce some standard notation. At any state $h$, define the (soft) "value function" to be the (forward-looking) expected sum of future rewards $f_\phi(h, x)$ as well as entropies $H(\pi(\cdot|h))$. Specifically, let $V_\phi^{\pi_\theta}(h)$ and $Q_\phi^{\pi_\theta}(h, x)$ be given as follows (we omit explicit notation for $t$, as any influence of time is implicit through dependence on variable-length histories):

$$V_\phi^{\pi_\theta}(h) := \mathbb{E}_{\tau \sim p_\theta}[\sum_{u=t}^T f_\phi(h_u, x_u) + H(\pi_\theta(\cdot|h_u))|h_t = h] \quad (59)$$

$$Q_\phi^{\pi_\theta}(h, x) := f_\phi(h, x) + \mathbb{E}_{\tau \sim p_\theta}[\sum_{u=t+1}^T f_\phi(h_u, x_u) + H(\pi_\theta(\cdot|h_u))|h_t = h, x_t = x] \quad (60)$$

Let $\pi_{\theta^*}$ denote the optimal policy (i.e. that minimizes loss $\mathcal{L}_{\text{policy}}$), and $Q_\phi^{\pi_{\theta^*}}$ its corresponding value function. An elementary result is that the optimal policy assigns probabilities to $x$ proportional to the exponentiated expected returns of energy and entropy terms of all trajectories that begin with $(h, x)$:

$$\pi_{\theta^*}(x|h) = \frac{\exp(Q_\phi^{\pi_{\theta^*}}(h, x))}{\int_\mathcal{X} \exp(Q_\phi^{\pi_{\theta^*}}(h, x))dx} \quad (61)$$

Now, for any transition policy $\pi_\theta$ (i.e. not necessarily optimal with respect to $f_\phi$), the value function $Q_\phi^{\pi_\theta}$ is the unique fixed point of the following (soft) Bellman backup operator $\mathbb{B}_\phi^{\pi_\theta} : \mathbb{R}^{\mathcal{H} \times \mathcal{X}} \to \mathbb{R}^{\mathcal{H} \times \mathcal{X}}$:

$$(\mathbb{B}_\phi^{\pi_\theta} Q)(h, x) := f_\phi(h, x) + \mathbb{E}_{x' \sim \pi_\theta(\cdot|h')}[Q(h', x') - \log \pi_\theta(x'|h')] \tag{62}$$

and hence—in theory—$Q_\phi^{\pi_\theta}$ may be computed iteratively by repeatedly applying the operator $\mathbb{B}_\phi^{\pi_\theta}$ starting from any function $Q \in \mathbb{R}^{\mathcal{H} \times \mathcal{X}}$; this is referred to as the (soft) "policy evaluation" procedure.

Using $Q_\phi^{\pi_\theta}$, we may then perform (soft) "policy improvement" to update the policy $\pi_\theta$ towards the exponential of its value function, and is guaranteed to result in an improved policy (in terms of $Q_\phi^{\pi_\theta}$):

$$\theta' \leftarrow \arg\min_\theta D_{\mathrm{KL}}\left(\pi_{\theta'}(\cdot|h) \Big\| \frac{\exp(Q_\phi^{\pi_\theta}(h, \cdot))}{\int_{\mathcal{X}} \exp(Q_\phi^{\pi_\theta}(h, x))dx}\right) \tag{63}$$

for all $h \in \mathcal{H}$. In theory, then, finding the optimal policy can be approached by repeatedly applying the above policy evaluation and policy improvement steps starting from any initial policy $\pi_\theta$; this is referred to as (soft) "policy iteration". However, in large continuous domains (such as $\mathcal{H} \times \mathcal{X}$) doing this exactly is impossible, so we need to rely on function approximation for representing value functions.

**Practical Algorithm** Precisely, the soft actor-critic approach is to introduce a function approximator to represent the value function (i.e. the "critic") parameterized by $\psi$, in addition to the policy itself (i.e. the "actor") parameterized by $\theta$, and to alternate between optimizing both with stochastic gradient descent [57]. Specifically, the actor performs soft policy improvement steps as before, but now using $Q_\psi$:

$$\mathcal{L}_{\mathrm{actor}}(\theta; \phi, \psi) := \mathbb{E}_{h \sim \mathcal{B}} \mathbb{E}_{x \sim \pi_\theta(\cdot|h)}[\log \pi_\theta(x|h) - Q_\psi(h, x)] \tag{64}$$

where $\mathcal{B}$ is a replay buffer of samples generated by $\pi_\theta$ (that is, instead of enumerating all $h \in \mathcal{H}$, we are relying on $h \sim \mathcal{B}$). Note that the normalizing constant is dropped as it does not contribute to the gradient. The critic is trained to represent the value function by minimizing squared residual errors:

$$\mathcal{L}_{\mathrm{critic}}(\psi; \phi) := \mathbb{E}_{h, x \sim \mathcal{B}}(Q_\psi(h, x) - Q_\psi^{\mathrm{target}}(h, x))^2 \tag{65}$$

with (bootstrapped) targets:

$$Q_\psi^{\mathrm{target}}(h, x) := f_\phi(h, x) + \mathbb{E}_{x' \sim \pi_\theta(\cdot|h')}[Q_\psi(h', x') - \log \pi_\theta(x'|h')] \tag{66}$$

Together, this provides a way to minimize the policy loss $\mathcal{L}_{\mathrm{policy}}$ (Equation 12). The complete Time-GCI algorithm simply alternates between this and minimizing the energy loss $\mathcal{L}_{\mathrm{energy}}$ (Equation 11):

$$\mathcal{L}_{\mathrm{energy}}(\phi; \theta) := -\mathbb{E}_{\tau \sim p_s} \log d_{\theta, \phi}(\tau) - \mathbb{E}_{\tau \sim p_\theta} \log\left(1 - d_{\theta, \phi}(\tau)\right) \tag{67}$$

Hence in Algorithm 1, gradient updates for the energy, policy, and critic are interleaved with policy rollouts. Note that several standard approximations are being used. First, the replay buffer provides samples $h \sim \mathcal{B}$ for optimizing the policy (in both actor and critic updates), instead of covering the entire space $\mathcal{H}$ (which is uncountable). Second, in the energy loss negative samples $\tau \sim \mathcal{B}$ are used in lieu of sampling fresh from $p_\theta$ at every iteration (this is known to give the benefit of providing more diverse negative samples). Finally, also per usual samples from the dataset $\tau \sim \mathcal{D}$ is used in lieu of $p_s$.

**Practical Considerations** First, in practice we must use a *vector representation* of histories $h \in \mathcal{H}$; here we use RNNs to encode histories into fixed-length vectors, which can then be treated as regular "states" in continuous space. Like our choice of policy optimization, this is also an arbitrary design choice—we could just as conceivably have used e.g. temporal convolutions, attention mechanisms, etc.

Second, *interleaving* multiple gradient updates of different networks requires some care: In soft actor-critic itself, policy updates have to be sufficiently small, and/or critic updates have to be sufficiently frequent, to prevent divergence. The situation is analogous when interleaving this with energy gradient updates as well: Both actor and energy updates have to be sufficiently small, and/or critic updates have to be sufficiently frequent. That said, the energy updates are indeed decoupled from the policy updates: Regardless of how quickly/slowly the policy is learning, the energy can learn on their negative samples. In practice, we perform multiple critic updates for every update of the policy and energy functions.

Finally, note that in large continuous domains such as $\mathcal{H} \times \mathcal{X}$ it is necessary to *pre-train* the networks beforehand such that optimization of the complete algorithm actually converges: On the one hand, the policy side requires a sufficiently good energy signal to actually make progress, and on the other hand, the energy side requires a sufficiently good policy providing challenging enough negative samples to actually make progress. Pre-training networks separately is standard in actor-critic methods (see for instance [35]); here we take a similar approach but with the addition of the energy update step as well:

1. Policy-only: $\pi_\theta$ is pre-trained using maximum likelihood;
2. Energy-only: $f_\phi$ is pre-trained using $\mathcal{L}_{\text{energy}}(\phi; \theta)$, holding $\pi_\theta$ fixed;
3. Critic-only: $Q_\psi$ is pre-trained using $\mathcal{L}_{\text{critic}}(\psi; \phi)$, holding $\pi_\theta$, $f_\phi$ fixed; and finally,
4. All: $f_\phi$, $\pi_\theta$, and $Q_\psi$ are trained on $\mathcal{L}_{\text{energy}}(\phi; \theta)$, $\mathcal{L}_{\text{actor}}(\theta; \phi, \psi)$, and $\mathcal{L}_{\text{critic}}(\psi; \phi)$ (cf. Algorithm 1).

## C   Details on Experiments

**Benchmark Algorithms**  Except where components are standardized (see below), we use the publicly available source code when constructing the benchmark algorithms; references are in the following:

- T-Forcing [5]: (straightforward MLE with ground-truth conditioning)
- P-Forcing [10]: `https://github.com/anirudh9119/LM_GANS`
- C-RNN-GAN [18]: `https://github.com/olofmogren/c-rnn-gan`
- COT-GAN [20]: `https://github.com/tianlinxu312/cot-gan`
- RC-GAN [21]: `https://github.com/ratschlab/RGAN`
- TimeGAN [12]: `https://github.com/jsyoon0823/TimeGAN`

**Dataset Sources**  We use the original source code for preprocessing sines and UCI datasets from TimeGAN (`https://github.com/jsyoon0823/TimeGAN`). For MIMIC-III, we extract 52 clinical covariates including vital signs (e.g. respiratory rate, heart rate, O2 saturation) and lab tests (e.g. glucose, hemoglobin, white blood cell count) aggregated every hour during their ICU stay up to 24 hours.

- Sines [5]: `https://github.com/jsyoon0823/TimeGAN`
- Energy [10]: `archive.ics.uci.edu/ml/datasets/Appliances+energy+prediction`
- Gas [18]: `archive.ics.uci.edu/ml/datasets/Gas+sensor+array+temperature+modulation`
- Metro [20]: `archive.ics.uci.edu/ml/datasets/Metro+Interstate+Traffic+Volume`
- MIMIC-III [21]: `https://physionet.org/content/mimiciii/1.4/`

**Encoder Networks**  For fair comparison, analogous network components across all benchmarks share the same architecture where possible. In particular, all components taking $h_t$ as input require an encoder network to construct fixed-length vector representations of variable-length histories $(x_1, ..., x_t)$. To do so, these components use an LSTM network with a hidden layer of size 32 to compute hidden states for representing $h$. These are not shared: separate components have their own encoder networks.

**Task-Specific Networks**  Then, for mapping from $h_t$ and/or $x_t$ to task-specific output variables, we use a fully-connected network with two hidden layers of size 32 and ELU activations. Where both $h_t$ and $x_t$ serve as inputs, their vectors are concatenated. For instance, the energy network for TimeGCI contains one such network for computing $f_\phi$ (as well as trainable parameter for $Z_\phi$). The same applies to the mapping from $h_t$ and/or $x_t$ to implicit generator outputs (as in C-RNN-GAN and RC-GAN), black-box discriminator output (as in P-Forcing, C-RNN-GAN, and RC-GAN), transition policies (as in T-Forcing and P-Forcing), and critic values (for TimeGCI). Note that TimeGAN and COT-GAN were designed with additional unique components/losses that operate as a unit (e.g. the embedding/recovery networks for generating/discriminating within latent space); we use their original source code.

**Replay Buffer**  The replay buffer $\mathcal{B}$ has a fixed size; once filled, new samples stored replace the oldest still in the buffer. Sampling from the buffer operates as follows: In updating the energy, we require $\tau \sim \mathcal{B}$ (that is, in lieu of $p_\theta$, cf. Equation 67); this is done by randomly sampling a batch of trajectories from the replay buffer, without replacement. In policy the actor, we require $h \sim \mathcal{B}$ (cf. Equation 64); this is done by first randomly sampling a batch of trajectories from the replay buffer, and then randomly sampling a cutoff time $t$ to obtain a batch of subsequences $h_t$. Finally, in updating the critic, we require $h, x \sim \mathcal{B}$ (cf. Equation 65); this is similarly done by first randomly sampling a batch of trajectories from the replay buffer, then randomly sampling a cutoff $t$ to yield a batch of $(h_t, x_t)$ pairs.

**Hyperparameters**  In all experiments, we use the following hyperparameters for TimeGCI and for benchmarks (wherever applicable): The replay buffer is of size $|\mathcal{B}| = 10,000$. The hidden dimension of all encoder and task-specific networks is 32. The entropy regularization (in the actor/policy loss) is $\alpha = 0.2$. The policy network is pre-trained for 2,000 steps, energy for 4,000, and critic for 20,000. The complete algorithm is trained for up to 50,000 steps with checkpointing and early stopping (triggerable every 1,000 iterations, if performance does not improve). The Adam optimizer is used for all losses, and batch size $M = 64$. Learning rates: For energy networks $\lambda_{\text{energy}} = 0.0001$, for policy networks $\lambda_{\text{policy}} = 0.0001$ (same for implicit generator networks), for critic networks $\lambda_{\text{critic}} = 0.001$, and for black-box discriminator networks $\lambda_{\text{discrim}} = 0.001$. (Note that the critic/discriminator networks are

updated more greedily). Per usual in soft actor-critic algorithms, we also employ a lagged target critic network (i.e. used for bootstrapping); this is updated using polyak averaging at a rate of $\tau = 0.005$.

**Performance Metrics** We use the original source code for computing the TSTR metric (i.e. Predictive Score), publicly available at: `https://github.com/jsyoon0823/TimeGAN`; this is straightforwardly modified to compute TSTR scores for horizons of lengths three (+3 Steps Ahead) and five (+5 Steps Ahead). Likewise, we use the original source code for computing the cross-correlation score ($x$-Corr. Score), this is also publicly available at: `https://github.com/tianlinxu312/cot-gan`.

# D    Clarifying the Analogy

This section clarifies the analogy of "generation as imitation". While the point of our investigation is to explicitly invite an analogy between synthetic time-series generation and imitation learning, they are different problems with different considerations. In particular, TimeGCI is not suitable for imitation learning per se, and we make no claim that it is (Appendix D.1). Conversely, while existing adversarial imitation learning methods may be naively applied to time-series generation, their optimization objectives are different, and we empirically verify they do not perform well (Appendix D.2).

## D.1    Can TimeGCI be used for Imitation Learning?

At its core, time-series generation is the problem of modeling a distribution of trajectories $p(\tau)$ faithfully, and is what TimeGCI does. In MDP parlance, in time-series generation the "environment dynamics" are by construction *known* and *deterministic*: $\omega(\cdot|z_t, u_t)$ is the Dirac delta centered at $z_{t+1} = (u_1, \ldots, u_t)$. We have $p(\tau) = \prod_t \pi(u_t|z_t)$. However, in imitation learning the dynamics are generally *unknown* and *stochastic*. We have $p(\tau) = \prod_t \pi(u_t|z_t)\omega(z_t|z_{t-1}, u_{t-1})$. This difference is crucial. Consider walking through the logic of Section 3, but applying it to imitation learning instead. Begin with the gradient update for learning the Gibbs parameters (Equation 7). To avoid costly inner-loop optimization of $\theta$ to completion, we again consider (1) importance sampling, as well as our preferred method of (2) contrastive estimation. We will see that in imitation learning neither of these work out of the box. For (1), the importance weights now become $\exp(F_\phi(\tau))/p_\theta(\tau)$, where $p_\theta(\tau) = \prod_t \pi_\theta(u_t|z_t)\omega(z_t|z_{t-1}, u_{t-1})$. But the problem is that we do not have access to $\omega$. We can naively *estimate* it beforehand (clearly inadvisable). Or, we can perform a specific *approximation* in the energy model: $p_\phi(\tau) = \exp(F_\phi(\tau) - \log Z_\phi) \approx \frac{1}{Z_{\phi,\omega}}\exp(F_\phi(\tau))\prod_t \omega(z_t|z_{t-1}, u_{t-1})$ such that $\omega$ cancels out from the expression. Instead of modeling the distribution of trajectories $p(\tau)$ exactly, this now assumes that transition randomness has a limited effect on behavior and that the partition function is constant for all random outcome samples (see e.g. [67]). The situation is similar for (2), where it is easy to see that in imitation learning scenarios Equation 10 is no longer accessible without performing the above approximation to cancel out $\omega$. Three points bear emphasis. First, if we are willing to make this simplification, we are no longer speaking of TimeGCI anymore. Instead, it is easy to show that this effectively becomes a sort of trajectory-centric "maximum *causal* entropy" inverse optimal control (see e.g. [39]), which is no longer learning $p(\tau)$ with an exact objective. Second, in general we have little reason to believe that such a trajectory-centric approach would work well in imitation learning: The variance of sample-based estimates is now exacerbated by the unknown and stochastic environment dynamics in imitation learning, such that trajectory-based sampling is generally known to perform quite poorly (see e.g. [54]). In fact, for this reason modern adversarial imitation learning methods almost always take a transition-centric approach. Third, empirically a similar structured-classifier approach (with the above approximation) has indeed been found to be unworkable in practice due to high variance (see [54] Section 4, implementing [53]); moreover, an importance-sampling based approach (with the above approximation) has been made to work, but has required hand-crafted, domain-specific regularization to work, and the learned energy functions only explain the demonstrations locally [68]. For these reasons, we do not evaluate TimeGCI on imitation learning scenarios. It is simply not applicable without modifying it to become a different method altogether. Modern imitation learning methods are designed specifically to handle the unknown and stochastic nature of environment transitions. In particular, transition-centric approaches (i.e. scoring $(z, u)$ pairs) have been found to be more effective empirically. What we are doing with TimeGCI, on the other hand, is to focus on an exact objective for learning time-series trajectories $p(\tau)$, which requires a trajectory-centric approach. This is because this allows us to equate TimeGCI with a special kind of noise-contrastive estimation, for which we can enjoy some theoretical guarantees. (Note that these properties are lost when performing the above approximation in an imitation learning setting).

## D.2 Can AIL be used for Time-series Generation?

Table 4: *Performance Comparison of GAIL, AIRL, and TimeGCI.* Bold numbers indicate best-performing results.

| Benchmark | Metric | Sines | Energy | Gas | Metro | MIMIC-III |
|---|---|---|---|---|---|---|
| GAIL-SAC | Predictive Score | $0.447 \pm 0.002$ | $0.261 \pm 0.001$ | $0.022 \pm 0.002$ | $0.257 \pm 0.001$ | $0.017 \pm 0.001$ |
| | +3 Steps Ahead | $0.472 \pm 0.003$ | $0.262 \pm 0.001$ | $0.048 \pm 0.001$ | $0.250 \pm 0.001$ | $0.012 \pm 0.001$ |
| | +5 Steps Ahead | $0.542 \pm 0.004$ | $0.261 \pm 0.001$ | $0.074 \pm 0.002$ | $0.245 \pm 0.003$ | $0.011 \pm 0.001$ |
| | $x$-Corr. Score | $6.798 \pm 0.014$ | $142.457 \pm 0.471$ | $111.124 \pm 0.232$ | $1.044 \pm 0.031$ | $540.89 \pm 0.159$ |
| AIRL-SAC | Predictive Score | $0.223 \pm 0.006$ | $0.283 \pm 0.002$ | $0.0478 \pm 0.0015$ | $0.243 \pm 0.001$ | $0.018 \pm 0.005$ |
| | +3 Steps Ahead | $0.381 \pm 0.002$ | $0.295 \pm 0.003$ | $0.0883 \pm 0.0029$ | $0.250 \pm 0.001$ | $0.019 \pm 0.003$ |
| | +5 Steps Ahead | $0.349 \pm 0.005$ | $0.321 \pm 0.001$ | $0.1176 \pm 0.0040$ | $0.251 \pm 0.002$ | $0.019 \pm 0.001$ |
| | $x$-Corr. Score | $10.397 \pm 0.005$ | $202.61 \pm 0.119$ | $144.79 \pm 0.3199$ | $1.286 \pm 0.098$ | $2635.57 \pm 0.050$ |
| TimeGCI | Predictive Score | $\mathbf{0.097 \pm 0.001}$ | $\mathbf{0.251 \pm 0.001}$ | $\mathbf{0.018 \pm 0.000}$ | $\mathbf{0.239 \pm 0.001}$ | $\mathbf{0.002 \pm 0.000}$ |
| | +3 Steps Ahead | $\mathbf{0.104 \pm 0.001}$ | $\mathbf{0.251 \pm 0.001}$ | $\mathbf{0.042 \pm 0.001}$ | $\mathbf{0.239 \pm 0.001}$ | $\mathbf{0.001 \pm 0.000}$ |
| | +5 Steps Ahead | $\mathbf{0.109 \pm 0.001}$ | $\mathbf{0.251 \pm 0.001}$ | $\mathbf{0.067 \pm 0.001}$ | $\mathbf{0.239 \pm 0.001}$ | $\mathbf{0.001 \pm 0.000}$ |
| | $x$-Corr. Score | $\mathbf{1.195 \pm 0.011}$ | $\mathbf{105.2 \pm 0.433}$ | $\mathbf{47.91 \pm 0.811}$ | $\mathbf{0.738 \pm 0.019}$ | $\mathbf{194.3 \pm 0.180}$ |

Conversely to the preceding, we can also ask: Can adversarial imitation learning methods work for time-series generation (hence can we compare them against TimeGCI)? The answer is "yes", they can be *applied* to time-series generation, but "no", there is no reason to expect they would perform better. Modern adversarial imitation learning methods come in two broad flavors: (1) GAIL-based [107–110], which do not recover a reward function, and (2) AIRL-based [54, 69, 111], which do recover a reward function. Both are transition-centric (i.e. scoring $(z, u)$ pairs). For (1): Here, optimization is of the familiar GAN-like objective: $\min_\theta \max_\phi \mathbb{E}_{z,u \sim \mu_s} \log d_\phi(z, u) + \mathbb{E}_{z,u \sim \mu_\theta} \log \left(1 - d_\phi(z, u)\right)$, which minimizes the JS-divergence between the *state-action* occupancy measures (i.e. distribution of transitions, in time-series generation terms) induced by the learned and source policies. Note that this is the same as the TimeGAN objective [12], modulo entropic/supervised regularization. As noted before, matching the transition marginals is indirectly performing the "global" sort of moment-matching. However, as pertains time-series generation, the key difference between GAIL and TimeGAN is that the former performs policy optimization for the generator, whereas the latter simply performs backpropagation. Otherwise, note that both methods are adversarial (viz. saddle-point optimization) and do not learn an explicit energy (viz. black-box discriminator). In light of this, we may consider GAIL for experimental comparison (to be consistent, we also use SAC as the policy optimization method); see Table 4 above. For (2): Here, the discriminator is first constructed to be *transition*-based: $d_{\theta,\phi}(z, u) = (\exp(g_\phi(z, u))/(\exp(g_\phi(z, u)) + \pi_\theta(u|z)))$. Note that this actually breaks the relationship with modeling the distribution of *trajectories* directly $p_\phi(\tau) = \exp(F_\phi(\tau) - \log Z_\phi)$. Importantly, if we apply AIRL to time-series generation, we lose the convergence property that comes from the relationship with noise-contrastive estimation (cf. Proposition 4), and the gradient of the discriminator is no longer related to the original trajectory-wise energy-based gradient (cf. Proposition 5). In fact, all we are left with is that the global optimum is a correct optimum—but this by itself is not helpful (e.g. the global optimum of naive MLE is also a correct optimum, but there is no guarantee that it can be found effectively). In fact, it has been formally shown that the original theoretical justifications for AIRL are incorrect (see e.g. [111] Sec. 2.4.2, or [112] Sec. B.2). (In stochastic domains in imitation learning, it is true that the empirical benefit of variance-reduction from sampling in this transition-based manner still dominates. In time-series generation, however, there is no stochasticity in the "environment", and doing it this way may unnecessarily bias our objective of learning $p(\tau)$). In light of this, we may consider AIRL for experimental comparison (to be consistent, we also use SAC for policy optimization); see Table 4.

## D.3 Ablation Studies on Sequence Lengths

First, we compare the performance of T-Forcing to TimeGCI, using sequence lengths $T \in \{2, 8, 24\}$. We compute the TSTR predictive score, +3 steps ahead, +5 steps ahead, and $x$-Corr. score as usual for each of these settings. Note that for $T = 2$ we cannot ask the TSTR evaluation model to predict more than one step ahead, since there is no data for more than one step ahead; we indicate this with "N/A". The Gas dataset is used. For ease of interpretation, we consider T-Forcing as the "baseline", and also express the TimeGCI numbers as a fraction of the T-Forcing numbers. The results are consistent with what we would expect: The performance advantage enjoyed by TimeGCI over T-Forcing diminishes as the sequence lengths of the input dataset decreases, and increases as the sequence lengths of the input dataset increases. This is true regardless of what metric we are talking about (i.e. 1/3/5-step TSTR score, or feature correlation score). Moreover, at $T = 2$ there is almost no difference between

the TSTR scores of T-Forcing and TimeGCI. This shows that T-Forcing appears to preserve quite well the one-step ahead relationships of the original data (viz. $T = 2$). However, for longer sequences TimeGCI performs relatively better, which is consistent with the motivation behind using an objective that matches the distribution of trajectories, instead of simply matching transition conditionals:

| Benchmark | Metric | $T = 2$ | $T = 8$ | $T = 24$ |
|---|---|---|---|---|
| T-Forcing | Predictive Score | $0.038 \pm 0.001$ | $0.036 \pm 0.007$ | $0.035 \pm 0.003$ |
| | +3 Steps Ahead | N/A | $0.076 \pm 0.007$ | $0.080 \pm 0.001$ |
| | +5 Steps Ahead | N/A | $0.114 \pm 0.003$ | $0.111 \pm 0.001$ |
| | $x$-Corr. Score | $160.3 \pm 0.346$ | $154.2 \pm 0.219$ | $150.8 \pm 0.067$ |
| TimeGCI | Predictive Score | $0.037 \pm 0.001$ | $0.022 \pm 0.000$ | $0.018 \pm 0.000$ |
| | +3 Steps Ahead | N/A | $0.048 \pm 0.001$ | $0.042 \pm 0.001$ |
| | +5 Steps Ahead | N/A | $0.071 \pm 0.001$ | $0.067 \pm 0.001$ |
| | x-Corr. Score | $140.4 \pm 0.173$ | $50.72 \pm 0.816$ | $47.91 \pm 0.811$ |
| Ratio of Mean | Predictive Score | 97.37% | 61.11% | 51.43% |
| | +3 Steps Ahead | N/A | 63.16% | 52.50% |
| | +5 Steps Ahead | N/A | 62.28% | 60.36% |
| | x-Corr. Score | 87.59% | 32.89% | 31.77% |

Second, we use a new synthetic simulation where what is being generated is a set of multi-dimensional sinusoidal waves which—if not perturbed by noise—correspond to different frequencies, phases, etc. This allows us to manually inject noise where we want, and also allows us to simulate "ground truth". First, we train both models (T-Forcing and TimeGCI) using the same training data generated by the simulator, and save these learned models. Then during validation, we obtain a sequence from the simulator, then add some additional noise. Specifically, we add independent Gaussian noise $\mathcal{N}(\mu, \sigma)$, using $\mu = 0$ and $\sigma = 0.1$, to each of the feature dimensions of the wave at time $K$. Then, our task is to predict $t$ steps ahead with the learned models. To be clear, we ask the learned models to predict ahead using the information up to and including this (perturbed) $K$-th step; if the prediction is for $t > 1$, we perform open-loop sampling as usual. We can measure the model's prediction error (i.e. evaluating how well the model forecasts the future based on the provided histories as input, along with the perturbation). Note that this is entirely different from the TSTR predictive score hitherto used (i.e. evaluating how well the model's freely generated output dataset preserves the characteristics of the input dataset); this is only done as an additional sensitivity analysis: By analogy to imitation learning, we can interpret this experiment as a special case of "action-matching" (i.e. based on ground-truth state/history as input, we can measure the "prediction error" between the action chosen by the imitator policy and the actual action taken by the expert)—but specifically where the state/history has been perturbed with additional noise. We ask each model to perform $t$-steps ahead prediction, where $t \in \{1, 2, 3, 4, 5\}$ as a sensitivity. When we add noise, we use values $\{\sigma, 2\sigma, 3\sigma, 4\sigma, 5\sigma\}$ as a sensitivity. Given any sequence, the step $K$ at which noise is artificially added (and at which point the models are asked to perform $t$-step ahead predictions) is uniformly randomly sampled from $K \in \{1, 2, ..., T-1\}$.

| Noise at $K$ | Benchmark | 1-Ahead MSE | 2-Ahead MSE | 3-Ahead MSE | 4-Ahead MSE | 5-Ahead MSE |
|---|---|---|---|---|---|---|
| $\sigma$ | T-Forcing | $0.024 \pm 0.000$ | $0.029 \pm 0.001$ | $0.036 \pm 0.001$ | $0.041 \pm 0.001$ | $0.047 \pm 0.001$ |
| | TimeGCI | $0.024 \pm 0.000$ | $0.025 \pm 0.001$ | $0.026 \pm 0.001$ | $0.028 \pm 0.001$ | $0.031 \pm 0.001$ |
| $2\sigma$ | T-Forcing | $0.055 \pm 0.000$ | $0.059 \pm 0.001$ | $0.065 \pm 0.001$ | $0.071 \pm 0.002$ | $0.076 \pm 0.001$ |
| | TimeGCI | $0.055 \pm 0.001$ | $0.055 \pm 0.001$ | $0.057 \pm 0.001$ | $0.059 \pm 0.000$ | $0.061 \pm 0.001$ |
| $3\sigma$ | T-Forcing | $0.106 \pm 0.001$ | $0.109 \pm 0.002$ | $0.116 \pm 0.001$ | $0.121 \pm 0.001$ | $0.127 \pm 0.002$ |
| | TimeGCI | $0.106 \pm 0.001$ | $0.106 \pm 0.001$ | $0.107 \pm 0.001$ | $0.109 \pm 0.001$ | $0.111 \pm 0.001$ |
| $4\sigma$ | T-Forcing | $0.176 \pm 0.001$ | $0.181 \pm 0.002$ | $0.185 \pm 0.001$ | $0.191 \pm 0.002$ | $0.197 \pm 0.002$ |
| | TimeGCI | $0.176 \pm 0.002$ | $0.176 \pm 0.002$ | $0.177 \pm 0.001$ | $0.179 \pm 0.002$ | $0.181 \pm 0.001$ |
| $5\sigma$ | T-Forcing | $0.266 \pm 0.003$ | $0.271 \pm 0.002$ | $0.277 \pm 0.002$ | $0.280 \pm 0.002$ | $0.287 \pm 0.002$ |
| | TimeGCI | $0.266 \pm 0.002$ | $0.267 \pm 0.003$ | $0.267 \pm 0.002$ | $0.270 \pm 0.003$ | $0.272 \pm 0.003$ |

The results give some orthogonal intuition as to why the proposed method is better; specifically, is it because of better first-step prediction, or is it because of better robustness when having small errors in the previous steps? Observe that for *single*-step prediction, there is virtually no difference between the performance of T-Forcing and TimeGCI when evaluating the prediction MSE. The more noise added, the worse both models perform; but there is little difference between them, no matter the noise. On the other hand, for *multi*-step prediction, when predicting multiple steps ahead using open-loop sampling, T-Forcing performs worse than TimeGCI. In fact, the gap between their performances increases as $t$ increases. So it appears that it is not the case that TimeGCI simply has better first-step prediction per se; also, it appears that TimeGCI does have better robustness when having errors in the previous steps. Because, while the 1-step performance of both T-Forcing and TimeGCI are impacted almost equally, TimeGCI appears to have an advantage in terms of "propagating" less of that error into later time steps.