# OpenReview forum: "Time-series Generation by Contrastive Imitation"
_NeurIPS.cc/2021/Conference — NeurIPS 2021 Poster_

### Official Review · Reviewer_WGYR · 2021-07-16

**Rating:** 6
**Confidence:** 4

**Summary:**

The paper proposes a method for sequential data generation that tries to avoid the pitfalls of prior MLE- and GAN-based objectives. The proposed approach starts with the global moment-matching objective (that mitigates the compounding errors). The true data distribution is modeled as an EBM and the parameters of the energy function are learned with contrastive learning (NCE). The trajectory distribution induced by a (learned) generation policy is used as the “noise” for NCE. In the inner loop, the generation policy is optimized to reduce the reverse-KL to the EBM’s trajectory distribution – which is equivalent to entropy-regularized RL with the energy function as the reward. Experiments using 5 tabular time-series datasets pits TimeCGI favorably in comparison to prior methods on the metrics of predictive and correlation scores.

**Ethical Concerns:**

I don't see any ethical concerns with this paper.

**Limitations And Societal Impact:**

There is a good discussion on these issues in Section 6. I am further interested to know the directions to improve the scalability of TimeCGI (point 4. in the main review).

**Main Review:**

Overall, I find the paper to be well-written and nicely organized. The authors address an important problem of sequential data generation over long-horizon, which has many real-world use cases. I would like the authors to consider the following issues:
1.	Comparison to [63]: Equations 10-12 in section 3.2 bear a lot of resemblance to the work in [63]. Specifically, the idea of replacing the black-box GAN discriminator with a structured classifier with decoupled generator density and EBM for the true data was first proposed in [63], to the best of my knowledge. Eq. 10 in this paper (after taking into account the errata from the appendix) is the same as the third equation in section 3.1 of [63]. Eq. 11-12 of this paper appear in [63] as well. I appreciate the important differences between RL/IRL and the time-series generation setting considered in this paper. Yet, I believe that the phrasing in Section 3.2 gives the impression that the structured classification and the NCE training are the new contributions of this paper. Could the authors contrast their optimization objectives (Eq. 10-12) with those used in [63]?

2.	Proposition 5: This statement in its current form does not appear to be true. As the appendix shows, the gradient equality only holds when p_\phi is normalized, i.e., at the global optimum of the energy function. I would recommend the text in the main section be revised to add the proper caveats.

3.	Information on datasets: For the datasets considered, could you provide some information regarding:

     a.	The trajectory-length (horizon) of the true and the generated sequences

     b.	The dimension of the features space

     c.	The stochasticity and/or noise characteristics of the true data generation process


4.	Improving Scalability: TimeCGI is closest to [63] in terms of learning the structured classifier over the entire sequence, instead of stepwise transitions. However, [64] argues that [63] is difficult to apply to complex problem settings since it operates at the level of the entire trajectory. It would be informative to expand on the potential challenges when scaling TimeCGI to long-horizon sequences with stochastic transitions. Could the variance of the gradient-estimator for the energy function be a problem? Thoughts on possible directions to improve scalability would be interesting.


$$\newline$$
Possible typos?
1.	Line 119: should there be a negative sign with the likelihood-loss?
2.	Line 186: should it be product over \pi, rather than sum?

$$\newline$$
[63] A connection between generative adversarial networks, inverse reinforcement learning, and energy-based models, Finn et al.

[64] Learning robust rewards with adversarial inverse reinforcement learning, Fu et al.


&nbsp;
&nbsp;

---

**Post-rebuttal update:** Thank you to the authors for a detailed rebuttal. I maintain my position that the paper is (marginally) above the acceptance threshold. What would make it a clear accept is evaluation on datasets that highlight the robustness of the method to transition stochasticity (i.e. high "branching factor") and trajectory horizon (current datasets generate sequences of 20-30 steps).






**Time Spent Reviewing:**

5

---

> ### Author Response · Authors · 2021-08-10
> **Response to Reviewer WGYR [Part 5/5]**
>
> ---
>
> ### **(E) Typos**
>
> ---
>
> **Line 119 (negative sign)**: You are correct. This is now fixed.
>
> **Line 186 (product sign)**: You are correct. This is now fixed.
>
> ---
>
> ### **References**
>
> ---
>
> - [101] Hastie et al. (2009). Unsupervised as supervised learning, elements of statistical learning.
> - [102] Mikolov et al. (2013). Distributed representations of words and phrases and their compositionality.
> - [103] Mnih & Kavukcuoglu (2013). Learning word embeddings efficiently with noise-contrastive estimation.
> - [104] Kim & Bengio (2016). Deep directed generative models with energy-based probability estimation.
> - [105] Zhang et al. (2016). Collective noise contrastive estimation for policy transfer learning.
> - [106] Arenz et al. (2020). Non-adversarial imitation and connections to adversarial methods.
> - [107] Ni et al. (2020). f-irl: Inverse reinforcement learning via state marginal matching.

---

> ### Author Response · Authors · 2021-08-10
> **Response to Reviewer WGYR [Part 4/5]**
>
> ---
>
> ### **(B) Proposition 5**
>
> ---
>
> We agree that the current truncated statement is misleading.
>
> ***Update***: We have now revised Proposition 5 in the main text to state the more general form, along with the caveat from Appendix A.
>
> ---
>
> ### **(C) Information on datasets**
>
> ---
>
> We agree that the information mentioned would be helpful for reference.
>
> Note that we use the original source code for preprocessing sines and the UCI datasets by [12], publicly available at [90], which produces sliced sequences of length 24. This is important for comparison purposes, since [12] is the recent method using publicly available datasets. For context, note that the range of features and sequence lengths are comparable to experiments in other works on *tabular* time-series data settings. For instance, RC-GAN [21] uses sines of length 30, smooth functions of length 30, and intensive care unit (ICU) patient data of length 16. COT-GAN [20] uses autoregressive processes of length 20, noisy oscillations of length 20, and sprites of length 13. TimeGAN [12] uses sines of length 24, as well as similar UCI datasets of length 24.
>
> | Dataset | Dimension | Length | Avg. autocor. | +3 Lag | +5 Lag |
> |:--|:--|:--|:--|:--|:--|
> |Sines    | 5|24|0.875|0.623|0.377|
> |Metro    | 9|24|0.429|0.200|0.029|
> |Gas      |20|24|0.656|0.382|0.170|
> |Energy   |29|24|0.702|0.411|0.176|
> |MIMIC-III|52|24|0.532|0.212|0.059|
>
> Also note that we do not consider specialized *media*, such as audio/signal waves (often much finer---but structured---temporal resolutions) or video frames (much finer---but structured---spatial resolution); media sequences require specialized architectures in the generator that involve technical details orthogonal to our contributions.
>
> ***Update***: We have now included this information in a (new) *Table 4* and accompanying remarks as part of Appendix C.
>
> ---
>
> ### **(D) Improving scalability**
>
> ---
>
> We agree that scalability is a (very) important potential challenge. First, regarding [64] implementing [63], as noted above the primary difference there is that they are doing IL, and in IL the unknown and stochastic nature of the environment dynamics greatly exacerbates the variance of trajectory-centric sample-based estimates. In contrast, in TG the transition policy $\pi$ alone controls the outcome of a sequence (the "environment" $\omega$ simply deterministically transitions into a next "state" where the next-step feature vector is appended to the current history).
>
> Second, while experimenting on longer sequences (e.g. audio) are beyond the scope of this initial study, future work may investigate using the array of existing tools and for incorporating long-term time dependencies---e.g. convolutional / wavenet / attention-based deep architectures / black-box latent-variable models, physics / expert model-based approaches, etc. We emphasize that innovations on these fronts are orthogonal to our contributions. As noted in the manuscript, the only requirement is that $\pi_{\theta}$ can be sampled and evaluated efficiently.
>
> You are correct that if the transitions are highly stochastic (i.e. the "branching factor" is high), then the variance could become very high, and sample-based estimates could degrade with the horizon. For the datasets we consider, we find---as usual---that pre-training the policy with maximum likelihood, combined with a small enough learning rate, had the most impact in ensuring that learning occurs. An interesting future direction is to see whether training on fixed subsequence lengths would work for longer sequences. For instance, assuming stationarity, it may be the case that sampling subsequences of a shorter lengths (and training on those) is sufficient for generating beyond those lengths.
>
> ***Update***: We have now included a longer discussion of scalability and future work at the end of Appendix C.

---

> ### Author Response · Authors · 2021-08-10
> **Response to Reviewer WGYR [Part 3/5]**
>
> ---
>
> *c. Empirically, does it work*:
>
> There is no empirical verification in [63]. As you also noted, an attempt was made later by the authors of [64], who found that "[their] experimental results show that this results in very poor learning". This is because the *variance* of trajectory-centric sample-based estimates is now exacerbated by the unknown and stochastic environment dynamics in IL. (Instead, they go ahead and propose a transition-centric approach).
>
> In TG, by contrast, we find empirical evidence that this trajectory-centric approach is viable for the types of datasets we consider. Intuitively this may be because, while in IL the interleaved environment dynamics lead to exponentially more random transitions, in TG the transition policy *alone* dictates the outcome of a sequence.
>
> *d. Adversarial IRL*:
>
> Finally, to be doubly thorough, we can also look more closely at the transition-centric alternative proposed by [64]. It forms the basis for a large class of modern adversarial IRL methods, and also uses a structured classifier. However, they do so differently than either case above: it methods work at the transition level, and is not directly optimizing the trajectory-based objective of TG. Specifically, consider AIRL-based methods for IL [64, 65, 106, etc]. Here, the discriminator is first constructed to be *transition*-based:
>
> $$
> d_{\theta,\phi}(z,u)=\tfrac{\exp(g_{\phi}(z,u))}{\exp(g_{\phi}(z,u))+\pi_{\theta}(u|z)}
> $$
>
> Note that this actually breaks the relationship with modeling the distribution of *trajectories* directly $p_{\phi}(\tau)=\exp(F_{\phi}(\tau)-\log Z_{\phi})$. Importantly, if we apply AIRL to TG, we lose the convergence property that comes from the relationship with NCE, and the gradient of the discriminator is no longer related to the original trajectory-wise energy-based gradient. In fact, all we are left with is that the global optimum is a correct optimum---but again this by itself is not helpful (e.g. the global optimum of naive MLE is also a correct optimum, but there is no guarantee that it can be found effectively). In fact, it has been formally shown that the original theoretical justifications for AIRL are incorrect (see e.g. [106] Sec. 2.4.2, or [107] Sec. B.2).
>
> Now, in stochastic domains in IL, it is true that the *empirical* benefit of variance-reduction from sampling in this transition-based manner still dominates. In TG, however, there is no stochasticity in the "environment", and doing it this way may unnecessarily bias our objective of learning $p(\tau)$.
>
> For completeness, we may add AIRL for experimental comparison (to be consistent, we also use SAC as the policy optimization method). This is now done (TimeGCI results reproduced here for comparison):
>
> | Method | Metric | Sines | Energy | Gas | Metro | MIMIC-III |
> |:--|:--|:--|:--|:--|:--|:--|
> |AIRL-SAC|Predictive Score|0.223 ± 0.006|0.283 ± 0.002|0.0478 ± 0.0015|0.243 ± 0.001|0.018 ± 0.005|
> |AIRL-SAC|+3 Steps Ahead  |0.381 ± 0.002|0.295 ± 0.003|0.0883 ± 0.0029|0.250 ± 0.001|0.019 ± 0.003|
> |AIRL-SAC|+5 Steps Ahead  |0.349 ± 0.005|0.321 ± 0.001|0.1176 ± 0.0040|0.251 ± 0.002|0.019 ± 0.001|
> |AIRL-SAC|x-Corr. Score   |10.397 ± 0.005|202.61  ± 0.119|144.79 ± 0.3199|1.286 ± 0.098|2635.57 ± 0.050|
> |TimeGCI|Predictive Score|0.097 ± 0.001|0.251 ± 0.001|0.018 ± 0.000|0.239 ± 0.001|0.002 ± 0.000|
> |TimeGCI|+3 Steps Ahead  |0.104 ± 0.001|0.251 ± 0.001|0.042 ± 0.001|0.239 ± 0.001|0.001 ± 0.000|
> |TimeGCI|+5 Steps Ahead  |0.109 ± 0.001|0.251 ± 0.001|0.067 ± 0.001|0.239 ± 0.001|0.001 ± 0.000|
> |TimeGCI|x-Corr. Score   |1.195 ± 0.011|105.2 ± 0.433|47.91 ± 0.811|0.738 ± 0.019|194.3 ± 0.180|
>
> ***Update***: We have now included an extended version of this discussion in the (new) *Appendix D*, which clarifies the similarities and differences between the objectives and methods for time-series generation and imitation learning, especially as pertains adversarial IRL. This appendix discussion is referenced in the main manuscript in Section 4, which should clarify the nuances of our contributions, and avoid giving the (erroneous) impression that the structured classification and NCE training *per se* are the new contributions of the paper. Last but not least, the above empirical results are now also included in a (new) *Table 3*.

---

> ### Author Response · Authors · 2021-08-10
> **Response to Reviewer WGYR [Part 2/5]**
>
> ---
>
> ### **(A) Comparison to [63]**
>
> ---
>
> You are correct, [63] is closely related. We agree that additional details regarding our context would benefit the exposition. We emphasize that the novelty here is not generative modeling through structured classification per se (that idea comes from statistics and actually pre-dates [63] by far). Rather, our primary contribution is in simultaneously bringing together three separate strands of work: (1) *density estimation by comparison*, (2) *maximum entropy* IRL, and (3) *synthetic time-series generation*---in particular showing that, in the context of (3), a partial analogy with IL allows us to leverage results from (1), which has actually not been possible/done in (2). Allow us to explain.
>
> **1. Density estimation by comparison**: First, the idea that density estimation can be performed by logistic regression goes back at least to [101]. Noise-contrastive estimation formalized this idea additionally for unnormalized models [51]. Together with negative sampling [102], these ideas have been used often in the sequential domain for (conditional) probabilistic models of language/translation (see e.g. [102, 103]).
>
> **2. Maximum entropy IRL**: Separately, the idea of variational training of energy-based models also goes back, with e.g. [104] discussing its use in the context of deep neural networks. This has then been specifically applied to the idea of maximum entropy IRL with importance sampling [62]. (Note: This application is actually *inexact*---more on that below). With the arrival of GANs, the work of [63] then attempts to show that maximum entropy IRL is "mathematically equivalent" to GANs with a structured discriminator.
>
> **3. Time-series generation**: *In TG, our central focus is in modeling* $p(\tau)$. Since we are also operating in a sequential, free-running domain, we agree that the formalism of TimeGCI (for TG) bears close resemblance to that in [63] (for IL). However, we call to attention four important distinctions regarding contributions:
>
> *a. Mathematical relationship*:
>
> Actually, maximum entropy IRL is *not* mathematically equivalent to GANs with their structured discriminator. That claim is incorrect (and has been noted by e.g. [107]). Even ignoring the practicalities of sample-based estimation for now, the two objectives are in fact different: What is shown in [63] (Sec 3.2) is simply that the derivative of the discriminator loss is equal to the IRL gradient, if we assume that we are already at the global optimum (but without the "weighted" argument of our appendix Prop. 5). This is notable, but on its own is clearly of limited use (e.g. even in theory, does it converge?).
>
> In TG, by contrast, an important contribution is that we identify that such a structured classifier (in the context of sequential generation) is a special instance of *noise-contrastive estimation*. While this is not "equivalent" to variational learning of an EBM, this allows us to leverage results that give us more theoretical guarantees than simply knowing that certain properties hold at the global optimum. In particular, assuming realizability, we have that the estimated parameters actually converge in probability to their true values. The next point addresses why this is specific to TG (and not IL).
>
> *b. Theoretically, does it work*:
>
> Taken literally as proposed in [63] for IL, the structured discriminator cannot actually learn $p(\tau)$. Recall in IL the dynamics are generally *unknown* and *stochastic*. We have $p(\tau)=\prod_{t}\pi(u_{t}|z_{t})\omega(z_{t}|z_{t-1},u_{t-1})$. Consider the third equation in their Sec 3.1: The problem is that we do not have access to $\omega$, and cannot actually evaluate the generator density. To make it work (and this is a point that is often overlooked) we must either naively *estimate* it beforehand (clearly inadvisable), or we must perform a specific *approximation* in the energy model:
>
> $$
> p_{\phi}(\tau)=\exp(F_{\phi}(\tau)-\log Z_{\phi})\approx\tfrac{1}{Z_{\phi,\omega}}\exp(F_{\phi}(\tau))\prod_{t}\omega(z_{t}|z_{t-1},u_{t-1})
> $$
>
> such that $\omega$ cancels out from the expression. Instead of modeling the distribution of trajectories $p(\tau)$ exactly, this now assumes that transition randomness has a limited effect on behavior and that the partition function is constant for all random outcome samples (see e.g. [61]). We actually have no guarantee how good this approximation is. But we do know that the optimization is no longer targeting $p(\tau)$ exactly.
>
> In TG, by contrast, in MDP parlance the "environment dynamics" are by construction *known* and *deterministic*: $\omega(\cdot|z_{t},u_{t})$ is the Dirac delta centered at $z_{t+1}=(u_{1},…,u_{t})$. We have $p(\tau)=\prod_{t}\pi(u_{t}|z_{t})$. So our (fixed) Eq. 10 can actually be evaluated exactly, and so we can target $p(\tau)$ exactly.

---

> ### Author Response · Authors · 2021-08-10
> **Response to Reviewer WGYR [Part 1/5]**
>
> ---
>
> Thank you for your thoughtful comments and suggestions. We give answers to each of the following in turn, as well as pointing out corresponding updates to the revised manuscript:
>
> - (A) Comparison to [63]
> - (B) Proposition 5
> - (C) Information on datasets
> - (D) Improving scalability
> - (E) Typos
>
> ---
>
> By way of preface, kindly allow us to reiterate that our main contributions are: (1) **Formalism**---framing the time-series generation problem in terms of moment-matching to tackle error compounding, which invites an analogy with imitation learning; (2) **Method**---equating a non-adversarial algorithm with a special case of noise-contrastive estimation, which comes with theoretical guarantees; and (3) **Experiments**---illustrating the viability of this approach.
>
> *Notation*: In the following, we abbreviate our problem of time-series generation "TG", and the problem of imitation learning "IL". For consistency, here we use the MDP notation of Section 4 for IL, i.e. $z_{t}$ for states and $u_{t}$ for actions. For consistency, for TG we shall use the same, with the implicit understanding that $z_{t}:=h_{t}$, and $u_{t}:=x_{t}$ (viz. Corollary 6).

---

> ### Author Response · Authors · 2021-08-20
> **Dear Reviewer WGYR**
>
> ---
>
> We are sincerely grateful for your time and energy in the review process.
>
> In light of our responses (Aug 10) and appendix/manuscript updates (Aug 19), we would appreciate if the reviewer kindly let us know of any leftover concerns in the time remaining. We would be happy to do our utmost to address them.
>
> Thank you!
>
> Paper10659 Authors

---

> ### Author Response · Authors · 2021-08-27
> **RE: Dear Reviewer WGYR**
>
> Thank you again for your time and expertise during this review process!
>
> If there were any leftover concerns, we would sincerely appreciate the opportunity to clarify them---before the discussion period for authors ends. We believe our responses (Aug 10) have addressed in detail the full set of questions you had raised, along with corresponding appendix/manuscript updates (Aug 19):
>
> - 1. Comparison to [63]; see Response (A)
> - 2. Proposition 5; see Response (B)
> - 3. Information on datasets; see Response (C)
> - 4. Improving scalability; see Response (D)
> - Typos; see Response (E)
>
> We would appreciate if the reviewer kindly let us know if there were any further questions in the very limited time remaining. We are eager to do our utmost to address them!
>
> Thank you,
>
> Paper10659 Authors

---

### Official Review · Reviewer_QAxC · 2021-07-16

**Rating:** 7
**Confidence:** 3

**Summary:**

The paper proposes an imitation learning method that simultaneously learns a policy and an energy (“discriminator”) function. This is similar to a variational approach to learn a Gibbs distribution over trajectories, where the variational parameters consist of the policy and variational inference consists of running RL.

However, rather than alternating the normal Gibbs energy loss and the variational inference loss of the variational approach, the submission replaces the Gibbs energy loss with an alternate energy loss based on noise-contrastive estimation, which explicitly trains the energy to discriminate between “noise” samples (drawn from the current policy) and expert samples.

It is proved that under the noise-contrastive loss, the energy converges to the expert energy regardless of the noise distribution (under weak conditions), which justifies only partially optimizing the RL step in the alternating optimization. This results in a method potentially much faster than the “naive” variational approach, as it is not necessary to fully solve an inner-loop RL problem for each outer-loop iteration.


**Limitations And Societal Impact:**

Yes

**Main Review:**

Strengths:
+ The method seems easier to train than comparable adversarial imitation learning algorithms
+ Like MaxEnt IRL, but sidesteps the difficulty of running full inference in the training loop
+ Theoretical results provide intuition for why this is a reasonable thing to do
+ Experimentally compared to a variety of other methods and datasets

Weaknesses:
- No experimental comparison to any adversarial imitation learning methods, which have similar properties
- No evaluation on imitation learning scenarios
- Presentation meanders a bit
- Experimental gains are a bit modest

Impact:
Overall, the submission presents an interesting idea that may have the potential to make a significant impact on the community, with some caveats. Although the paper is framed as a generic time series prediction method, it does use terminology and ideas from the imitation learning (IL) literature, and I don’t see why it wouldn’t really be applicable to IL. So, I found it odd that the paper seemed to distance itself from IL.

Anyhow, viewed as an IL method, the proposed method seems appealing in that it would seem to combine the mitigation of error compounding that is a feature of adversarial IL with a more stable and/or faster optimization algorithm. I think it has generally been believed anecdotally that adversarial IL and the closely related MaxEnt IRL fix the error compounding problem present in naive behavioral cloning, and I have seem some theoretical results of this nature (see, e.g., chapter 14 of this book: https://rltheorybook.github.io/), but most of these methods tend to be impractical to actually use, either due to unstable optimization, or slowness due to having to run RL in the inner loop.

This submission seems to propose a reasonable solution to those obstacles, with the caveat that it still requires running an RL step, which could be difficult to manage in practice. The main benefit of the method is that the method used to train the Gibbs energy (noise contrastive estimation) is relatively robust to inaccuracies in the “variational” distribution--i.e., the policy. This means that it is not strictly necessary to run the slow “inner loop” RL step to convergence for every outer loop iteration in order for the outer loop to converge. This should overall improve training stability relative to adversarial IL and should provide speed-ups relative to MaxEnt IRL because of the relaxation of the requirement to run RL to convergence. I believe this is the main takeaway of the paper, and the main reason why this work has the potential for impact.

The experimental results show a pretty significant improvement in the cross-correlation score, which is very encouraging. However, as mentioned elsewhere, trying the method in IL settings would probably be more interesting.

Rigor:
The submission scores some positive points in terms of theoretical rigor for including results that build intuition and support the main claims. The experiments also score points for rigor in that several datasets were tried, along with a variety of competing methods.

One negative aspect in this regard is the fact that the method was not compared to any adversarial imitation learning methods, or evaluated on imitation learning problems. Again, I found it a little perplexing that the work was framed as not being an IL method, despite it seeming very much like an IL method. Although I think this work has potential for impact, that potential is probably much reduced without comparisons to adversarial IL methods in IL settings. Also, the GAIL paper (https://arxiv.org/abs/1606.03476) should be cited, as that features some similar ideas, and is probably the most well-known adversarial IL method.

Another note about the experiments: the “predictive score” does not seem that intuitive or informative to me. For methods that produce a PDF for the policy, one could just compute the data likelihood (i.e., cross entropy) on a held-out dataset.

I did not thoroughly check the theoretical results--in particular, I did not check the proofs of propositions 4 and 5. I skimmed the other results, and they seem reasonable. I did notice one minor issue with Lemma 2: its premise seems stronger than intended. Namely, the premise looks like $\max_f E_{x \sim p_s} f(x) - E_{x \sim p_t} f(x) \leq \epsilon$, where $f$ is unrestricted. However, if $f$ is unrestricted, then the previous expression must equal either $0$ or $\infty$--it can’t be equal to a finite value, because scaling $f$ always leads to a larger value, unless the value is $0$. So, I believe $f$ must be assumed bounded in order for the premise to make sense.

Also, section 3 casually mentions swapping the order of the min and max using the minimax theorem, but this is misleading, because “the” minimax theorem only applies for concave-convex games. This would therefore be inapplicable if the discriminator were parameterized as a nonlinear function.

Novelty:
As far as I know, the main idea of essentially applying NCE in MaxEnt IRL is novel--although one could argue that the general idea of using different generator or discriminator objectives in adversarial training is pretty common. In terms of the theoretical results, Lemma 1 seems like a variation of a well-known result that behavioral cloning suffers from a compounding error phenomenon--this should probably be made clearer. I’m not really sure about the novelty of Lemma 2, but there is a very similar result in chapter 14 of this book: https://rltheorybook.github.io/rltheorybook_AJKS.pdf (sample complexity of DM-ST) as well as a similar result for GAIL here: https://arxiv.org/abs/2010.11876. Proposition 3 seems like a relatively straightforward application of the existing result for NCE. I did not really examine propositions 4 and 5 closely, so I can’t comment as to their novelty.

Clarity of presentation:
The presentation could use significant work. In particular, although I found Lemma 1 and Lemma 2 to be useful to build intuition, overall, sections 2.2 and 2.3 seem like a diversion from the main idea. I also thought that equations 3 and 5 were introduced in a slightly suspicious way, by implying that this is some sort of generalization of MaxEnt duality when the features are represented as nonlinear functions. I don’t think that’s really the case. So, I think more care should be taken to motivate these equations.

Also, as previously mentioned, the min/max swap in section 3 is also a bit misleading. Altogether, this gives the impression of a sloppy line of reasoning leading up to the main result. If I were to rewrite the paper, I would omit sections 2.2 and 2.3, and motivate equation 6 directly as a variational training objective, instead of motivating it through somewhat suspect analogies. I would then only swap the min and max to motivate the analysis, which shows that under some conditions, optimizing this objective leads to moment-matching and non-compounding error.


**Time Spent Reviewing:**

10

---

> ### Author Response · Authors · 2021-08-10
> **Response to Reviewer QAxC [Part 7/7]**
>
>
> ---
>
> ### **References**
>
> ---
>
> Numbering continued from bibliography in appendices:
>
> - [101] Ho & Ermon (2016). Generative adversarial imitation learning.
> - [102] Baram et al. (2017). End-to-end differentiable adversarial imitation learning.
> - [103] Jeon et al. (2018). A Bayesian approach to generative adversarial imitation learning.
> - [104] Arenz et al. (2020). Non-adversarial imitation and connections to adversarial methods.
> - [105] Ni et al. (2020). f-irl: Inverse reinforcement learning via state marginal matching.
> - [106] Ross et al. (2010). Efficient reductions for imitation learning.
> - [107] Agarwal et al. (2020). Reinforcement learning: theory and algorithms.
> - [108] Xu et al. (2020). Error bounds of imitating policies and environments.

---

> ### Author Response · Authors · 2021-08-10
> **Response to Reviewer QAxC [Part 6/7]**
>
> ---
>
> **Lemma 2 Premise**
>
> Thank you for pointing out that additional clarity is needed. First, you are correct that if $f$ is unbounded, the expression must equal $0$ or $\infty$. Second, you are also correct that in practice we assume---as usual---that $f$ is bounded. (We currently do so in Line 99).
>
> Actually, we are making a much simpler point here: While at the global min-max optimum---regardless whether $f$ is bounded---the discrepancy must be zero (i.e. equality constraints enforced), *in practice* there may be a variety of reasons why this global optimum is not perfectly achieved (e.g. error in estimating expectations, error in function approximation, error in optimization, etc). The main point of Lemma 2 is that, *whatever the reason* we end up being $\epsilon$ away from eliminating the moment-matching discrepancy, the quality difference is guaranteed to be at most linear in the horizon.
>
> ***Update***: We have now included a note in Lemmas 1 and 2 to ensure that this distinction is made clear.
>
> **Swapping Min/Max**
>
> You are correct. In (nonlinear) parameter space the optimization is not concave-convex, and we have now revised Lines 163--164 to correct for this. In the first instance, the motivation leading up to Section 3 (i.e. Lemma 1, Lemma 2, and Eq. 6) begins with the *linear* case, and then proceeds to the *non-linear* case. Like moment-matching (clarified above), the applicability of the minimax theorem is also not generalizable to the nonlinear case without a level of technical care (viz. convexity, compactness, finitude, etc.) beyond the scope of this work. We are simply doing this as is common in adversarial / variational learning algorithms in the literature (esp. in generative modeling and imitation), for which their "dual" forms are generally known to work well in practice.
>
> ***Update***: We have now revised the introduction of Eq. 6 to better clarify this.
>
> **GAIL paper**
>
> We agree that citing this would be beneficial.
>
> ***Update***: This is now already done in the (new) *Appendix D*, viz. response section (B) above.
>
> ---
>
> ### **(D) Experiments**
>
> ---
>
> **Results vs.GAIL/AIRL**
>
> Kindly refer to the new results tables re: response section (B) above.
>
> **Predictive score**
>
> Actually, in synthetic time-series generation the *train-on-synthetic, test-on-real* framework is the standard. As noted in the Experiments section, it is employed by most recent work in synthetic time-series generation [12, 14, 21, 22, 26, 97], as well as for tabular synthetic data of any kind [95, 96, 98]. We also use the original source code for computing these metrics. As an aside, using data likelihood is problematic in this space where the most typically competitive methods are GAN-based (e.g. C-RNN-GAN, COT-GAN, RC-GAN, TimeGAN), so effectively we would not be able to compare against any of them. That said, we do agree that an ongoing issue in generative modeling (esp. in time-series generation) is in the evaluation metric. We already discuss this in Sections 5 and 6, and we once again reiterate that this should be investigated more deeply in future work.

---

> ### Author Response · Authors · 2021-08-10
> **Response to Reviewer QAxC [Part 5/7]**
>
> ---
>
> ### **(C) Presentation**
>
> ---
>
> **Lemma 1 Related Work**
>
> You are correct, Lemma 1 is similar in spirit to results for error accumulation in behavioral cloning. The most well-known one is [106], where a quadratic bound is given with respect to the *probability* that the learned policy makes a small mistake. Another well-known one is in [107], where the bound is given with respect to *sample complexity*. In contrast, in order to motivate our algorithm from the perspective from moment-matching, here our bound is given with respect to the *moment-matching* discrepancy.
>
> ***Update***: We have now expanded the remarks in Appendix A (i.e. accompanying the proof for Lemma 1), referencing these variations on the horizon bound, to give better context.
>
> **Lemma 2 Related Work**
>
> You are correct, there exist other (different) results for error accumulation using distribution-matching methods. Like you point out, [108] does so in terms of *divergences* in occupancy measures, while [107] does so in terms of *sample complexity*. In contrast, similar to Lemma 1 above, Lemma 2 serves to motivate our algorithm from the perspective from moment-matching, so our bound is given with respect to the *moment-matching* discrepancy.
>
> Kindly allow us to reiterate that the primary novelty here is not in the linear/quadratic forms of the bounds themselves. Rather, the novelty is that within the realm of *time-series generation*, we specifically use *moment-matching* to arrive at a more intuitive and principled algorithm---unlike any of the existing time-series generative works that tackle free-running generation (in Table 1), almost all of which are motivated purely heuristically.
>
> ***Update***: We have now expanded the remarks in Appendix A (i.e. accompanying the proof for Lemma 2), referencing these related distribution-matching results, to give better context.
>
> **Logic of Eq. 3 and 5**
>
> Thank you for pointing out that additional clarity is needed. You are correct that Lemmas 1 and 2 are for building intuition, and we are *not* implying that Eq. 3 and 5 are direct generalizations of MaxEnt duality. We shall explain Eq. 5 in more detail; Eq. 3 is similar. Consider first the case of finite $\mathcal{X}$ (hence finite $\mathcal{T}$). In the usual linear case, given basis functions $T(\tau)$, the optimization problem is:
>
> $\arg\min_{\theta}E_{\tau\sim p_{\theta}}\log p_{\theta}(\tau)$ s.t. $E_{\tau\sim p_{s}}T(\tau)=E_{\tau\sim p_{\theta}}T(\tau)$
>
> Internalizing the constraint, we write $\textstyle\arg\min_{\theta}(E_{\tau\sim p_{\theta}}\log p_{\theta}(\tau)+\max_{F}(\langle F,E_{\tau\sim p_{s}}T(\tau)\rangle-\langle F,E_{\tau\sim p_{\theta}}T(\tau)\rangle))$. To generalize to the nonlinear case, let us now define $T(\tau)$ to be the indicator function (i.e. a finite-length vector whose zero-one entries correspond to the elements in $\mathcal{T}$). Then note that $\langle F,E_{\tau\sim p}T(\tau)\rangle=\langle F,p\rangle=E_{\tau\sim p}F(\tau)$, so we write:
>
> $$
> \textstyle\arg\min_{\theta}(E_{\tau\sim p_{\theta}}\log p_{\theta}(\tau)+\max_{F}(E_{\tau\sim p_{s}}F(\tau)-E_{\tau\sim p_{\theta}}F(\tau)))
> $$
>
> Finally, to arrive at Eq. 5 we use the fact that $E_{\tau\sim p}\sum_{t}f(h_{t},x_{t})=T E_{h\sim\mu,x\sim\pi(\cdot|h)}f(h,x)$, as already noted. All in all, this is a "generalization" from the case of linearity in *known* basis functions, to the case of *unknown* basis functions (where we "linearize" the expression using indicator functions). However, we cannot use the same logic to claim the same for *infinite* $\mathcal{X}$, which is what we ultimately use it for. So again, you are correct and we are not implying any generalization of MaxEnt duality.
>
> ***Update***: We have now expanded the text around Eq. 3 and 5 to ensure that this logic is made clear, so that we no longer erroneously give the impression that this is a generalization of duality.

---

> ### Author Response · Authors · 2021-08-10
> **Response to Reviewer QAxC [Part 4/7]**
>
> ---
>
> **For (2)**: Here, the discriminator is first constructed to be *transition*-based:
>
> $$
> d_{\theta,\phi}(z,u)=\tfrac{\exp(g_{\phi}(z,u))}{\exp(g_{\phi}(z,u))+\pi_{\theta}(u|z)}
> $$
>
> Note that this actually breaks the relationship with modeling the distribution of *trajectories* directly $p_{\phi}(\tau)=\exp(F_{\phi}(\tau)-\log Z_{\phi})$. Importantly, if we apply AIRL to TG, we lose the convergence property that comes from the relationship with NCE (cf. Prop. 4), and the gradient of the discriminator is no longer related to the original trajectory-wise energy-based gradient (cf. Prop. 5). In fact, all we are left with is that the global optimum is a correct optimum---but this by itself is not helpful (e.g. the global optimum of naive MLE is also a correct optimum, but there is no guarantee that it can be found effectively). In fact, it has been formally shown that the original theoretical justifications for AIRL are incorrect (see e.g. [104] Sec. 2.4.2, or [105] Sec. B.2).
>
> (In stochastic domains in IL, it is true that the empirical benefit of variance-reduction from sampling in this transition-based manner still dominates. In TG, however, there is no stochasticity in the "environment", and doing it this way may unnecessarily bias our objective of learning $p(\tau)$).
>
> We agree that to be thorough, we may add AIRL for experimental comparison (to be consistent, we also use SAC as the policy optimization method). This is now done (TimeGCI results reproduced here for comparison):
>
> | Method | Metric | Sines | Energy | Gas | Metro | MIMIC-III |
> |:--|:--|:--|:--|:--|:--|:--|
> |AIRL-SAC|Predictive Score|0.223 ± 0.006|0.283 ± 0.002|0.0478 ± 0.0015|0.243 ± 0.001|0.018 ± 0.005|
> |AIRL-SAC|+3 Steps Ahead  |0.381 ± 0.002|0.295 ± 0.003|0.0883 ± 0.0029|0.250 ± 0.001|0.019 ± 0.003|
> |AIRL-SAC|+5 Steps Ahead  |0.349 ± 0.005|0.321 ± 0.001|0.1176 ± 0.0040|0.251 ± 0.002|0.019 ± 0.001|
> |AIRL-SAC|x-Corr. Score   |10.397 ± 0.005|202.61  ± 0.119|144.79 ± 0.3199|1.286 ± 0.098|2635.57 ± 0.050|
> |TimeGCI|Predictive Score|0.097 ± 0.001|0.251 ± 0.001|0.018 ± 0.000|0.239 ± 0.001|0.002 ± 0.000|
> |TimeGCI|+3 Steps Ahead  |0.104 ± 0.001|0.251 ± 0.001|0.042 ± 0.001|0.239 ± 0.001|0.001 ± 0.000|
> |TimeGCI|+5 Steps Ahead  |0.109 ± 0.001|0.251 ± 0.001|0.067 ± 0.001|0.239 ± 0.001|0.001 ± 0.000|
> |TimeGCI|x-Corr. Score   |1.195 ± 0.011|105.2 ± 0.433|47.91 ± 0.811|0.738 ± 0.019|194.3 ± 0.180|
>
> ***Update***: As above, we have now included an extended version of this discussion in the (new) *Appendix D*. Moreover, the above empirical results are now also included in a (new) *Table 3(b)*.

---

> ### Author Response · Authors · 2021-08-10
> **Response to Reviewer QAxC [Part 3/7]**
>
> ---
>
> ### **(B) Adversarial Imitation Learning applied to TG**
>
> **TL;DR**: Adversarial IL methods work at the transition level, and are not directly optimizing the trajectory-based objective of TG. For completeness, we have now added new empirical results.
>
> ---
>
> Conversely to part (A) above, we can also ask: Can adversarial IL methods work for TG (hence can we compare them against TimeGCI)? The answer is "yes", they can be *applied* to TG, but "no", there is no reason to expect they would perform better.
>
> Modern adversarial IL methods come in two broad flavors: (1) GAIL-based [101, 102, 103, etc.], which do not recover a reward function, and (2) AIRL-based [64, 65, 104, etc.], which do recover a reward function. Both are transition-centric (i.e. scoring $(z,u)$ pairs).
>
> **For (1)**: Here, optimization is of the familiar GAN-like objective ("\mathbb" fails to render correctly here, so "$E$" denotes expectation):
>
> $$
> \textstyle\min_{\theta}\max_{\phi}E_{z,u\sim \mu_{s}}\log d_{\phi}(z,u)+E_{z,u\sim \mu_{\theta}}\log\big(1-d_{\phi}(z,u)\big)
> $$
>
> This minimizes the JS-divergence between the *state-action* occupancy measures (i.e. distribution of transitions, in TG terms) induced by the learned and source policies. Note that this is *the same* as the TimeGAN objective [12], modulo entropic/supervised regularization. As noted in the manuscript, matching the transition marginals is indirectly performing the "global" sort of moment-matching. However, as pertains TG, the key difference between GAIL and TimeGAN (which we already compared) is that the former performs policy optimization for the generator, whereas the latter simply performs backpropagation. Otherwise, note that both methods are adversarial (viz. saddle-point optimization) and do not learn an explicit energy (viz. black-box discriminator).
>
> We agree that to be thorough, we may add GAIL for experimental comparison (to be consistent, we also use SAC as the policy optimization method). This is now done (TimeGCI results reproduced here for comparison):
>
> | Method | Metric | Sines | Energy | Gas | Metro | MIMIC-III |
> |:--|:--|:--|:--|:--|:--|:--|
> |GAIL-SAC|Predictive Score|0.447 ± 0.002|0.261 ± 0.001|0.022 ± 0.002|0.257 ± 0.001|0.017 ± 0.001|
> |GAIL-SAC|+3 Steps Ahead  |0.472 ± 0.003|0.262 ± 0.001|0.048 ± 0.001|0.250 ± 0.001|0.012 ± 0.001|
> |GAIL-SAC|+5 Steps Ahead  |0.542 ± 0.004|0.261 ± 0.001|0.074 ± 0.002|0.245 ± 0.003|0.011 ± 0.001|
> |GAIL-SAC|x-Corr. Score   |6.798 ± 0.014|142.457 ± 0.471|111.124 ± 0.232|1.044 ± 0.031|540.89 ± 0.159|
> |TimeGCI|Predictive Score|0.097 ± 0.001|0.251 ± 0.001|0.018 ± 0.000|0.239 ± 0.001|0.002 ± 0.000|
> |TimeGCI|+3 Steps Ahead  |0.104 ± 0.001|0.251 ± 0.001|0.042 ± 0.001|0.239 ± 0.001|0.001 ± 0.000|
> |TimeGCI|+5 Steps Ahead  |0.109 ± 0.001|0.251 ± 0.001|0.067 ± 0.001|0.239 ± 0.001|0.001 ± 0.000|
> |TimeGCI|x-Corr. Score   |1.195 ± 0.011|105.2 ± 0.433|47.91 ± 0.811|0.738 ± 0.019|194.3 ± 0.180|
>
> ***Update***: Like response (A), we have now included an extended version of this discussion in the (new) *Appendix D*. Moreover, the above empirical results are now also included in a (new) *Table 3(a)*.

---

> ### Author Response · Authors · 2021-08-10
> **Response to Reviewer QAxC [Part 2/7]**
>
> ---
>
> ### **(A) TimeGCI is not suitable for IL**
>
> **TL;DR**: We do not evaluate TimeGCI on IL scenarios, because it is designed for TG, not IL---and would not work well for IL. Adapting TimeGCI for IL is beyond the scope of this work.
>
> ---
>
> At its core, TG is the problem of modeling a *distribution of trajectories* $p(\tau)$ faithfully, and is what TimeGCI does. In MDP parlance, in TG the "environment dynamics" are by construction *known* and *deterministic*: $\omega(\cdot|z_{t},u_{t})$ is the Dirac delta centered at $z_{t+1}=(u_{1},…,u_{t})$. We have $p(\tau)=\prod_{t}\pi(u_{t}|z_{t})$. However, in IL the dynamics are generally *unknown* and *stochastic*. We have $p(\tau)=\prod_{t}\pi(u_{t}|z_{t})\omega(z_{t}|z_{t-1},u_{t-1})$.
>
> This difference is crucial. Consider walking through the logic of Section 3, but applying it to IL instead. Begin with the gradient update for learning the Gibbs parameters (Eq. 7). To avoid costly inner-loop optimization of $\theta$ to completion, we again consider (1) importance sampling, as well as our preferred method of (2) contrastive estimation. We will see that in IL neither of these work out of the box. For (1), the importance weights now become $\tfrac{\exp(F_{\phi}(\tau))}{p_{\theta}(\tau)}$, where $p_{\theta}(\tau)=\prod_{t}\pi_{\theta}(u_{t}|z_{t})\omega(z_{t}|z_{t-1},u_{t-1})$. But the problem is that we do not have access to $\omega$. We can naively *estimate* it beforehand (clearly inadvisable). Or, we can perform a specific *approximation* in the energy model:
>
> $$
> p_{\phi}(\tau)=\exp(F_{\phi}(\tau)-\log Z_{\phi})\approx\tfrac{1}{Z_{\phi,\omega}}\exp(F_{\phi}(\tau))\prod_{t}\omega(z_{t}|z_{t-1},u_{t-1})
> $$
>
> such that $\omega$ cancels out from the expression. Instead of modeling the distribution of trajectories $p(\tau)$ exactly, this now assumes that transition randomness has a limited effect on behavior and that the partition function is constant for all random outcome samples (see e.g. [61]). The situation is similar for (2), where it is easy to see that in IL scenarios Eq. 10 is no longer accessible without performing the above approximation to cancel out $\omega$.
>
> Three points bear emphasis. First, if we are willing to make this simplification, *we are no longer speaking of TimeGCI anymore*. Instead, it is easy to show that this effectively becomes a sort of trajectory-centric "maximum *causal* entropy" inverse optimal control (see e.g. [39]), which is *no longer* learning $p(\tau)$ with an exact objective. Second, in general we have little reason to believe that such a trajectory-centric approach would work well in IL: The *variance* of sample-based estimates is now exacerbated by the unknown and stochastic environment dynamics in IL, such that trajectory-based sampling is generally known to perform poorly (see e.g. [64]). In fact, for this reason modern adversarial IL methods almost always take a transition-centric approach. Third, *empirically* a similar structured-classifier approach (with the above approximation) has indeed been found to be unworkable in practice due to high variance (see [64] Sec. 4, implementing [63]); moreover, an importance-sampling based approach (with the above approximation) has been made to work, but has required hand-crafted, domain-specific regularization to work, and the learned energy functions only explain the demonstrations locally [62].
>
> For these reasons, we do not evaluate TimeGCI on IL scenarios. It is simply not applicable without modifying it to become a different method altogether. Modern IL methods are designed specifically to handle the unknown and stochastic nature of environment transitions. In particular, *transition-centric* approaches (i.e. scoring $(z,u)$ pairs) have been found to be more effective empirically. What we are doing with TimeGCI, on the other hand, is to focus on an exact objective for learning time-series trajectories $p(\tau)$, which requires a *trajectory-centric* approach. Because, this allows us to equate TimeGCI with a special kind of noise-contrastive estimation, for which we can enjoy some theoretical guarantees. (These properties are lost when performing the above approximation in an IL setting).
>
> ***Update***: We have now included an extended version of this discussion in the (new) *Appendix D*, which clarifies the similarities and differences between the objectives and methods for time-series generation and imitation learning.

---

> ### Author Response · Authors · 2021-08-10
> **Response to Reviewer QAxC [Part 1/7]**
>
> ---
>
> Thank you for your thoughtful comments and suggestions. We give answers to each of the following in turn, as well as pointing out corresponding updates to the revised manuscript:
>
> - (A) No evaluation on imitation learning scenarios
> - (B) No experimental comparison to adversarial imitation learning methods
> - (C) Presentation
> - (D) Experiments
>
> ---
>
> By way of preface, kindly allow us to reiterate that we are expressly focused on the problem of **synthetic time-series generation**. In doing so, we propose a method (TimeGCI) that invites an *analogy* between time-series generation and imitation learning---but they also have crucial differences. As we shall elaborate, (A) TimeGCI is *not* suitable for imitation learning per se (thus we make no claim that it is). Conversely, (B) while existing adversarial imitation learning methods *may* be naively applied to time-series generation, their optimization objectives are different; we empirically verify they do not perform quite as well.
>
> We agree that the terseness of the Discussion section may have left this distinction unclear, so we have now included the following clarifying points below in a (new) *Appendix D*, which is now referenced in Section 4.
>
> *Notation*: In the following, we abbreviate our problem of time-series generation "TG", and the problem of imitation learning "IL". For consistency, here we use the MDP notation of Section 4 for IL, i.e. $z_{t}$ for states and $u_{t}$ for actions. For consistency, for TG we shall use the same, with the implicit understanding that $z_{t}:=h_{t}$, and $u_{t}:=x_{t}$ (viz. Corollary 6).

---

> ### Author Response · Authors · 2021-08-20
> **Dear Reviewer QAxC**
>
> ---
>
> We are sincerely grateful for your time and energy in the review process.
>
> In light of our responses (Aug 10) and appendix/manuscript updates (Aug 19), we would appreciate if the reviewer kindly let us know of any leftover concerns in the time remaining. We would be happy to do our utmost to address them.
>
> Thank you!
>
> Paper10659 Authors

---

> ### Author Response · Authors · 2021-08-27
> **RE: Dear Reviewer QAxC**
>
> Thank you again for your time and expertise during the review process!
>
> If there were any leftover concerns, we would sincerely appreciate the opportunity to clarify them---before the discussion period for authors ends. We believe our responses (Aug 10) have addressed in detail the full set of questions you had raised, along with corresponding appendix/manuscript updates (Aug 19):
>
> - No evaluation on imitation learning scenarios; see Response (A)
> - No experimental comparison to adversarial imitation learning methods; see Response (B)
> - Presentation; see Response (C)
> - Experiments; see Response (D)
>
> We would appreciate if the reviewer kindly let us know if there were any further questions in the very limited time remaining. We are eager to do our utmost to address them!
>
> Thank you,
>
> Paper10659 Authors

---

> > ### Comment · Reviewer_QAxC · 2021-09-02
> > **Post-rebuttal response**
> >
> > I apologize for the delayed response. Thanks for the detailed notes. After reading the rebuttal, I am willing to upgrade my assessment on account of the following factors:
> > 1. I now believe that the proposed method does fill a slightly different niche than most AIL methods.
> > 2. The new experimental comparison to GAIL addresses a significant shortcoming in the original results
> > 3. The various small updates to the exposition will hopefully make the presentation clearer
> >
> > Regarding the issue of whether TimeGCI should be considered a "generic" IL method or not: I did originally overlook the point brought up in the rebuttal, which is that a model of dynamics would be necessary in order to apply eq 10, whereas most IL methods are "model free", in the sense that they do explicitly evaluate a dynamics model. I have not run through all the details, but I believe that TimeGCI could still probably be applied as a model-based IL method, akin to MaxEnt IRL. Note that although the model-free approach has advantages, it is by no means clear that model-free methods are more widely applicable or overall superior compared to model-based methods. So, I believe that viewing TimeGCI as a model-based IL method may be an interesting avenue to explore in future work.

---

### Official Review · Reviewer_YqJQ · 2021-08-11

**Rating:** 6
**Confidence:** 4

**Summary:**

The paper proposes to perform time-series generation via a method inspired by imitation learning in a non-adversarial manner. The idea is to fit an energy-based model over the trajectories with noise contrastive estimation, with the noise being provided by a parametrized policy. The policy is also optimized to fit the energy-based model to provide a sound noise distribution. The method demonstrates superior empirical performance on various time-series datasets, on prediction error, and cross-correlation.

**Limitations And Societal Impact:**

Yes, the authors have adequately addressed the limitations and potential negative social impact of their work.

**Main Review:**

**Update**
I think the ablation studies address my concerns. More robustness of TImeGCI is shown, and may be a reason why we observe better performance (despite having the same "amount" of supervision). I am willing to raise the score because of this.

**Originality and significance**

While the paper discusses the relationship between time-series prediction and imitation learning, it seems that imitation learning methods are not given enough credit. For example, existing works in imitation learning [1,2] have already proposed methods to learn an energy-based model from the trajectories and then fir the policy to the energy function.

[1] Kim, K., Jindal, A., Song, Y., Song, J., Sui, Y. and Ermon, S., 2020. Imitation with Neural Density Models. arXiv preprint arXiv:2010.09808.
[2] Arenz, O. and Neumann, G., 2020. Non-Adversarial Imitation Learning and its Connections to Adversarial Methods. arXiv preprint arXiv:2008.03525.

Given these prior works, the novelty of the proposed method would take a hit, as time-series prediction problems can be posed as an imitation learning problem with the "states" in the MDP being the "history" of the previous timesteps, and "action" in the MDP transitions to the next "state" concatenated with "history".

Even in the context of time series, the application of NCE is not entirely novel, such as https://www.cs.toronto.edu/~amnih/papers/ncelm.pdf

**Clarity**

**Explanation of empirical success** Another question that I have for the paper is why the method performs better and whether the success of "trajectory-level" imitation learning methods can be replicated here; one reason that GAIL-like methods outperform behavior cloning is that the RL algorithm can make use of external signals (e.g., "done") given by the environment. I have personally done experiments with GAIL and behavior cloning before and found it hard to outperform the latter without such information. The proposed method here does not appear to use any additional information than T-forcing or P-forcing, so the advantages must come from the loss function. However, there are no other experiments/ablation studies that show why the proposed method is better -- is it because of better first-step prediction, or is it because of better robustness when having small errors in the previous steps. I believe it is necessary to include such artificial ablation studies to see why the method is better (I am not entirely convinced by the theory alone, see below).

**Explanation of non-adversarial training** The fact that the method is non-adversarial may seem confusing to some readers (especially if they are familar with GAN or GAIL), and I do not believe the current writing is clear enough to show this. It may seem easier for readers to understand why the method is non-adversarial through a variational interpretation: the goal is to minimize a divergence (seems in your case, reverse KL); adversarial methods minimize a variational lower bound (and thus have to be minimax optimized), whereas your case minimizes a variational upper bound (like the ELBO for log-likelihood).

**Quality**

Other than the aforementioned issues (insufficient credit to imitation learning methods, explanation of non-adversarial, etc.), the paper is written well.

**Minor issues**

While Lemma 2 appears to show that by minimizing the discrepancy at the trajectory level, one obtains an expected quality difference of O(T) instead of O(T^2), the $\epsilon$ value bounds a different quantity (global discrepancy for T steps in Lemma 2, as opposed to local discrepancy for 1 step in Lemma 1). Thus, I do not think Lemma 2 fully explains why trajectory-centric methods are better (since the assumption is also "stronger"). Also, Lemma 1 seems to be quite relevant to the DAgger paper, which I don't think you have cited.

The concept of expected quality difference is highly relevant to integral probability metrics. As Lemma 1 and 2 uses the max "expected quality difference", that is essentially an IPM; there is no need to invent another wheel.



**Time Spent Reviewing:**

4

---

> ### Author Response · Authors · 2021-08-15
> **Response to Reviewer YqJQ [Part 8/8]**
>
> ---
>
> ### References
>
> ---
>
> - [R1] Kim et al. (2020). Imitation with neural density models. arXiv preprint arXiv:2010.09808.
> - [R2] Arenz et al. (2020). Non-adversarial imitation learning and its connections to adversarial methods. arXiv preprint arXiv:2008.03525.
> - [R3] Ke et al. (2020) Imitation Learning as f-Divergence Minimization. International Workshop on the Algorithmic Foundations of Robotics (WAFR).
> - [R4] Mnih et al. (2012). A fast and simple algorithm for training neural probabilistic language models. International Conference on Machine Learning (ICML).
> - [R5] Ho & Ermon (2016). Generative adversarial imitation learning. Advances in Neural Information Processing Systems (NIPS).
> - [R6] Baram et al. (2017). End-to-end differentiable adversarial imitation learning. International Conference on Machine Learning (ICML).
> - [R7] Jeon et al. (2018). A Bayesian approach to generative adversarial imitation learning. Advances in Neural Information Processing Systems (NIPS).
> - [R8] Agarwal et al. (2020). Reinforcement learning: theory and algorithms.

---

> ### Author Response · Authors · 2021-08-15
> **Response to Reviewer YqJQ [Part 7/8]**
>
> ---
>
> ### (C) Minor Issues
>
> ---
>
> **(C.1) "The $\epsilon$ bounds a different quantity"**:
>
> There are three distinct quantities here:
>
> - (a) What is $\epsilon$?
> - (b) What does bounding $\epsilon$ accomplish?
> - (c) How well can we practically minimize $\epsilon$?
>
> First, let us look at (a) and (b). In Lemma 1:
>
> - (a) $\epsilon$ is the "local" discrepancy (i.e. both $\pi_{s}$ and $\pi_{\theta}$ receive **external data** $h\sim\mu_{s}$).
> - (b) Bounding $\epsilon$ gives an $O(T^{2}\epsilon)$ result.
>
> In Lemma 2:
>
> - (a) $\epsilon$ is the "global" discrepancy (i.e. $\pi_{\theta}$ receives **its own rollout histories** $h\sim\mu_{\theta}$).
> - (b) Bounding $\epsilon$ gives an $O(T\epsilon)$ result.
>
> So, you are correct that $\epsilon$ bounds a different quantity in Lemmas 1 and 2. However *that is the whole point*. Precisely, the learning objective of conditional MLE is related to minimizing the local discrepancy; here, Lemma 1 applies. (Importantly, there is some information that is discarded here, namely the fact that certain groups of transitions come from the same trajectory). The point of Lemma 2 is to say that ***if*** we could instead bound the global discrepancy (which would use more information than before, i.e. the fact that certain groups of transitions come from the same trajectory), ***then*** we could do better. We then propose a practical algorithm with this in mind.
>
> Now, the last piece is---of course---(c): practically, how well we can actually minimize those quantities. That is a valid question, and which is why empirical experiments are done. Also, the manuscript already explicitly notes that while we are guided by the notion of global moment-matching, in practice there is no guarantee that this is accomplished well during optimization (Section 6, Lines 370-371). *Importantly, however, this is the same tradeoff faced by the vast majority of machine learning methods*: A better objective gives a better bound, BUT we are always going to be reliant on how well practical optimization of that objective can be performed.
>
> **(C.2) Lemma 1 related work**:
>
> You are correct, Lemma 1 is similar in spirit to results for error accumulation in behavioral cloning. The most well-known one is from DAgger [54/57] as you point out, where a quadratic bound is given with respect to the *probability* that the learned policy makes a small mistake. (Note that we actually already cite these, just not next to the Lemma). Another example is in [R8], where the bound is given with respect to *sample complexity*. In contrast, in order to motivate our algorithm from the perspective from moment-matching, here our bound is given with respect to the *moment-matching* discrepancy.
>
> ***Update***: We have now expanded the remarks in Appendix A (i.e. accompanying the proof for Lemma 1), referencing these variations on the horizon bound, to give better context.
>
> **(C.3) Integral probability metrics**:
>
> You are correct. We agree that our "expected quality difference" is related to IPMs. We originally did not draw this connection, simply because we do not use any IPM-specific ideas or algorithms in the paper. However, we agree that it is good to point this out to be clear.
>
> ***Update***: On Lines 130 and 152 (Section 2) in the original manuscript, we have now added references to a (new) footnote, which explains precisely the connection with IPMs.

---

> ### Author Response · Authors · 2021-08-15
> **Response to Reviewer YqJQ [Part 6/8]**
>
> ---
>
> **(B.3) External "done" signals from the environment**:
>
> You are correct. We do not use any form of "done" signal. But these only exist in the context of reinforcement learning environments, which is not what we have.
>
> In synthetic time-series generation, there is no "environment", therefore there is no "done" signal.
>
> **(B.4) BC vs. GAIL**
>
> In the context of IL, we agree that---empirically---in the literature it appears that BC often performs comparably to GAIL when using similar information. However, that is a question about *comparing two IL methods*, which is beyond the scope of our work.
>
> To be clear, while BC is analogous to T-Forcing, *GAIL is not analogous to TimeGCI at all* (please kindly refer to our response to (B.2) above). If anything, GAIL is most analogous to TimeGAN modulo entropic/supervised regularization (viz. Eq. 8 in [12] compared to Eq. 16 in [R5]).
>
> And as already mentioned in our response to (B.2), this matches transition marginals, which is only indirectly performing the "global" sort of moment-matching. Not to mention it is adversarial, which TimeGCI is not (this point is actually important, as we do empirically observe several instances of mode collapse in our GAN-based benchmarks; this is noted on Line 364. This is visible in our results, given that a key strength of TSTR evaluation is its sensitivity to mode collapse).
>
> **(B.5) Further ablation study**
>
> We agree that ablation study is useful to provide sensitivities for understanding.
>
> Please find the following ablation study that sheds light on the performance advantage. We compare the performance of T-Forcing to TimeGCI, using sequence lengths $T\in\{2, 8, 24\}$. We compute the TSTR predictive score, +3 steps ahead, +5 steps ahead, and $x$-Corr. score as usual for each of these settings. (Note that for $T=2$ we cannot ask the TSTR evaluation model to predict more than one step ahead, since there is no data for more than one step ahead; we indicate this with "N/A"). The Gas dataset is used. For ease of interpretation, we consider T-Forcing as the "baseline", and also express the TimeGCI numbers as a fraction of the T-Forcing numbers:
>
> | Method | Metric | $T=2$ | $T=8$ | $T=24$ |
> |:--|:--|:--|:--|:--|
> | T-Forcing     |TSTR Predictive|0.038 ± 0.001|0.036 ± 0.007|0.035 ± 0.003|
> | T-Forcing     |+3 Steps Ahead |N/A|0.076 ± 0.007|0.080 ± 0.001|
> | T-Forcing     |+5 Steps Ahead |N/A|0.114 ± 0.003|0.111 ± 0.001|
> | T-Forcing     |x-Corr. Score  |160.3 ± 0.346|154.2 ± 0.219|150.8 ± 0.067|
> | TimeGCI       |TSTR Predictive|0.037 ± 0.001|0.022 ± 0.000|0.018 ± 0.000|
> | TimeGCI       |+3 Steps Ahead |N/A|0.048 ± 0.001|0.042 ± 0.001|
> | TimeGCI       |+5 Steps Ahead |N/A|0.071 ± 0.001|0.067 ± 0.001|
> | TimeGCI       |x-Corr. Score  |140.4 ± 0.173|50.72 ± 0.816|47.91 ± 0.811|
> | Ratio of Mean |TSTR Predictive|97.37%|61.11%|51.43%|
> | Ratio of Mean |+3 Steps Ahead |N/A|63.16%|52.50%|
> | Ratio of Mean |+5 Steps Ahead |N/A|62.28%|60.36%|
> | Ratio of Mean |x-Corr. Score  |87.59%|32.89%|31.77%|
>
> This is consistent with what we expect. Namely, as the training sequence lengths increase in magnitude, the more TimeGCI appears to have an advantage with respect to T-Forcing. This shows that T-Forcing appears to preserve quite well the one-step ahead relationships of the original data (viz. $T=2$). However, for longer sequences TimeGCI performs relatively better, which is consistent with the motivation behind using an objective that matches the distribution of trajectories, instead of simply matching transition conditionals.
>
> ***Update***: We have now included these results in the (new) *Appendix D*, in a (new) *Table 4*, along with the above remarks on interpreting the results and its consistency with our expectations.
>
> **(B.6) Explanation of "non-adversarial"**
>
> Thank you for the suggestion. We completely agree that the "variational" interpretation you propose is very clear indeed, and may be easier to understand for readers, esp. those coming from GAN/GAIL).
>
> ***Update***: On Line 59 (Section 1), we have now added a (new) footnote to the bullet point, next to the word "non-adversarially", which explains precisely the distinction you suggested. The footnote reads as follows:
>
> - Clarification on "adversarial-ness": When minimizing a divergence, adversarial methods minimize a variational lower bound (and thus have to be minimax optimized), whereas---as we shall see---our case ends up minimizing a variational upper bound (analogous to the ELBO for log-likelihood).

---

> ### Author Response · Authors · 2021-08-15
> **Response to Reviewer YqJQ [Part 5/8]**
>
> ---
>
> ### (B) Clarity
>
> ---
>
> **(B.1) "Why the method performs better"**:
>
> Thank you for the question. To be clear, we take this to mean why TimeGCI performs better than existing synthetic time-series generation methods".
>
> Note that Sections 1, 2, and 3 of the manuscript already carefully walk through the rationale of why existing methods may not do well, how we designed TimeGCI, as well as why we expect it to do better. Specifically:
>
> - Section 1 (Lines 34-50) gives an overview of the objectives of existing work. Note that the vast majority of time-series generation methods---including those that specifically recognize the challenge of open-loop sampling---are motivated heuristically as variants of the standard GAN framework (see Table 1).
>
> - Section 2 (Lemmas 1 and 2) builds intuition for why methods that focus on learning conditionals---such as any conditional maximum likelihood-based method---may not do well, as well as why a method that focuses on matching the distribution of entire trajectories---which we use to motivate TimeGCI---may do better.
>
> - Section 3 (Propositions 3, 4, and 5) builds intuition for why the actual proposed algorithm using noise-contrastive estimation is expected to behave (i.e. optimality, convergence, etc).
>
> All of these are good reasons, in our view, of why we may expect TimeGCI to do better than other synthetic time-series generation methods.
>
> **(B.2) "Whether the success of "trajectory-level" imitation learning methods can be replicated here"**:
>
> Thank you for the question. We agree that whether IL methods can be "plugged in" to synthetic time-series generation is a valid one, esp. given the analogy that is Corollary 6.
>
> However, note that modern IL methods typically do not operate on the basis of expectations over entire trajectories. That is one reason we might not expect "plugging in" IL methods here would necessarily do as well, in addition to the second reason that these methods are typically adversarial. Modern IL methods come in two broad flavors: (1) GAIL-based [R5, R6, R7, etc.], which do not recover a reward function, and (2) AIRL-based [64, 65, R2, etc.], which do recover a reward function. Both are transition-centric (i.e. scoring $(z,u)$ pairs).
>
> For (1): Here, optimization is of the familiar GAN-like objective ("\mathbb" fails to render correctly here, so "$E$" denotes expectation):
>
> $$
> \textstyle\min_{\theta}\max_{\phi}E_{z,u\sim \mu_{s}}\log d_{\phi}(z,u)+E_{z,u\sim \mu_{\theta}}\log\big(1-d_{\phi}(z,u)\big)
> $$
>
> This minimizes the JS-divergence between the *state-action* occupancy measures (i.e. distribution of transitions, in our terminology for time-series generation) induced by the learned and source policies. As noted in the manuscript, matching the transition marginals is only indirectly performing the "global" sort of moment-matching. Note that this method is adversarial (viz. saddle-point optimization) and do not learn an explicit energy (viz. black-box discriminator).
>
> We agree that to be thorough, we may add GAIL for experimental comparison (to be consistent, we also use SAC as the policy optimization method). This is now done (TimeGCI results reproduced here for comparison):
>
> | Method | Metric | Sines | Energy | Gas | Metro | MIMIC-III |
> |:--|:--|:--|:--|:--|:--|:--|
> |GAIL-SAC|Predictive Score|0.447 ± 0.002|0.261 ± 0.001|0.022 ± 0.002|0.257 ± 0.001|0.017 ± 0.001|
> |GAIL-SAC|+3 Steps Ahead  |0.472 ± 0.003|0.262 ± 0.001|0.048 ± 0.001|0.250 ± 0.001|0.012 ± 0.001|
> |GAIL-SAC|+5 Steps Ahead  |0.542 ± 0.004|0.261 ± 0.001|0.074 ± 0.002|0.245 ± 0.003|0.011 ± 0.001|
> |GAIL-SAC|x-Corr. Score   |6.798 ± 0.014|142.457 ± 0.471|111.124 ± 0.232|1.044 ± 0.031|540.89 ± 0.159|
> |TimeGCI|Predictive Score|0.097 ± 0.001|0.251 ± 0.001|0.018 ± 0.000|0.239 ± 0.001|0.002 ± 0.000|
> |TimeGCI|+3 Steps Ahead  |0.104 ± 0.001|0.251 ± 0.001|0.042 ± 0.001|0.239 ± 0.001|0.001 ± 0.000|
> |TimeGCI|+5 Steps Ahead  |0.109 ± 0.001|0.251 ± 0.001|0.067 ± 0.001|0.239 ± 0.001|0.001 ± 0.000|
> |TimeGCI|x-Corr. Score   |1.195 ± 0.011|105.2 ± 0.433|47.91 ± 0.811|0.738 ± 0.019|194.3 ± 0.180|
>
> ***Update***: We have now included an extended version of this discussion in the (new) *Appendix D*. Moreover, the above empirical results are now also included in a (new) *Table 3(a)*.
>
> For (2): Here, the discriminator is first constructed to be *transition*-based:
>
> $$
> d_{\theta,\phi}(z,u)=\tfrac{\exp(g_{\phi}(z,u))}{\exp(g_{\phi}(z,u))+\pi_{\theta}(u|z)}
> $$
>
> Note that this actually breaks the relationship with modeling the distribution of *trajectories* directly $p_{\phi}(\tau)=\exp(F_{\phi}(\tau)-\log Z_{\phi})$. Importantly, if we apply AIRL to synthetic time-series generation, we lose the convergence property that comes from the relationship with NCE (cf. Prop. 4), and the gradient of the discriminator is no longer related to the original trajectory-wise energy-based gradient (cf. Prop. 5). In fact, all we are left with is that the global optimum is a correct optimum---but this by itself is not helpful (e.g. the global optimum of naive MLE is also a correct optimum, but there is no guarantee that it can be found effectively). In fact, it has been formally shown that the original theoretical justifications for AIRL are incorrect (see e.g. [104] Sec. 2.4.2, or [105] Sec. B.2).
>
> (In stochastic domains in IL, it is true that the empirical benefit of variance-reduction from sampling in this transition-based manner still dominates. In synthetic time-series generation, however, there is no stochasticity in the "environment", and doing it this way may unnecessarily bias our objective of learning $p(\tau)$).
>
> We agree that to be thorough, we may add AIRL for experimental comparison (to be consistent, we also use SAC as the policy optimization method). This is now done (TimeGCI results reproduced here for comparison):
>
> | Method | Metric | Sines | Energy | Gas | Metro | MIMIC-III |
> |:--|:--|:--|:--|:--|:--|:--|
> |AIRL-SAC|Predictive Score|0.223 ± 0.006|0.283 ± 0.002|0.0478 ± 0.0015|0.243 ± 0.001|0.018 ± 0.005|
> |AIRL-SAC|+3 Steps Ahead  |0.381 ± 0.002|0.295 ± 0.003|0.0883 ± 0.0029|0.250 ± 0.001|0.019 ± 0.003|
> |AIRL-SAC|+5 Steps Ahead  |0.349 ± 0.005|0.321 ± 0.001|0.1176 ± 0.0040|0.251 ± 0.002|0.019 ± 0.001|
> |AIRL-SAC|x-Corr. Score   |10.397 ± 0.005|202.61  ± 0.119|144.79 ± 0.3199|1.286 ± 0.098|2635.57 ± 0.050|
> |TimeGCI|Predictive Score|0.097 ± 0.001|0.251 ± 0.001|0.018 ± 0.000|0.239 ± 0.001|0.002 ± 0.000|
> |TimeGCI|+3 Steps Ahead  |0.104 ± 0.001|0.251 ± 0.001|0.042 ± 0.001|0.239 ± 0.001|0.001 ± 0.000|
> |TimeGCI|+5 Steps Ahead  |0.109 ± 0.001|0.251 ± 0.001|0.067 ± 0.001|0.239 ± 0.001|0.001 ± 0.000|
> |TimeGCI|x-Corr. Score   |1.195 ± 0.011|105.2 ± 0.433|47.91 ± 0.811|0.738 ± 0.019|194.3 ± 0.180|
>
> ***Update***: As above, we have now included an extended version of this discussion in the (new) *Appendix D*. Moreover, the above empirical results are now also included in a (new) *Table 3(b)*.
>
> Lastly, we of course recognize that there are many other variants of IL methods out there. But---at risk of repetition---we are focusing on synthetic time-series generation. We already propose a novel method and algorithm specifically tailored to this problem. Whether or not yet other methods in IL can be more cleverly "applied" or "adapted" to this problem is certainly an interesting question, and is precisely the perspective we wish to convey (i.e. to bridge the two communities). However, that is certainly future work.

---

> ### Author Response · Authors · 2021-08-15
> **Response to Reviewer YqJQ [Part 4/8]**
>
> ---
>
> **(A.3) "Given these prior works, the novelty of the proposed method would take a hit, as time-series prediction problems can be posed as an imitation learning problem"**:
>
> We politely disagree, on the following counts:
>
> (A.3.1) We are not doing time-series prediction:
>
> First, to get a nit out of the way: We are doing synthetic time-series *generation*. We are not doing time-series *prediction*. It is very important that there is no misunderstanding here! Prediction is a wholly different problem (from either synthetic time-series generation, or imitation learning). In synthetic time-series generation, we are "generating from scratch". In time-series prediction, we are given real data, and asked to predict ahead. We must be very clear that these are different. Whether or not *prediction* can be posed as an IL problem is an interesting problem, *but not what we are doing at all*. (Going forward, we will assume that you are referring to synthetic time-series generation being posable as a an IL problem).
>
> (A.3.2) The analogy itself is novel:
>
> To avoid repetition, please kindly see our above response titled "Preface on Novelty". There, we make it clear that (1) this analogy itself---of "generation as imitation" (Corollary 6)---is novel, as well as the method inspired by it; that (2) the concrete algorithm we propose is novel, including the simultaneous satisfaction of the three criteria, and the fact that we identify the equivalence with noise-contrastive estimation, which allows us to use theoretical guarantees; and that (3) the empirical evaluation and results are novel, with respect to prior literature on synthetic time-series generation / sequence generation. Please kindly refer to that response for the detailed discussion.
>
> **(A.4) "Even in the context of time series, the application of NCE is not entirely novel, such as [R4]"**:
>
> We respectfully point out that this is incorrect.
>
> - Before we begin, note that we already reference [R4] in the manuscript: It is simply [76].
>
> - First, we already explicitly discuss how [R4=76] is different in the manuscript: "conditional EBMs have
> been trained with NCE for text generation [76–78] ... these are confined to the case where *external input* tokens are available for conditioning at each step—--and not free-running as in our time-series setting" (Lines 298--301). Moreover, these generate *discrete* tokens for language modeling, not continuous tabular data as in our case.
>
> - Second, to be very clear: "a statistical language model is simply a collection of conditional distributions for the next word, indexed by its context" (first sentence in Section 2, from [R4=76]). In other words, these are *conditional* EBMs! They have no relationship at all to the *trajectory-wise* EBMs that we are considering in TimeGCI, because the role of NCE here is different in that it is simply helping to learn the conditional EBM. In contrast, for us the whole point is in doing trajectory-wise EBMs to attempt to mitigate error compounding.
>
> For these reasons, we respectfully disagree that [R4=76] is doing anything remotely similar.

---

> ### Author Response · Authors · 2021-08-15
> **Response to Reviewer YqJQ [Part 3/8]**
>
> ---
>
> ### (A) Originality and Significance
>
> ---
>
> **(A.1) "Imitation learning methods are not given enough credit"**:
>
> We politely disagree, on the following counts:
>
> (A.1.1) We reference very many:
>
> Out of the 100 references to related work, we include 16 specifically in imitation learning: [39], [54], [55], [56], [57], [58], [59], [60], [61], [62], [63], [64], [65], [66], [67], [100], including the most famous works of Ziebart's dissertation [39], Ross and Bagnell's DAgger and related [54, 57], the classic survey of imitation learning from Osa [55], Ng, Russell, and Abbeel's original IRL papers [59, 60], the first maximum entropy inverse reinforcement learning paper [61], the first paper on sample-based estimation by Finn [62], and the original adversarial inverse reinforcement learning papers [63, 64, 65] that set the groundwork for the multitude of variants. In a paper specifically about synthetic time-series generation, this is a lot.
>
> (A.1.2) We discuss extensively how IL is related:
>
> There is an entire Discussion section (Section 4) explicitly proposing that our method is reminiscent of imitation learning (mindful that they are---we must stress---different problems, but there is a notable resemblance that we point out). In this discussion, an explicit mapping of "Generation as Imitation" is provided (Corollary 6). After that, we point out explicitly how TimeGCI is analogous to imitation by IRL, and how methods from [59], [60], [61], [62], [63], [64], and [65] are related to ours (Lines 270-276). Moreover, we also point out how IL and synthetic time-series generation are also very different (Lines 277-283). Again, in a 9-page paper that focuses on solving the synthetic time-series generation problem, this is a lot.
>
> (A.1.3) The title of the paper gives credit to imitation learning:
>
> We chose the title "Time-series Generation by Contrastive Imitation" precisely to capture the fact that our mission is to bring together three separate strands of research: (1) synthetic time-series generation, (2) density estimation by contrastive learning, and (3) imitation learning---in particular, showing that a novel method for (1) inspired by (2) yields an analogy with (3). We strongly believe that given all of the above, we have given enough credit and relevance to the field of imitation learning.
>
> **(A.2) Existing EBM-based IL methods x2**:
>
> Thank you for suggesting [R1] and [R2] as additional related work. These are very interesting IL methods, and we will gladly include them. *However, note that they are very different from TimeGCI*---of course, in addition to the obvious fact that they do IL (which is---again---not our problem setting):
>
> (A.2.1) Different: state-action vs. trajectory-based:
>
> They maximize a lower bound on the negative reverse KL between the expert's and imitator's state-action occupancy measures. In general, while matching state-action occupancy measures should indirectly match the overall distribution of trajectories, they are not doing the same thing (this is actually simple to see: e.g. Lemma 2, Theorem 1 in [R3]). In the manuscript we also point out that TimeGAN [12] is also of this nature, that it "matches the induced distribution of sub-trajectories instead" (Lines 43-44), i.e. "state-action" distribution. In contrast, our explicit goal in this work is to match the distribution of trajectories directly (Lines 157-161). This distinction is important.
>
> (A.2.2) Different: alternating vs. two-staged optimization:
>
> In particular, [R1] takes a two-staged approach, which includes a (1) density estimation phase, followed by a (2) density matching phase based on maximum (occupancy) entropy RL. In contrast, we take an alternating optimization approach. Besides, while it is true that this method also uses an energy-based approach, the motivation, mathematics, and variational approach is completely different. *Just because an energy-based model is involved does not make the methods the same*. (Otherwise, practically every other IL method would be "the same" as each other!)
>
> (A.2.3) Different: [R2] does *not* use an energy-based approach:
>
> No, not at all. Nowhere in their entire paper does the word "energy" even appear, except in reference to Ziebart's paper. So we respectfully disagree that [R2] uses an energy-based model at all.

---

> ### Author Response · Authors · 2021-08-15
> **Response to Reviewer YqJQ [Part 2/8]**
>
> ---
>
> 3. **Empirical Evaluation**. In light of the above four points, we are also the first to show empirical viability of the resulting method---in the context of synthetic time-series methods and sequence modeling more broadly. We emphasize in particular:
>
> 	(3.1) **The results are good**. In the context of synthetic time-series generation (i.e. Table 1), our method is *consistently better* based on standard TSTR ("train-on-synthetic, test-on-real") metrics. This is the main takeaway. And within "sequence modeling" more broadly (cf. point 1.1 above re: [35], [72], [73], [80], etc.), we are the *first* to show viability of explicit policy-based generation in the general, continuous tabular setting.
>
> 	Note that since we are generating synthetic time-series data, the standard metric for evaluation is the *TSTR framework* (this is explicitly discussed in Section 5, viz. [12], [14], [21], [22], [26], [95]-[98]). This is yet another way in which we differ from IL evaluation, where the standard metric involves some "ground-truth" *reward*---which simply does not exist for synthetic time-series generation.
>
> 	(3.2) **Can we just "plug in" IL methods**? The short answer is "yes"---that's certainly *one way* to interpret Corollary 6, but there is no reason to expect that would perform the best. Kindly see our Response (B.2) below on imitation learning methods "plugged in" to synthetic time-series generation, where we further discuss differences and *empirically* verify that naively "applying" e.g. GAIL or AIRL in fact does worse than our proposed trajectory-centric approach. Both an extended version of that discussion and empirical results are now included in a (new) Appendix D and (new) Table 3.
>
> 	Ultimately, our mission is to bring together three separate strands of research: (1) synthetic time-series generation, (2) density estimation by comparison, and (3) IL---in particular, showing that a novel method for (1) inspired by (2) yields an analogy with (3). We have shown this performs better than methods in (1), or direct applications of (3). Shall *other methods* in IL be more cleverly designed/adapted to time-series generation? That is of course future work, but that is precisely the perspective we wish to convey.
>
> **Method of Exposition**:
>
> Of course, one can ask: "Why not just start by explaining the intricacies of IL proper, *then* "apply" / "adapt" it to synthetic time-series generation?"
>
> - **Short answer**: Because this is a work on synthetic time-series generation, aimed at the synthetic time-series generation community, where presumed knowledge and relevant benchmarks are on synthetic time-series generation methods (i.e. Table 1).
>
> - **Long answer**: Basically, we have two options:
>
> 	- *Option #1*: First, explain RL and IL, and explain the variety of modern IL methods and their motivations. Then---starting from IL---derive a novel algorithm (TimeGCI), showing that something resembling a trajectory-centric variation of an alternating energy-based algorithm works effectively for synthetic time-series generation.
>
> 	- *Option #2*: First, assume that the audience is familiar with synthetic time-series generation. Then---starting from first principles---derive a novel algorithm (TimeGCI), showing that it invites an analogy to imitation learning, whence we formally introduce the perspective of "generation as imitation", giving the required IL context.
>
> 	The first option suffers from two major flaws. First, it leaves little room to actually contextualize TimeGCI against standard synthetic time-series generation methods---*which is the proper context for the problem*. Second, it is *pedagogically inefficient*, as there are many practical differences between IL and time-series generation, e.g.
>
> 	- role of "environment" stochasticity/determinism,
> 	- Markovianity/non-Markovianity of settings,
> 	- relative cost of "environment" interaction,
> 	- existence/irrelevance of "ground-truth" rewards,
> 	- non-existence/importance of TSTR evaluation,
> 	- helpfulness/irrelevance of "done" indicators,
> 	- etc.
>
> 	---all of which have a major impact on the design and practical effectiveness of IL vs. synthetic time-series generation algorithms. But this first option starts from IL, which means we have to trudge through these nuances simply to justify the (much more fundamental) requirements of synthetic time-series generation. This would greatly obscure the essential motivations behind TimeGCI (viz. Section 3).
>
> 	The second option is what we have chosen, with flip-side advantages. First, starts by contextualizing TimeGCI against standard synthetic time-series generation methods (viz. Section 1 and Table 1). Second, it is more pedagogically efficient, as it does not presuppose/require external knowledge of IL until Section 4, whence more can be referenced or explained as required. (Of course, we do agree that there can always be more---additional parallels, connections, and comparisons can always be made. See below for our specific responses to those requested additions).

---

> ### Author Response · Authors · 2021-08-15
> **Response to Reviewer YqJQ [Part 1/8]**
>
> ---
>
> Thank you for your thoughtful comments and suggestions. We give answers to each of the following in turn, as well as pointing out corresponding updates to the revised manuscript:
>
> - (A) Originality and Significance
> - (B) Clarity
> - (C) Minor Issues
>
> ---
>
> ### Preface on Novelty
>
> ---
>
> Before we begin, kindly allow us to emphasize that we are expressly focused on the problem of **synthetic time-series generation**. We propose a method that improves on (state-of-the-art) generative models for general tabular **time-series data**---that is, the immediate context being TimeGAN [12], RC-GAN [21], COT-GAN [20], P-Forcing [10], etc. (viz. Table 1). To be clear, while we explicitly invite an analogy between time-series generation and imitation learning, we are **not** proposing an IL method.
>
> *IL and synthetic time-series generation are very different problems*. (See also our response titled "Method of Exposition" below, as well as the differences already enumerated in Section 4). The fact that we voluntarily draw a parallel between the two is an attempt to make connections between them, but not to imply that they are "one and the same". Presuming that any and all IL methods are *a priori* interchangeable with synthetic time-series generative models would be incorrect---and to be fair, we already reference **100** related works in both synthetic time-series generation and imitation learning.
>
> Purely in assessing novelty, our contributions are three-fold:
>
> 1. **Generation as Imitation**. We formalize the time-series generative modeling problem in terms of moment-matching to tackle compounding errors in open-loop sampling. The resulting method invites a formal analogy with imitation learning (viz. Corollary 6). Two important points:
>
> 	(1.1) **The analogy is novel**. Up till now, the time-series generation literature has run in a research track largely *separate* from imitation learning. See e.g. any of the literature in Table 1, as well as their respective related works. We are the first in attempting to *connect* the two communities together, proposing a novel method that explicitly identifies a parallel between them.
>
> 	No existing literature does this. Several come close in relating sequences to reinforcement learning, but have fundamentally different focuses, e.g. optimizing with *predefined* metrics [35], tackling *imputation* of missing data [72], predicting states in *markovian* settings [73], generating *discrete* language tokens [80], etc. (Note that these are all referenced in the Related Work already).
>
> 	(1.2) **The method is novel**. The vast majority of time-series generation methods---including those that specifically recognize the challenge of open-loop sampling---are motivated *heuristically* as variants of the standard GAN framework (see Table 1). We are the first in explicitly using a *moment-matching* argument to formalize this problem, giving a more principled motivation for our method.
>
> 	Of course, this is reminiscent of IL (hence the title of our paper). But despite the analogy, there are big differences (as noted in the Discussion). Methods designed for one problem are *not interchangeable* with the other (see point 3.2 below). In fact, the only IL work that proposed to directly match trajectory distributions (like us) has been found unworkable (see [64] Sec. 4, implementing [63]).
>
> 2. **Concrete Algorithm**. We propose an algorithm satisfying three key criteria as pertains sampling, evaluating, and learning. Then, we equate it with a special case of noise-contrastive estimation, which comes with theoretical guarantees. Two important points:
>
> 	(2.1) **The criteria are novel**. No existing method for synthetic time-series generation *simultaneously* satisfies all three desiderata (viz. Lines 51-59). Specifically, none simultaneously (1) respects trajectory-wise distributions during stepwise sampling, (2) provides generic measures of sample quality, and (3) can be trained non-adversarially (see Table 1 for a detailed comparison).
>
> 	Without doubt, learning an energy function in a sequential setting is closely related to inverse reinforcement learning, in particular MaxEnt IRL [61]-[63]. In fact, such relevant works are already referenced in the Related Work. However, again *we are not doing IL here*: A nuanced comparison with every single IL method is beyond the scope of this work on *synthetic time-series generation*.
>
> 	(2.2) **The NCE usage is novel**. Since we directly target the trajectory distribution, we are able to equate our structured classifier with an *adaptive-noise* case of NCE. This is new. While NCE is well-applied in text generation [76]-[78], these are limited to *conditional* next-word distributions using (external) input---entirely different from our (free-running) setting with *trajectory-wise* NCE.
>
> 	Even compared against IL (which---again---is *not* the problem we are solving), this is still new: For instance, while AIRL-based methods use a structured discriminator [64]-[65], they do so on a *state-action* basis, i.e. $d_{\theta,\phi}(z,u)$ $=$ $\exp(g_{\phi}(z,u))$ $/$ $(\exp(g_{\phi}(z,u))$ $+$ $\pi_{\theta}(u|z))$. This invalidates the relationship with NCE over trajectory distributions, hence *cannot* enjoy the guarantees that it yields.

---

> ### Author Response · Authors · 2021-08-20
> **Dear Reviewer YqJQ**
>
> ---
>
> We are sincerely grateful for your time and energy in the review process.
>
> In light of our responses (Aug 15) and appendix/manuscript updates (Aug 19), we would appreciate if the reviewer kindly let us know of any leftover concerns in the time remaining. We would be happy to do our utmost to address them.
>
> Thank you!
>
> Paper10659 Authors

---

> > ### Comment · Reviewer_YqJQ · 2021-08-20
> > **Response the the author response**
> >
> > Thank you for providing such a detailed response. I read through these in detail, and list some of the parts I am fully convinced of:
> > - "The NCE usage is novel"
> > - "Existing EBM-based IL methods x2"
> > - "Related works"
> >
> > I am not entirely convinced by the following points, and I will elaborate:
> > - "Why the method performs better" + "The bounds a different quantity": I understand your point in that *if* you are able to optimize the relevant quantity to be smaller than $\epsilon$, *then* accumulated error will be smaller. I am not entirely sure that *if* is a reasonable assumption when we control the modeling capacity to be the same: one is the error of one step, and the other is the error of all the steps, with the same hypothesis class, the latter is expected to be higher?
> >
> > - "Generation as Imitation": We can design a special environment such that the process of generating a time-series is an MDP. Suppose that original states are $s_t \in \mathcal{S}$ for time step $t$. In the MDP, the augmented states are $S_t = [s_0, \ldots, s_t]$, the action space are $\mathcal{S}$, and the transition probability is simply $S_{t+1} = concat(S_t, a_t)$. Here we have interpreted time series generation as an MDP, so it is a special case of imitation learning in this regard. Your policies are predicting future $s_t$ ("actions" in MDP) given previous $s$'s ("augmented state"), so I fail to see how they are "very different problems".
> >
> > - "Ablation study":
> >
> > The experiments in the paper already demonstrate that TimeGCI performs better than T-forcing in terms of prediction error, so I fail to understand how using different $T$ would add more to our understanding to the success of TimeGCI. If T-forcing does well in 1-step, then should it also do well simply by induction (good 1 step prediction leads to good 2 steps prediction and so on).
> >
> > Perhaps I did not explain this clearly in the review earlier: one experiment that I am curious about can be done under a more "synthetic" setup, suppose we can simulate the ground truth data (possibly with some noise), and we train both models using the same training data. Then during validation, we obtain a sequence from the simulator, then add some additional noise (say at time $K$); our task is to predict $t$ steps ahead with the learned models, and we can compare that with the "ground truth" (note that you likely cannot do it with real data since the steps beyond $K$ are based on noisy sequences). The errors here could demonstrate that your model is more "robust" than the baselines, and this could be why you accumulate fewer errors during sequence generation.
> >
> > I think you only need to address the ablation study question.
> > I don't think the novelty itself is grounds for rejection (it only drives my opinion away from a strong accepted paper), and I will not argue for the rejection of the paper.

---

> > > ### Author Response · Authors · 2021-08-25
> > > **RE: Response the the author response [Part 4/4]**
> > >
> > > ---
> > >
> > > **(RE.3.3) Second ablation study**:
> > >
> > > With your clarified request, it is now clear what you mean. We agree that this second ablation study---using a synthetic setup---illustrates your point very intuitively.
> > >
> > > **Simulation**: *Waves with Manually-Injected Noise*. To do so, we use a new synthetic simulation where what is being generated is a set of multi-dimensional sinusoidal waves which---if not perturbed by noise---correspond to different frequencies, phases, etc. Note that this does allow us to manually inject noise where we want, and also this allows us to simulate "ground truth", like you point out.
> > >
> > > **Experiment**: First, we train both models (T-Forcing and TimeGCI) using the same training data generated by the simulator, and save these learned models. Then during validation, we obtain a sequence from the simulator, then add some additional noise. Specifically, we add independent Gaussian noise $\mathcal{N}(\mu,\sigma)$, using $\mu=0$ and $\sigma=0.1$, to each of the feature dimensions of the wave at time $K$. Then, our task is to predict $t$ steps ahead with the learned models. To be clear, we ask the learned models to predict ahead using the information up to and including this (perturbed) $K$-th step; if the prediction is for $t>1$, we perform open-loop sampling as usual.
> > >
> > > **Assessment**: Since you specifically requested, here we shall measure the model's prediction error (i.e. evaluating how well the model can forecast the future based on the provided histories as input, along with the perturbation); recall from our discussion above that this is entirely different from the TSTR score we have been hitherto using (i.e. evaluating how well the model's freely generated output dataset preserves the characteristics of the input dataset). In this ablation study, we agree that this makes sense as a sensitivity analysis; by analogy to imitation learning, we can interpret this experiment as a special case of "action-matching" (i.e. based on ground-truth state/history as input, we can measure the "prediction error" between the action chosen by the imitator policy and the actual action taken by the expert)---but specifically where the state/history has been perturbed with additional noise.
> > >
> > > **Sensitivities**: We ask each model to perform $t$-steps ahead prediction, where $t\in\\{1,2,3,4,5\\}$ as a sensitivity. When we  add noise, we use values $\\{\sigma,2\sigma,3\sigma,4\sigma,5\sigma\\}$ as a sensitivity. Given any sequence, the step $K$ at which noise is artificially added (and at which point the models are asked to perform $t$-step ahead predictions) is uniformly randomly sampled from $K\in\\{1,2,...,T-1\\}$.
> > >
> > > | Noise at $K$ | Method | 1-Ahead MSE | 2-Ahead MSE | 3-Ahead MSE | 4-Ahead MSE | 5-Ahead MSE |
> > > |:--|:--|:--|:--|:--|:--|:--|
> > > |  $\sigma$ | T-Forcing | 0.024 ± 0.000 | 0.029 ± 0.001 | 0.036 ± 0.001 | 0.041 ± 0.001 | 0.047 ± 0.001 |
> > > |       *id*. | TimeGCI   | 0.024 ± 0.000 | 0.025 ± 0.001 | 0.026 ± 0.001 | 0.028 ± 0.001 | 0.031 ± 0.001 |
> > > | 2$\sigma $ | T-Forcing | 0.055 ± 0.000 | 0.059 ± 0.001 | 0.065 ± 0.001 | 0.071 ± 0.002 | 0.076 ± 0.001 |
> > > |       *id*. | TimeGCI   | 0.055 ± 0.001 | 0.055 ± 0.001 | 0.057 ± 0.001 | 0.059 ± 0.000 | 0.061 ± 0.001 |
> > > | 3$\sigma $ | T-Forcing | 0.106 ± 0.001 | 0.109 ± 0.002 | 0.116 ± 0.001 | 0.121 ± 0.001 | 0.127 ± 0.002 |
> > > |       *id*. | TimeGCI   | 0.106 ± 0.001 | 0.106 ± 0.001 | 0.107 ± 0.001 | 0.109 ± 0.001 | 0.111 ± 0.001 |
> > > | 4$\sigma $ | T-Forcing | 0.176 ± 0.001 | 0.181 ± 0.002 | 0.185 ± 0.001 | 0.191 ± 0.002 | 0.197 ± 0.002 |
> > > |       *id*. | TimeGCI   | 0.176 ± 0.002 | 0.176 ± 0.002 | 0.177 ± 0.001 | 0.179 ± 0.002 | 0.181 ± 0.001 |
> > > | 5$\sigma $ | T-Forcing | 0.266 ± 0.003 | 0.271 ± 0.002 | 0.277 ± 0.002 | 0.280 ± 0.002 | 0.287 ± 0.002 |
> > > |       *id*. | TimeGCI   | 0.266 ± 0.002 | 0.267 ± 0.003 | 0.267 ± 0.002 | 0.270 ± 0.003 | 0.272 ± 0.003 |
> > >
> > > **Results**: In your original review, you asked for further empirical evidence as to "why the proposed method is better -- is it because of better first-step prediction, or is it because of better robustness when having small errors in the previous steps". *This second ablation study answers that question*. Specifically, observe that:
> > >
> > > - (a) **Single-Step prediction**: There is virtually no difference between the performance of T-Forcing and TimeGCI when evaluating the 1-step ahead prediction MSE. The more noise added, the worse both models perform; but there is little difference between them, no matter the noise.
> > >
> > > - (b) **Multi-step prediction**: When predicting multiple steps ahead using open-loop sampling, T-Forcing performs worse than TimeGCI. In fact, the gap between their performances increases as $t$ increases. You can see this trend by reading "across" the table from left to right.
> > >
> > > **Conclusion**: This answers your original two-part question above. In particular, from (a) it appears that it is *not* the case that TimeGCI simply has better first-step prediction per se; also, from (b) it appears that TimeGCI *does* have better robustness when having errors in the previous steps. Because, while the 1-step performance of both T-Forcing and TimeGCI are impacted almost equally, TimeGCI appears to have an advantage in terms of "propagating" *less* of that error into later time steps.
> > >
> > > ***Update***: We have now included a version of this discussion in the (new) Appendix D.4. Moreover, the above empirical results are now also included in a (new) Table 5.
> > >
> > > ---
> > >
> > > Once agin, thank you for clarifying your questions and your request for ablation study. We hope that we have dispelled any remaining misunderstandings in the preceding. Please kindly let us know if you require any additional clarification.

---

> > > > ### Comment · Reviewer_YqJQ · 2021-08-25
> > > > **Thank you for the ablation studies.**
> > > >
> > > > Thank you for performing the ablation experiments. This clarifies my concerns about "why TimeGCI works", and I am willing to raise the score.
> > > >
> > > > Minor comment: I understand that you are performing time-series generation which is more difficult than prediction alone. However, the current experiments mostly evaluate MSE / cross-correlation metrics that can also be used with prediction models? I understand that having a more "holistic" metric can be challenging (e.g. FID in images, BLEU in natural language); it may be interesting as future work to consider contrastive methods that evaluate the "quality" of the generated trajectories.

---

> > > ### Author Response · Authors · 2021-08-25
> > > **RE: Response the the author response [Part 3/4]**
> > >
> > > ---
> > >
> > > ### (RE.3) "Ablation study"
> > >
> > > ---
> > >
> > > Thank you for clarifying your request. In the sequel, the three following points are explained in more detail:
> > >
> > > - (RE.3.1) *Response to your claim*: "Good 1 step prediction leads to good 2 steps prediction and so on".
> > > - (RE.3.2) *First ablation study*: Using different $T$ *does* actually add to our understanding, as we shall explain.
> > > - (RE.3.3) *Second ablation study*: The newly requested ablation study has been performed, as we shall describe.
> > >
> > > By way of preface, we must re-emphasize that *we are not doing time-series prediction*. This is already re-emphasized in our original response (A.3.1). To be triply clear, we are doing synthetic time-series generation (i.e. "generating a new dataset from scratch"), and not doing time-series prediction (i.e. "predicting forward values based on ground-truth history"). In particular, these correspond to two distinct evaluation measures that should not be conflated:
> > >
> > > **TSTR Predictive Score**. In evaluating a synthetic time-series generative model, the "*Predictive Score*" is the TSTR score computed using the synthetic dataset generated from scratch by the model, not conditioned on anything. We are evaluating the entire *output dataset* from the model. This is already explained in detail in the manuscript (see "Evaluation and Results" subsection within Section 5), and is moreover the standard method of evaluating synthetic time-series data generation. *Important note*: There is no equivalent to this in imitation learning. The graphical model is as follows:
> > >
> > > - Real Dataset --> Learned Model (e.g. TimeGCI) --> Synthetic Dataset
> > > - Synthetic Dataset --> TSTR predictive score
> > >
> > > **Model Prediction Error**. The above TSTR score (i.e. evaluating how well the model's output dataset preserves the characteristics of the input dataset) is entirely different from the prediction error of *the model itself* (i.e. evaluating how well the model can forecast the future based on ground-truth histories). This distinction is fundamental (see e.g. ICML 2021 tutorial on synthetic data generation). Thus far---and as is standard in synthetic data generation---we have not used the learned model to "predict" anything directly, conditioned on ground-truth input/history. If we did, the graphical model would be as follows:
> > >
> > > - Real Dataset --> Learned Model (e.g. Transformer)
> > > - Learned Model + Ground-truth history --> Predicted time-steps --> Prediction error
> > >
> > > Nevertheless, can we evaluate a generative model ***as if*** it were a prediction model (i.e. compute model prediction error of a synthetic time-series generative model)? The answer is "yes" (and we have done so below in Point (RE.3.3), as requested), but with a very strong caveat: Synthetic time-series generative models are not specifically optimized as predictive algorithms. They are optimized as generative algorithms. If time-series prediction were our central concern (which it is not), there exists a whole other universe of literature for doing that directly, entirely beyond the scope of this paper. So, while comparing the model prediction error (instead of TSTR metrics) can provide additional insight by way of "sensitivity analysis", it is not a *direct* method of assessing the synthetic time-series generative model.
> > >
> > > ---
> > >
> > > **(RE.3.1) Response to your claim**:
> > >
> > > You write:
> > >
> > > - "If T-forcing does well in 1-step, then should it also do well simply by induction (good 1 step prediction leads to good 2 steps prediction and so on)."
> > >
> > > We respectfully point out that this is incorrect.
> > >
> > > This is clearly false, simply because of *compounding error*. Learning a single-step predictor (e.g. T-Forcing, which only learns transition conditionals) may do well on a single-step basis, but behave more poorly on average during open-loop, multi-step sampling at test time---because it is only exposed to an exogenous dataset during closed-loop training (see Lines 123-127, 145-149 of the manuscript). The whole point of the paper is that "not all mistakes are equal" (Line 131): Some mistakes lead us further astray from the occupancy measure of the original dataset. Now, simply looking at the magnitude of the single-step score/error does not tell us what *kinds* of mistakes have been made, in this respect. So, looking at multi-step scores/errors always conveys more information than single-step scores/errors.
> > >
> > > Bottom line:
> > >
> > > - *Two methods may display similar magnitudes of single-step scores/errors, yet display very different magnitudes of multi-step scores/errors*.
> > >
> > > So no, the claim is incorrect: Good 1-step prediction *does not* automatically lead to good 2-steps prediction, etc. Moreover, there is no "induction" argument that is possible here!
> > >
> > > ---
> > >
> > > **(RE.3.2) First ablation study**:
> > >
> > > Recall the following as pertains T-Forcing vs. TimeGCI (among other differences):
> > >
> > > - T-Forcing: The learning objective does *not* use trajectory-level information at all (i.e. the fact that groups of transitions belong to the same trajectory). As a result, we suspect that T-Forcing is prone to multi-step compounding mistakes.
> > > - TimeGCI: The learning objective *does* use such trajectory-level information (and we use an energy model to help leverage it); recall our Response (RE.1) above. As a result, we hope that TimeGCI is less prone to this phenomenon.
> > >
> > > *The first ablation study tests for this effect*.
> > >
> > > If sequences are short, then the fact that groups of transitions belong to the same trajectory (i.e. what T-Forcing discards, and what TimeGCI does not discard) must not contain that much information. In other words, there is not much "long-term information" for there to be lost/gained, in shorter sequences. In the extreme case of $T=2$, a single transition pair contains the same information as the entire trajectory, so in that setting there is no extra information that TimeGCI can leverage.
> > >
> > > Please kindly refer back to the table of results in our original Response (B.5). Indeed, we observe that:
> > >
> > > - The performance advantage enjoyed by TimeGCI over T-Forcing diminishes as the sequence lengths of the input dataset decreases, and increases as the sequence lengths of the input dataset increases. This is true regardless of what metric we are talking about (i.e. 1/3/5-step TSTR score, or x-correlation score).
> > > - At $T=2$ there is almost no difference between the TSTR scores of T-Forcing and TimeGCI---which is as expected. (We see that there is still some tiny remaining discrepancy, but this is understandable since TimeGCI still has a different effective capacity by simultaneously learning an energy function).
> > >
> > > For these reasons, we politely maintain that varying $T$ as in this ablation study *does* in fact add to our understanding. Please kindly let us know if you require any further clarification.
> > >
> > > (Of course, as it turns out, this was not actually what you were requesting. Given your clarification as to your request, the second ablation study below performs what you requested).

---

> > > ### Author Response · Authors · 2021-08-25
> > > **RE: Response the the author response [Part 2/4]**
> > >
> > > ---
> > >
> > > ### (RE.2) "Generation as Imitation"
> > >
> > > ---
> > >
> > > Thank you for restating your question. You have articulated an MDP environment such that generating a time series is formulated as an MDP. But this is precisely a restatement of our Corollary 6 ("Generation as Imitation") in the manuscript, which we proposed. So it is not immediately clear what the criticism here is. There are two possibilities:
> > >
> > > - It is believed that, this "generation as imitation" analogy is not novel.
> > > - It is believed that, given this analogy, generation is "not different" from imitation.
> > >
> > > We respectfully disagree on both counts.
> > >
> > > ---
> > >
> > > **(RE.2.1) "Generation as Imitation" is Novel**:
> > >
> > > Please kindly refer to our original response titled "Preface on Novelty". There, we justify in detail---and with references and comparisons---the facts that:
> > >
> > > - This *analogy* is novel---as first to connect the largely separate research communities in synthetic time-series generation and IL; this is explained in detail in Point (1.1) in "Preface on Novelty";
> > > - The *proposed method* is novel---as first to use a moment-matching argument as principled motivation for an algorithm; this is explained in detail in Point (1.2) in "Preface on Novelty";
> > > - The *design criteria* are novel---as first to simultaneously satisfy all three desiderata identified in the motivation; this is explained in detail in Point (2.1) in "Preface on Novelty";
> > > - The *NCE usage* is novel---as first to equate our structured classifier with an adaptive-noise case of noise-contrastive estimation; this is explained in detail in Point (2.2) in "Preface on Novelty"; and
> > > - The *empirical demonstration* is novel---as first to show viability of such an approach in the general, continuous tabular setting; this is explained in detail in Point (3.1) in "Preface on Novelty".
> > >
> > > In your most recent response, you only acknowledge the point on "NCE usage" (i.e. point 2.2 in the preface)---which is only one of the above five points of novelty. Please kindly let us know if you disagree on any of the other points, or require any further clarification. We stand firmly by all our original arguments.
> > >
> > > ---
> > >
> > > **(RE.2.2) Learning Considerations are Different**:
> > >
> > > We must distinguish between *problem formalism* and *learning considerations*. Equivalence in the former does not imply equivalence in the latter: Even if two problems share formalisms on some level of abstraction, they may still be very different in terms of learning considerations.
> > >
> > > - *Simple Example*: The problem of learning a 2D convolutional classifier e.g. for images, is abstractly the same as the problem of learning a 1D convolutional classifier e.g. for audio. In fact, on the level of abstraction of "data as vectors", the problem formalism of the latter is a "special case" of the former. However, one can hardly argue that the study of the latter is the same as the study of the former! The learning considerations are entirely different between the two domains, with different emphases on what is important, different evaluation metrics, and different practical considerations.
> > >
> > > Now, this much is clear: Corollary 6 casts the synthetic time-series generation problem as an IL problem, and you also articulate an MDP environment that is precisely identical to a restatement of Corollary 6. So, using the IL *problem formalism*, it is certainly true that synthetic time-series generation can be formulated as a "special case"---indeed, this is precisely one of the main points we make in the paper.
> > >
> > > However, it is incorrect to assume that the *learning considerations* are therefore the same. They are not. We already enumerate multiple concrete distinctions between synthetic time-series generation and imitation learning in our original response titled "Method of Exposition" as well as Section 4 of the differences already enumerated in Section 4. In the following, we focus on the most salient differences that influence the design of actual learning algorithms:
> > >
> > > - (RE.2.2.1) *How Performance Considerations affect design*. Synthetic time-series generative models are per standard evaluated by the *TSTR framework* (see Section 5, viz. [12], [14], [21], [22], [26], [95]-[98]). The learned model generates synthetic data, and the synthetic data is evaluated based on how faithful it is to the original dataset. So, capturing $p(\tau)$ is of paramount importance. In IL, this notion does not exist, and is in any case not applicable. In IL, the learned model itself is deployed and evaluated based on average *ground-truth reward*. So, capturing $p(s,a)$ is sufficient. In synthetic time-series generation, there is no "ground-truth reward". It is easy to see that capturing $p(s,a)$ alone is insufficient to preserve $p(\tau)$.
> > >
> > > - (RE.2.2.2) *How "Environment" Interaction affects design*. In synthetic time-series generation, the "environment" is *deterministic* (viz. Corollary 6) and rollouts are *free* (i.e. requires no interaction with any actual environment). This means that *trajectory-based* algorithms are more likely to succeed because there is no additional variance associated with a stochastic environment, and moreover we are not constrained by the real cost of interaction with an actual environment. In IL, the environment is usually *stochastic* and rollouts are costly. This means that *state-action based* algorithms are more likely to do better, because there is additional variance from stochastic transitions, and online sample complexity is a practical limitation.
> > >
> > > - (RE.2.2.3) *How Optimization is affected by the design*. As a result of both of the above factors, the design of TimeGCI is explicitly based on the objective of learning a trajectory distribution $p(\tau)$ directly. As we propose, this allows us to enjoy the *formal guarantees* of optimization provided by the noise-contrastive estimation through the adaptive-noise formulation. In contrast, in IL, successful algorithms (e.g. [64, 65, R2, R5, R6, R7, etc.]) typically use a state-action objective, which loses information (see e.g. Lemma 2, Theorem 1 in [R3]). Specifically, this means that the success of any similar structured discriminators in IL are more heuristic---since that formulation *invalidates* the relationship with NCE over trajectory distributions.
> > >
> > > - (RE.2.2.4) *Empirical Comparison*. We can also ask a very practical question: As a matter of fact, can we "plug in" IL methods to perform synthetic time-series generation? As explained in Point (3.2) in our original response titled "Preface on Novelty", the answer is yes, but they do not perform as well---which is as we would expect, from the reasons given above. In fact, we have now verified that this is indeed the case. Please kindly see our original Response (B.2) on imitation learning methods "plugged in" to synthetic time-series generation, where we further discuss differences and empirically verify that naively "applying" e.g. GAIL or AIRL in fact does not do as well as our proposed trajectory-centric approach.
> > >
> > > For these reasons, synthetic time-series generation and IL are *indeed* very different---despite Corollary 6. Most importantly, these differences mean that what kinds of algorithms work best for each problem are different---and TimeGCI is designed accordingly.
> > >
> > > In your most recent response, you only acknowledge the point on "NCE usage" (here, this corresponds to point RE.2.2.3 above). Please kindly let us know if you disagree on any of the other points, or require any further clarification. We stand firmly by all our original arguments.

---

> > > ### Author Response · Authors · 2021-08-25
> > > **RE: Response the the author response [Part 1/4]**
> > >
> > > ---
> > >
> > > Thank you for clarifying your questions and your request for ablation study. In the following, we give answers to each of the following:
> > >
> > > * (RE.1) "Why the method performs better" + "The $\epsilon$ bounds a different quantity"
> > > * (RE.2) "Generation as Imitation"
> > > * (RE.3) "Ablation study"
> > >
> > > **Note**: The further newly requested ablation study has been performed as specified, and is now included in the revised Appendix D.4 to the manuscript; see Response (RE.3.3) for details.
> > >
> > > ---
> > >
> > > ### (RE.1) "Why the method performs better" + "The $\epsilon$ bounds a different quantity"
> > >
> > > ---
> > >
> > > Thank you for the thoughtful clarification. Please kindly allow us to dispel two misconceptions.
> > >
> > > First, as pertains the two bounds, you refer to the "global discrepancy for *T* steps in Lemma 2, as opposed to local discrepancy for 1 step in Lemma 1" (cf. "Official Review of Paper10659 by Reviewer YqJQ"). Similarly, you state that "one is the error of one step, and the other is the error of all the steps" (cf. "Response the author response").
> > >
> > > - This is incorrect, as we shall explain in (RE.1.1).
> > >
> > > Second, you state that "when we control the modeling capacity to be the same [...] with the same hypothesis class, the latter is expected to be higher".
> > >
> > > - This is also incorrect, as we explain in (RE.1.2).
> > >
> > > ---
> > >
> > > **(RE.1.1) Exposure bias $\neq$ naive *T*-step error**:
> > >
> > > There are two different concepts here that must not be conflated: (1) First, we may speak of the distinction between bounding *global vs. local* discrepancies; this pertains to Lemmas 1 and 2 in the paper. (2) Separately, we may speak of bounding *one step vs. T-step* discrepancies; however, this distinction is not meaningful, and in any case is not what we are doing.
> > >
> > > Recall "$\epsilon$" from the premises of Lemmas 1 and 2. Let us call them "$\epsilon_{1}$" and "$\epsilon_{2}$" in the following:
> > >
> > > $$
> > > \textstyle\max_{f}\big(E_{h\sim\mu_{s};x\sim\pi_{s}(\cdot|h)}f(h,x)-E_{h\sim\mu_{s};x\sim\pi_{\theta}(\cdot|h)}f(h,x)\big)\leq\epsilon_{1}
> > > $$
> > >
> > > $$
> > > \textstyle\max_{f}\big(E_{h\sim\mu_{s};x\sim\pi_{s}(\cdot|h)}f(h,x)-E_{h\sim\mu_{\theta};x\sim\pi_{\theta}(\cdot|h)}f(h,x)\big)\leq\epsilon_{2}
> > > $$
> > >
> > > They only differ in one respect: The second expectation is over $h\sim\mu_{s}$ for the $\epsilon_{1}$-bound, whereas it is over $h\sim\mu_{\theta}$ for the $\epsilon_{2}$-bound. The former is "local", because the expectation conditions on $h$ drawn from the (exogenous) dataset alone, and the actual rollout distribution of $\pi_{\theta}$ is nowhere to be seen (cf. Lines 123-127). The latter is "global", because the expectation conditions on $h$ drawn from the (endogenous) occupancy measure $\mu_{\theta}$ from rolling out $\pi_{\theta}$ (cf. Lines 145-149).
> > >
> > > So the key idea is *exposure bias*: The $\epsilon_{1}$-bound suffers from it, but the $\epsilon_{2}$-bound does not.
> > >
> > > Note that this has nothing to do with how many "steps" are involved in the bounds! In particular, it is incorrect to say that the first bound is "one step" vs. the second bound being "*T* steps". Again, the bounds differ precisely---and only---in the *distributions* from which $h$ is drawn (viz. exposure bias). So they are different quantities, but it is certainly false to say that the second bound somehow contains *T* steps of error in the premise to begin with (clearly, $\epsilon_{2}$ is *not* bounding anything *T* times as large as $\epsilon_{1}$).
> > >
> > > In sum:
> > >
> > > - What Lemma 2 is saying: If you can eliminate **exposure bias**, you get a better bound.
> > > - What Lemma 2 is *not* saying: If you can eliminate ***T*** **times as much error**, you get a better bound.
> > >
> > > Note that the first statement is meaningful: By using trajectory-level information, we can take concrete steps to minimize exposure bias, such as the proposed algorithm does; see Response (RE.1.2) below. In contrast, the second statement is simply a tautology: Naively, we can---of course---multiply both sides of the $\epsilon_{1}$-bound by *T* to obtain a "*T* step" bound, but if we only use transition-level information, minimizing this is indeed *T* times as difficult.
> > >
> > > ---
> > >
> > > **(RE.1.2) Minimizing $\epsilon_{2}$ is not necessarily harder**:
> > >
> > > Lemmas 1 and 2 show that minimizing $\epsilon_{2}$ gives a better result than identically minimizing $\epsilon_{1}$. Now, all else equal, a bound that yields a better result is generally expected to be "harder" to achieve than a bound that yields a worse result. However, the key condition here is *all else equal*. In the context of Lemmas 1 and 2, **all else is *not* equal**:
> > >
> > > - Lemma 1's premise (and optimization Eq. 3) only involves expectations over $\mu_{s}$ (i.e. the logged data).
> > > - Lemma 2's premise (and optimization Eq. 5) involves expectations over $\mu_{\theta}$ (i.e. the generated data) as well.
> > >
> > > So, which $\epsilon$ is more difficult to minimize depends precisely on the relative difficulty of estimating the expectations over $\mu_{s}$ and $\mu_{\theta}$. For example,
> > >
> > > - In imitation learning, we may be accustomed to the notion that "rollouts" are costly (i.e. they involve interacting with an environment, which takes time and resources). With limited access to open-loop samples, approximating $\mu_{\theta}$ well can be difficult. Indeed, if we already have access to a large enough dataset drawn from $\mu_{s}$, simple behavioral cloning may do better than others.
> > > - In synthetic time-series generation, however, "rollouts" are free (i.e. they do not involve interacting with any environment, even in real-world data settings). Access to open-loop samples is virtually infinite, with the only limitation being training-time computation. If we can approximate $\mu_{s}$ as well as we can approximate $\mu_{\theta}$, minimizing $\epsilon_{2}$ is not necessarily harder than $\epsilon_{1}$.
> > >
> > > In sum, *a priori* there is no reason to believe that one $\epsilon$ is easier/harder to minimize than the other, since they depend on our ability to approximate different quantities. (Moreover, note that since $\Delta\bar{F}_s(\theta)$ $\in$ $O(T^{2}\epsilon_1)$ and $\Delta\bar{F}_s(\theta)$ $\in$ $O(T\epsilon_2)$, therefore all that is actually required is simply for $\epsilon_2<T\epsilon_1$ to be true, in order that the second approach give a better (i.e. sub-quadratic) result than the first).
> > >
> > > Concretely, the fact that "all else is not equal" is also reflected in the learning algorithms themselves:
> > >
> > > - **Online vs. Offline**: T-Forcing is "offline", relying only on the logged dataset for learning, i.e. involving only closed-loop training. TimeGCI is "online", relying not just on the logged dataset, but also on the freedom to perform its own open-loop sampling during training, from which it has access to approximating expectations over its own rollout distributions.
> > > - **Information Used**: T-Forcing really only requires datasets of the form $\mathcal{D}=\\{(h_{n},x_{n})\\}$, i.e. it discards the information contained in the fact that groups of transitions belong to the same trajectory. TimeGCI requires datasets of the form $\mathcal{D}=\\{\tau_{n}\\}$, i.e. it additionally leverages the information contained in the fact that groups of transitions belong to the same trajectory.
> > > - **Modeling Capacity**: T-Forcing and TimeGCI both optimize a policy function, within the same hypothesis class and with the same *representational* capacity. However, TimeGCI additionally optimizes an energy function during training---which is what takes advantage of the additional information not used by T-Forcing. Hence the *effective* capacity of the two approaches is different---by design. This distinction between representational and effective capacity here is very important (see e.g. Section 5.2 of Goodfellow's deep learning text for a discussion of this difference).
> > >
> > > All of these are valid reasons for hypothesizing that the proposed method may perform better.
> > >
> > > ---
> > >
> > > We hope that we have sorted out any remaining misunderstandings in the preceding, and that these points may be better interpreted alongside our original clarifications as to "why the method performs better" than existing synthetic time-series generation methods; see our original Responses (B.1) and (C.1). Please kindly let us know if you require any additional clarification.

---

### Author Response · Authors · 2021-08-19
**Revised Paper**

---

Thank you again to the reviewers for their generous comments and suggestions. Most of the feedback has resulted in straightforward clarifications that we have now aggregated into a **(new) Appendix D**.

We agree that including these remarks---and auxiliary results---for reference more precisely delineates our context and contributions. Overall, these emphasize the fact that, while we explicitly invite an *analogy* between synthetic time-series generation and IL, we are *not* proposing an IL-applicable method. IL and synthetic time-series generation are very different problems with different considerations (see e.g. our response titled "Method of Exposition" to Reviewer YqJQ, as well as the differences already enumerated in Section 4).

In addition, we have included additional remarks on related work to accompany the lemmas in **Appendix A**, as well as several clarifying comments in the main manuscript.

**Note**: While we cannot actually *upload* these revisions, all the content is already present in our original responses---which we specifically reference below.

---

### (New) Appendix D: "Clarifying the Analogy: Generation as Imitation"

---

[Reviewer QAxC]:

- **Appendix D.1** ("Can TimeGCI be used for Imitation Learning?"): This includes an extended version of our
Response (A) for Reviewer QAxC, which clarifies the similarities and differences between the objectives and methods for time-series generation and imitation learning.

[Reviewers QAxC, WGYR, and YqJQ]:

- **Appendix D.2** ("Can AIL be used for Time-series Generation?"): This includes an extended version of our Response (B) for Reviewer QAxC, Response (A) subtitled "Adversarial IRL" for Reviewer WGYR; and Response (B.2) for Reviewer YqJQ. Moreover, the newly obtained empirical results for GAIL and AIRL applied directly to synthetic time-series generation are now respectively included in (new) Tables 3(a) and 3(b).

[Reviewers WGYR and YqJQ]:

- **Appendix D.3** ("Further Comparisons with Related Work"): First, this includes a portion of our Response titled "Preface on Novelty" for Reviewer YqJQ, which explicitly re-iterates of our positioning and contributions with respect to synthetic time-series generation, including the novelty of the analogy itself, our key criteria, and the proposed method. Second, with respect to a specific comparison with [63], this also includes an extended version of our Response (A) for Reviewer WGYR. Third, with respect to EBMs and NCE more broadly, this includes a version of our Responses (A.2) and (A.4) for Reviewer YqJQ.

[Reviewer YqJQ]:

- **Appendix D.4** ("Ablation Study on Sequence Lengths"): This includes an extended version of our Response (B.5) for Reviewer YqJQ, which performs an ablation study to provide sensitivities for understanding. Moreover, the newly obtained empirical results are now included in a (new) Table 4.

---

### (Updated) Appendix A: "Proofs of Propositions"

---

[Reviewers QAxC and YqJQ]:

- **Revised Remarks for Lemma 1**: Per our Response (C) subtitled "Lemma 1 Related Work" for Reviewer QAxC, as well as our Response (C.2) for Reviewer YqJQ, we have now expanded the remarks accompanying the proof for Lemma 1, referencing existing variations on such bounds (e.g. with respect to probability of mistake, sample complexity, etc.) to give better context. Moreover, the manuscript now points to these remarks in the statement of the lemma.

[Reviewer QAxC]:

- **Revised Remarks for Lemma 2**: Similarly, per our Response (C) subtitled "Lemma 2 Related Work" for Reviewer QAxC, we have now expanded the remarks accompanying the proof for Lemma 2, referencing existing variations on such bounds (e.g. with respect to divergence measures, sample complexity, etc.) to give better context. Likewise, the manuscript now points to these remarks in the statement of the lemma.

---

### Miscellaneous Clarifications

---

[Reviewer QAxC]:

- **Reasoning of Equations**: Per our Response (C) for Reviewer QAxC, we have now clarified the reasoning around Eq. 3/5 as well as Eq. 6, and included a note in Lemmas 1 and 2 to clarify their premise and interpretation.

[Reviewer WGYR]:

- **Rewording in Prop. 5**: Per our Response (B) for Reviewer WGYR, we have now revised Proposition 5 in the main text to state the more general form, along with the caveat from Appendix A.

- **Information on datasets**: Per our Response (C) for Reviewer WGYR, we have now included the additional information on datasets in a new table plus accompanying remarks (i.e. a version of the response) in Appendix C.

- **Improving scalability**: Per our Response (D) for Reviewer WGYR, we have now included a longer discussion of scalability and future work (i.e. a version of the response) at the end of Appendix C.

[Reviewer YqJQ]:

- **Explanation of "non-adversarial"**: Per our Response (B.6) for Reviewer YqJQ, on Line 59 (Section 1), we have now added a (new) footnote to the bullet point, next to the word "non-adversarially", which explains precisely the distinction suggested.

- **Connection with IPMs**: Per our Response (C.3) for Reviewer YqJQ, on Lines 130 and 152 (Section 2), we have now added references to a (new) footnote, which explains the suggested connection with IPMs.

---

With our clarifications and updates, we hope that we have addressed the reviewers' concerns. Thank you for your kind consideration.

---

### Decision · Program_Chairs · 2021-09-27

**Decision:**

Accept (Poster)

**Comment:**

After very detailed replies by the authors all reviewers of the paper have recommended acceptance.